# Description of a global marine particulate organic carbon-13 isotope data set

Maria-Theresia Verwega[1,2], Christopher J. Somes[1], Markus Schartau[1], Robyn E. Tuerena[3], Anne Lorrain[4], Andreas Oschlies[1], and Thomas Slawig[2]

[1]GEOMAR - Helmholtz Centre for Ocean Research, Kiel, Germany
[2]Kiel University, Kiel, Germany
[3]Scottish Association for Marine Science, Dunstaffnage, Oban PA37 1QA
[4]Univ Brest, CNRS, IRD, Ifremer, LEMAR, F-29280 Plouzané, France

**Correspondence:** csomes@geomar.de

**Abstract.** Marine particulate organic carbon stable isotope ratios ($\delta^{13}C_{POC}$) provide insights in understanding carbon cycling through the atmosphere, ocean, and biosphere. They have for example been used to trace the input of anthropogenic carbon in the marine ecosystem due to the distinct isotopically light signature of anthropogenic emissions. However, $\delta^{13}C_{POC}$ is also significantly altered during photosynthesis by phytoplankton, which complicates its interpretation. For such purposes, robust spatio-temporal coverage of $\delta^{13}C_{POC}$ observations is essential. We collected all such available data sets, merged and homogenized them to provide the largest available marine $\delta^{13}C_{POC}$ data set (Verwega et al., 2021). The data set consists of 4732 data points covering all major ocean basins beginning in the 1960s. We describe the compiled raw data, compare different observational methods, and provide key insights in the temporal and spatial distribution that is consistent with previously observed large-scale patterns. The main different sample collection methods (bottle, intake, net, trap) are generally consistent with each other when comparing within regions. An analysis of 1990s median $\delta^{13}C_{POC}$ values in an meridional section across the best covered Atlantic Ocean shows relatively high values ($\geq -22‰$) in the low latitudes ($< 30°$) trending towards lower values in the Arctic Ocean ($\sim -24‰$) and Southern Ocean ($\leq -28‰$). The temporal trend since the 1960s shows a decrease of median $\delta^{13}C_{POC}$ by more than $3‰$ in all basins except for the Southern Ocean, which shows a weaker trend, but contains relatively poor multi-decadal coverage.

## 1 Introduction

Carbon is an essential element for life and it regulates climate via its atmospheric form $CO_2$, a long-living greenhouse gas. Understanding carbon cycling is fundamental to reliably project changes of the Earth's future climate. Carbon is subject to transformation and cycling throughout the ocean, land and atmosphere. It is a major part of organic matter of all living organisms which can both consume (e.g. photosynthesis) and produce (e.g. respiration) inorganic carbon. Besides the natural cycling processes, the total amount and distribution of carbon is strongly perturbed by human activity caused by the industrialization, most notably due to fossil fuel emissions, deforestation, farming, cement production and other industrial processes. Anthropogenic $CO_2$ emissions are one of the main driving forces of modern climate change which is likely to continue in the

future (IPCC, 2013). Only about 60 % of the anthropogenic $CO_2$ emissions have been compensated by natural sinks, including the dissolution of inorganic carbon in the ocean. This leaves the atmosphere enriched with anthropogenic carbon already by about 880 Gt $CO_2$ since 1750 (IPCC, 2014), which is driving the increase of global temperature levels. The ocean serves as an important buffer, as it absorbs a significant amount of anthropogenic carbon, with the ocean interior being the largest readily exchangeable reservoir of carbon in the Earth system.

Marine phytoplankton convert dissolved inorganic carbon (e.g. aqueous $CO_2$) into their organic carbon via photosynthesis in the euphotic surface layer. This organic carbon forms the base of the food web for higher tropic levels in marine ecosystems. Some particulate organic carbon (POC) sinks down to ocean depths, where it is either respired back to dissolved inorganic carbon by heterotrophic organisms or becomes buried in ocean sediments (Suess, 1980). This process is known as the soft-tissue biological carbon pump, an important mechanism for sequestering carbon to the deep ocean from the atmosphere (Volk and Hoffert, 1985; Banse, 1990; McConnaughey and McRoy, 1979). Since the deep ocean has a residence time of about a millennium, it is a key carbon reservoir influencing long-term climate change.

Carbon isotopes provide additional insights into the cycling of carbon in the Earth system (Zeebe and Wolf-Gladrow, 2001). The element carbon exists in two naturally occurring stable isotopes, $^{12}C$ and $^{13}C$, with abundances of around 98.9 % and 1.1 %, respectively. Knowledge of their pathways through carbon reservoirs can support deeper understanding of carbon transfer and can help identify carbon sources with different isotopic ratios (Rounick and Winterbourn, 1986). Relative abundances of carbon isotopes are usually given as the $\delta$-notation, which is based on the carbon isotope ratio $\frac{^{13}C}{^{12}C}$, standardized and given in parts per thousands as

$$\delta^{13}C = \left( \frac{\frac{^{13}C}{^{12}C}}{R_{std}} - 1 \right) \tag{1}$$

The constant $R_{std} = 0.0112372$ is a standard ratio, originally referring to the calcareous fossil PeeDee Belmnite. The values $^{12}C$ and $^{13}C$ are the absolute concentrations of the individual isotopes (Hayes, 2004).

Distributed within the carbon cycle the fractionation of $\delta^{13}C$ is influenced by biological and thermodynamic processes (Gruber et al., 1999). Air-sea gas exchange plays a dominant role at the ocean surface. Phytoplankton photosynthesis and POC remineralization increase their influence in the ocean interior (Gruber et al., 1999; Morée et al., 2018). The processes are dependent on circulation and temperature and thus their individual influence vary with geographic location (Gruber et al., 1999; Schmittner et al., 2013).

Phytoplankton preferentially incorporate (i.e. fractionate) the lighter $^{12}C$ carbon isotope into its organic matter. This fractionation causes phytoplankton organic $\delta^{13}C$ to be 10 to 25 ‰ lower than that of inorganic $\delta^{13}C$, which depends on a variety of environmental, ecological, and physiological conditions (e.g. Popp et al., 1989, 1998; Rau et al., 1989, 1996). The main factors that control phytoplankton fractionation are concentrations of $CO_2\,[aq]$, species-specific effects enforced by the phytoplankton composition, and cellular growth rate, although uncertainties remain regarding the quantification of the specific processes and mechanisms that cause variations in phytoplankton fractionation (e.g. Fry, 1996; Laws et al., 1995; Popp et al., 1998; Bidigare et al., 1997; Cassar et al., 2006).

$\delta^{13}\mathrm{C}_{POC}$ provides insights into physical and biological carbon cycle processes in the ocean (e.g. Fry and Sherr, 1989). It helps to diagnose carbon pathways from the atmosphere to the deep ocean including the biological carbon pump (e.g. Jasper and Hayes, 1990; Popp et al., 1989; Freeman and Hayes, 1992), assists reconstruction of oceanic carbon cycling and even plankton cell size and community structure (e.g. Tuerena et al., 2019; Lorrain et al., 2020). For example, anthropogenic carbon emissions have a distinctly low $\delta^{13}\mathrm{C}$ content, making $\delta^{13}\mathrm{C}$ a useful property for tracing anthropogenic carbon throughout the Earth system (Eide et al., 2017; Levin et al., 1989; Ndeye et al., 2017). Atmospheric $\delta^{13}\mathrm{C}_{CO_2}$ has decreased from -6.5 ‰ in preindustrial times to -8.4 ‰ presently (Rubino et al., 2013). The measurable decrease due to anthropogenic fossil carbon emissions is known as the Suess Effect (Keeling, 1979), which enters the ocean via air-sea gas exchange. However, changes in marine $\delta^{13}\mathrm{C}_{POC}$ are also significantly influenced by changes in phytoplankton fractionation due to other anthropogenic controls. For example increasing $CO_2\,[aq]$ concentrations increase surface $\delta^{13}\mathrm{C}$ fractionation (Young et al., 2013), changing phytoplankton communities and increasing temperature influences phytoplankton growth rates and $\delta^{13}\mathrm{C}$ fractionation over the air-sea interface (Zhang et al., 1995). But determination of the driving processes(es) of $\delta^{13}\mathrm{C}_{POC}$ spatial and temporal trends remains a challenge. We also stress that all of these processes are sensitive to temperature changes which adds additional complexity to understanding how fractionation may change in space and time. A better understanding of the contributions from all of these effects requires a robust global data set of $\delta^{13}\mathrm{C}_{POC}$.

Theoretical projection and understanding of changes associated with $\delta^{13}\mathrm{C}_{POC}$ can be executed by models of different scales, which include $\delta^{13}\mathrm{C}_{POC}$ circulation. Earth system models serve to simulate and test hypotheses in different scenarios as unbiased assessments (e.g. IPCC, 2014) and may support future decision making. Besides resolving mass flux of carbon, many models also simulate stable carbon isotopes (e.g. Schmittner and Somes, 2016; Buchanan et al., 2019; Hofmann et al., 2000; Jahn et al., 2015; Tagliabue and Bopp, 2008; Morée et al., 2018; Magozzi et al., 2017). For reliable calibrations and validations of such processed-based mechanistic models, a spatially and temporally comprehensive data set is essential. This additional constraint provided by marine $\delta^{13}\mathrm{C}_{POC}$ assists reconstruction of oceanic carbon cycling including how much anthropogenic carbon is entering marine ecosystems and exported to the deep ocean. But until today, there is a lack of suitable data sets as constraints. This results in large and mostly unknown uncertainties in model results.

Data sets of marine $\delta^{13}\mathrm{C}_{POC}$ improve our understanding of marine carbon cycling by providing another independent constraint. Recent model approaches support long-term past climate projections (Tjiputra et al., 2020) and assess estimations of the Suess effect (Liu et al., 2021). To date, numerous individual $\delta^{13}\mathrm{C}_{POC}$ data sets exist, while the number of accessible, merged data sets is lacking. Existing merged data sets contain data from several sources but were often focused on a specific region or process (e.g. Goericke, 1994; Tuerena et al., 2019). Individual data sets are usually collected during a specific cruise or time series station and are often neglected since they contain relatively few data. Such data sets can easily be accessed on data platforms such as PANGAEA and, when combined, they can represent an important and significant source of data.

In this study, we provide a novel merged seawater $\delta^{13}\mathrm{C}_{POC}$ data product (Verwega et al., 2021), that – to our knowledge – contains the most expansive spatio-temporal coverage to date. It contains all available $\delta^{13}\mathrm{C}_{POC}$ seawater data from PANGAEA and the merged data sets by Goericke (1994), Tuerena et al. (2019) and Young et al. (2013), as well as unpublished data from different cruises by Lorrain. No data were excluded, even if sampled at extreme locations (e.g, trenches, hydrothermal vents).

The meta-data comprise information about sampling location, time, depth and method as well as the original source, which makes original raw data values, method, and further technical description easily accessible. Provided data files are NetCDF files interpolated onto two different global grids and a csv file that includes the data and their anomalies with respect to their overall mean together with all corresponding available meta information.

The paper is structured as follows: we provide a brief overview of $\delta^{13}\mathrm{C}_{POC}$ data acquisition in section 2 and their compilation and metadata in section 3. The characteristics of the collected $\delta^{13}\mathrm{C}_{POC}$ data are shown in section 4. We present their spatial distribution in section 5 and temporal distribution in section 6. Lastly, we provide a short summary and concluding remarks.

## 2 Data acquisition

The data set includes 4732 entries for $\delta^{13}\mathrm{C}_{POC}$ from 185 different sources and ranges from the 1960s to the 2010s. In addition to many data sets from the data platform PANGAEA, we included unpublished data provided by Lorrain and the data products from Tuerena et al. (2019), Goericke (1994) and Young et al. (2013). The adjustments that we conducted are described in the following.

### 2.1 Data sources

As a basis of our data set, we chose the 1990s data collection by Goericke (1994). This was established to investigate variations in $\delta^{13}\mathrm{C}_{POC}$ with temperature and latitude. The $\delta^{13}\mathrm{C}_{POC}$ sample data and measurements were conducted by investigating zooplankton, net-plankton or particulate organic matter. We cross-checked and extended this data set by looking up all available primary sources. Goericke originally included 476 of $\delta^{13}\mathrm{C}_{POC}$ data points from 17 contributions. Largest contributions came from Fischer (1989) with 107 entries, Fontugne et al. (1991) with 97 and Fontugne and Duplessy (1981, 1978) with 78. Large extensions were possible e.g. in the Fischer (1989) and Eadie and Jeffrey (1973) data sets, incorporating more than 70 additional data points from these primary sources. With this extension, we could increase the data set to 626 data points for $\delta^{13}\mathrm{C}_{POC}$.

We collected most data from the PANGAEA data platform, an open access online library archiving and providing geo-referenced Earth system data, hosted and monitored by the Alfred-Wegener-Institut - Helmholtz Center for Polar and Marine Research (AWI) and the Center for Marine Environmental Sciences, University of Bremen (MARUM). With the data made available therein, we could further extend the data set by additional $\approx 3{,}500$ measurements of $\delta^{13}\mathrm{C}_{POC}$. Most $\delta^{13}\mathrm{C}_{POC}$ data from PANGAEA are associated with samples collected during the Joint Global Ocean Flux Study (JGOFS), with more than 2000 of $\delta^{13}\mathrm{C}_{POC}$ data points. Additional 529 samples are contributions by the Antarctic Environments Southern Ocean Process Study (AESOPS), 342 by the Archive of Ocean Data (EurOBIS Data Management Team) and 279 by the SFB313 (Thiede et al., 1988).

Other collected data were provided by Tuerena and Lorrain. Tuerena provided a data contribution coming from the data set mentioned in Tuerena et al. (2019), to which we will refer to as the Tuerena data set. This contains 595 data points including 501 from Young et al. (2013) and covers samples within the euphotic zone and an observation timeframe of 1964 - 2012.

Moreover, we included 69 unpublished data points provided by Lorrain, covering the years 2012 - 2015 and sampled during the cruises CASSIOPEE, PANDORA, OUTPACE, NECTALIS 3 and 4 and KH13. We refer to this data set as the Lorrain data set.

A recent collection of 303 measurements of $\delta^{13}C_{POC}$ has been provided by Close and Henderson (2020), largely based on data gathered from individual publications referenced therein. Since our analyses originally relied on data sources that differed from those of Close and Henderson (2020) we find our collection to be yet incomplete. Especially measurements from national data bases might provide a huge future benefit.

## 2.2 Adjustments made

All data were taken with as many details as possible from the sources and have eventually been reshaped to fit the structure. No rounding or cut off of detailed data were made. Spatial coordinates originally given as depth intervals were replaced by their respective mid points. Time intervals were not changed in this way. If they contained just one month or year this was taken, otherwise the time information was omitted. Sample depth given as "surface" was denoted as 1 m. Longitude values were converted to the format $[-180°, 180°]$ by the transformation

$$
Lon_{new} = \begin{cases} Lon_{old} - 360° & \text{for all } Lon_{old} \in (180°, 360°] \\ Lon_{old} & else \end{cases} \tag{2}
$$

Wherever possible the data were taken from their original publication. Changes made to the data by Goericke are described in Table 1, changes to all other data in Table 2. The complete structure is presented in Table 3.

Most data listed in the Goericke data set could be gathered from the original publications directly. Some data are not accessible from an original source, including those data labeled as "Harrison", "Hobson" and "Schell", which were included as unpublished data by personal communication in Goericke (1994). Also, we could not identify the original data sources of "Voss (1991)" and "Sackett et al. (1966)". Data from these sources are used as provided by Goericke. All other data could be directly compared with and linked to their origin. According to Table 3 we complemented the data with month, year, depth, sample method, cruise, trap duration and a references, wherever available. Special notes given in Goericke (1994) were conserved in our "project/cruise" named meta information. Rounded values were adjusted to their source values as well as data with interchanged longitudinal information, which is in detail shown in Table 1.

In two cases we identified multiple $\delta^{13}C_{POC}$ data sets from a single event (time, place, investigator), where the data had been subject to different stages of processing or different types of measurements: In Westerhausen and Sarnthein (2003), we chose the "mass spectrometer" data set because this was the originally measured one. In Trull and Armand (2013a) and in Trull and Armand (2013b), we used the "blanc corrections" data set of $\delta^{13}C$, since this set of $\delta^{13}C_{org}$ values is recommended to be considered (Trull and Armand, 2001).

The primary source of the Tuerena and Lorrain data was mentioned in our data set in the "Project/cruise" column. In the data set from Tuerena et al. (2019), this was originally labeled as "source", in the Lorrain data set as "campaign". In both data sets the Longitude was converted to $[-180°, 180°]$ from a $[0°, 360°]$ format by Equation 2. In the data of MacKenzie et al. (2019)

**Table 1.** Changes that were introduced to data taken from Goericke (1994): the first column names the publication or author of the primary data set. The second column lists in which part of the data we applied changes. The third and fourth columns show from what values to which values they have been changed and the last columns gives the reason for this.

| data set | changed | from | to | reason |
|---|---|---|---|---|
| Degens et al. (1968) | Lon | Goericke | source value | E, W interchanged |
| Eadie and Jeffrey (1973) | Lon | Goericke | source value | E, W interchanged |
| Fischer (1989) | Lon | Goericke | source value | E, W interchanged |
| Fontugne and Duplessy (1978) | Lon | Goericke | source value | E, W interchanged |
| Fontugne and Duplessy (1981), MD13 Osiris III | Lon | Goericke | source value | E, W interchanged |
| Francois et al. (1993) | Lon | Goericke | source value | E, W interchanged |
| Harrison[1] | Lon | Goericke | source value | E, W interchanged |
| Sacket et al. (1965) | Lon | Goericke | source value | E, W interchanged |
| Saupe et al. (1989) | Lon | Goericke | source value | E, W interchanged |
| Wada et al. (1987) | Lon | Goericke | source value | E, W interchanged |
| Eadie and Jeffrey (1973) | Lat, Lon | Goericke | source value | rounded in Goericke |
| Fischer (1989) all, but INDOMED leg-12 | Lat, Lon | Goericke | source value | rounded in Goericke |
| Fontugne and Duplessy (1978) | Lat, Lon | Goericke | source value | rounded in Goericke |
| Fontugne and Duplessy (1981) | Lat, Lon | Goericke | source value | rounded in Goericke |
| Francois et al. (1993) | Lat, Lon | Goericke | source value | rounded in Goericke |
| Sacket et al. (1965) | Lat, Lon | Goericke | source value | rounded in Goericke |
| Eadie and Jeffrey (1973) | $\delta^{13}C_{POC}$ | not included | added | not included in Goericke |
| Fischer (1989) | $\delta^{13}C_{POC}$ | not included | added | not included in Goericke |
| Sacket et al. (1965) | $\delta^{13}C_{POC}$ | not included | added | not included in Goericke |
| Wada et al. (1987) | $\delta^{13}C_{POC}$ | not included | added | not included in Goericke |
| Fischer (1989) | $\delta^{13}C_{POC}$ | Goericke | source value | rounded in Goericke |
| Fontugne and Duplessy (1978) | $\delta^{13}C_{POC}$ | Goericke | source value | rounded in Goericke |
| Fontugne and Duplessy (1981) | $\delta^{13}C_{POC}$ | Goericke | source value | rounded in Goericke |
| Fischer (1989) | temperature | Goericke | source value | rounded in Goericke |
| Fontugne and Duplessy (1981) | temperature | Goericke | source value | rounded in Goericke |
| Francois et al. (1993) | temperature | Goericke | source value | rounded in Goericke |
| Sacket et al. (1965) | temperature | Goericke | source value | rounded in Goericke |
| Fischer (1989) | $\delta^{13}C_{POC}$ | Goericke | deleted | not found in source |
| Fontugne and Duplessy (1978) | temperature | Goericke | deleted | not found in source |

[1]The original source was not available, but we highly suspected an error in the coordinates interchanged East and West.

**Table 2.** Changes made in other data: this table's structure is equivalent to Table 1. It refers to all changes made in general and any other than the Goericke (1994) data.

| data set | changed | from | to | reason |
|---|---|---|---|---|
| any | depth | "surface" | 1 | comparability |
| any | depth | depth range | average[1] | comparability |
| Trull and Armand (2013a) | $\delta^{13}C_{POC}$ | three available | "blank correction" | mentioned in Trull and Armand (2001) |
| Trull and Armand (2013b) | $\delta^{13}C_{POC}$ | three available | "blank correction" | mentioned in Trull and Armand (2001) |
| any using sediment traps | month, year | range | explicit value[2] | comparability |
| Chang et al. (2013) | month, year | range | explicit number | just one date for trap sampling given |
| Lorrain | Project/cruise | | "campaign" | provided by Lorrain |
| Tuerena | Project/cruise | | "source" | provided by Tuerena |
| Tuerena | Lon | $[0°, 360°]$ | $[-180°, 180°]^{3}$ | comparbility |
| Lorrain | Lon | $[0°, 360°]$ | $[-180°, 180°]^{3}$ | comparbility |
| MacKenzie et al. (2019) | depth | original | deleted | suspected typo |
| Voss and von Bodungen (2003) | trap duration | original | deleted | suspected typo |
| De Jonge et al. (2015a) | Method | multiple investigations (MULT) | in-situ pump | found in De Jonge et al. (2015b) |

[1] By arithmetic mean.

[2] Only for sample durations entirely within an explicit month and year, otherwise information of time frames has been discarded.

[3] We applied Equation 2.

we deleted a typo where the depth value was set equal to the negative Longitude value. We disregarded trap duration given in Voss and von Bodungen (2003), which was given as the negative value $-1$.

## 3 Content and structure of the data set

The data collection is made available in files of raw and interpolated values respectively (Verwega et al., 2021). The raw data are a csv file that includes the $\delta^{13}C_{POC}$ measurements, their anomalies with respect to their mean and all available meta information. The interpolated data are provided as NetCDF files on two different global grids: a $1.8° \times 3.6°$-resolution and 19 depth layers from a model that simulates $\delta^{13}C_{POC}$ (e.g. Schmittner and Somes, 2016), in the following referred to as the UVic grid, and the $1° \times 1°$-resolution and 102 depth layer grid of the World Ocean Atlas (Garcia et al., 2018), in the following referred to as the WOA grid. Interpolation required availability of full spatial information (latitude, longitude and depth) of included $\delta^{13}C_{POC}$ data to locate them on the grid.

On the WOA grid we provide thirteen NetCDF files containing only data with full spatio-temporal metadata: One is, averaging all observations from each year together, each year accounting for a time increment on the time axis. The other twelve files are averaging only observations from an individual month with again each year accounting for a time increment on the time axis. These files provide a variety of analysis opportunities, but also limited content of $\delta^{13}C_{POC}$ data.

On the UVic grid we provide seven individual NetCDF files: Six of them are each representing one of the decades 1960s to 2010s containing all data, which were able to assign to their respective decade. One file contains all available $\delta^{13}C_{POC}$ data completely independent of their measurement time. This individual provision of data on a decadal and overall time scale increases the fraction of usable $\delta^{13}C_{POC}$ data for the following analyses.

## 3.1 Raw data file

The csv-format data file includes $\delta^{13}C_{POC}$ measurements, anomalies and meta information in its columns. A full description of the content, value range and coverage of the individual columns is given in Table 3. Anomalies of $\delta^{13}C_{POC}$ were calculated, based on the arithmetic mean of the full data collection. The mean was calculated, rounded to two digits after the floating point and used as

$$mean_{\delta^{13}C_{POC}} = -23.96\text{‰} \tag{3}$$

Anomalies contain all relevant information with respect to variability of the $\delta^{13}C_{POC}$ data in space and time. This way it becomes easier to analyze bias information separately, e.g. during first steps of model calibration.

The reference includes the citations as detailed as possible. Wherever available, this is taken from the original source. Otherwise, we tried to include author, title, publication year and platform and doi. For unpublished data like Harrison's from the Goericke's data set or those included by the coauthors, we denoted from where we took the data.

Coordinates are given in decimal over $[-90°, 90°] \times [-180°, 180°]$. The sample depth is given in meters measured positively from the ocean surface downwards. Data having been published as measured at $0$ m were included as this, while no surface micro layer measurements were included. Month and year were used to describe the sample date, specific days are neglected.

Anomalies of $\delta^{13}C_{POC}$ are given in the $\delta$-ratio described in Equation 1. A sample method was added, wherever available. Any special sampling circumstances were given in the "Note" column. Activity duration of sediment traps was denoted in the last column.

The "Origin" columns listed the associated project or cruise or author's note. Some samples were given with multiple project connections, all of them were given in this column.

## 3.2 Interpolated data sets

The interpolated $\delta^{13}C_{POC}$ data are available as Network Common Data Form (NetCDF) files on two global grids with different resolutions. NetCDF files are machine-independent and support creation, accessing and sharing of array-oriented scientific data. On the UVic grid, we provide seven different files each of them independent of time and averaged over the available spatial information. Six of them contain an individual decade each (from the 1960s through the 2010s). The seventh file comprises a combined set of all interpolated $\delta^{13}C_{POC}$ data. On the WOA grid, we provide thirteen files including all $\delta^{13}C_{POC}$ measurements with complete spatial-temporal information, averaged across times and space.

One major aim of this work is to support reliable validation and calibration of $\delta^{13}C_{POC}$-simulating models. Hence, we chose the grid of the UVic model version 2.9, as used e.g. in Schmittner and Somes (2016). Horizontally, it consists of $100 \times 100$ cells

**Table 3.** Available data and meta information: the columns of the raw data set correspond to the provided data and meta information. Their names are given in the first column of this table. The second holds a short description of their content, the third their ranges of values. In the final column we give how well this data kind is covered relative to the size of the full data set.

| column | content | range of values | coverage[1] |
|---|---|---|---|
| Reference | citation[2] | description | full[3] |
| No | running index | $\{1, ..., 4732\}$ | full |
| Lat | latitude in decimal[4] | $[-90°, 90°]$ | 4604 / 4732 |
| Long | longitude in decimal[4] | $[-180°, 180°]$ | 4604 / 4732 |
| d13C | $\delta^{13}C_{POC}$[4] | $[-55.15, -4.5]$ | full |
| d13Canomaly | $\delta^{13}C_{POC} - mean_{\delta^{13}C_{POC}}$[5] | $[-31.19, 19.46]$ | full |
| Temp | temperature in °Celsius[4] | $[-1.8, 31.12]$ | 1622 / 4732 |
| Month | month as number | $\{1, ..., 12\}$ | 4114 / 4732 |
| Year | years A.D. | $\{1964, ..., 2015\}$ | 4483 / 4732 |
| Depth | depth in m | $[0, 4850]$ | 3917 / 4732 |
| Method | measurement method of $\delta^{13}C_{POC}$ | description | 3164 / 4732 |
| Origin | associated project or cruise | description | 3921 / 4732 |
| Note | special circumstances, if | description | 140 / 4732 |
| Trap duration | duration of trap activity in days | $[1, 133]$ | 533 / 587[6] |

[1] Ratio of available entries relative to the full number of data points.

[2] Wherever possible, this includes: author(s), year, title, journal name, full, number, issue, pages and doi.

[3] Primary source was not available in every case as a reference. A note, where the data were taken is included in this case.

[4] With as many decimal places as available.

[5] Rounded to two decimal places.

[6] Here, abundance is given relative to the full number of sediment trap samples.

with a resolution of $1.8° \times 3.6°$, arranged from 0 to $360°$ in longitude (LON) and $-90$ to $90°$ in latitude (LAT). Vertically, it is split up into 19 vertical layers (DEPTH), decreasing in resolution with depth. The two uppermost layers reach down to depths of 50 and 130 m respectively, and they are supposed to comprise the upper ocean's euphotic zone.

The the WOA grid is based on the $1° \times 1°$ grid of the World Ocean Atlas (Garcia et al., 2018). It has a horizontal resolution of 205    360 arranged from $-180$ to $180°$ in longitude (LON) and 180 arranged from $-90$ to $90°$ in latitude (LAT) direction. Vertically, it is split up into 102 layers (DEPTH). The time axis (TIME) increments for each year from 1964 to 2015 by one and has a size of 52. This interpolation only includes $\delta^{13}C_{POC}$ data with full spatio-temporal metadata coverage, i.e. additional to latitude, longitude and depth, we also required and included year and month information.

FERRET scripts were used for the interpolations. These averaged the irregularly measured data points within the ocean 210    grid to one single data point representing each covered grid cell. The interpolation function SCAT2GRIDGAUSS by NOAA's Pacific Marine Environmental Laboratory performed the spatial averaging under PyFerret v7.5. Calculations in this function

are based on a work by Kessler and McCreary (1992) and can be summarized as follows: let $(x_1, y_1), ..., (x_n, y_n) \subseteq \mathbb{R}^2$ be an equidistant grid and $(\tilde{x_1}, \tilde{y_1}), ..., (\tilde{x_m}, \tilde{y_m}) \subseteq \mathbb{R}^2$ be irregular measurement locations of a real tracer $D_j, j \in \{1, ..., m\}$. Then the value $D_i \in \mathbb{R}$ at grid point $(x_i, y_i), i \in \{1, ..., n\}$ becomes interpolated as

$$D_i := \frac{\sum_{j=1}^{m} D_j W_{i,j}}{\sum_{j=1}^{m} W_{i,j}} \tag{4}$$

where

$$W_{i,j} := \begin{cases} 0; & \tau_{i,j} < e^{-CX} \\ 0; & \tau_{i,j} < e^{-CY} \\ \tau_{i,j}; & else \end{cases} \tag{5}$$

with $\tau_{i,j} := \exp\left(-\left(\frac{(x_j - x_i)^2}{X^2} + \frac{(y_j - y_i)^2}{Y^2}\right)\right)$ is the Gaussian weight function and $X, Y \in \mathbb{R}$ are scaling arguments and $C \in \mathbb{R}$ the cut-off parameter. We set to $X = 1.8$, $Y = 0.9$ and $C = 1$ in the our script.

Since the interpolation into the WOA grid excluded all data without full spatio-temporal metadata coverage, we focus following descriptions of interpolated data on the UVic grid interpolations. These also include data without month-information in the six decadal files and even completely without temporal information in the seventh time-independent file.

## 4 Main data set characteristics

The final data set includes 4732 individual $\delta^{13}\text{C}_{POC}$ measurements of seawater samples. We show the distribution of $\delta^{13}\text{C}_{POC}$ values by Gaussian kernel density estimation (KDE) in Figure 1. KDEs are a non-parametric density estimation (Silverman, 1986) for approximation of probability density functions, which is theoretically similar to a histogram but with a continuous curve not dependent on rigid intervals. We applied a Python implementation from the SciPy stats-package (Virtanen et al., 2020) to create the results presented here. Likewise, we derived conditional probability densities of $\delta^{13}\text{C}_{POC}$ values, given the different measurement method applied (Figure 3).

### 4.1 Range and outlier values

The data distribution is presented by its KDE in Figure 1. The interval of $\delta^{13}\text{C}_{POC}$ values ranges over $[-55.15, -4.5]$ with a mostly smooth distribution. Most of our data exhibit values around $\delta^{13}\text{C}_{POC} \approx -24‰$, which becomes clearly identifiable as a single maximum in the KDE. Two smaller modes are visible at around $\delta^{13}\text{C}_{POC} \approx -27.5‰$ and $\delta^{13}\text{C}_{POC} \approx -22‰$ (see also Table A1 in the Appendix). A steep decline to zero is visible outside the two outer modes. The steep decline of the KDE stops at around $\delta^{13}\text{C}_{POC} = -37‰$ and $\delta^{13}\text{C}_{POC} \approx -14‰$. Between $\delta^{13}\text{C}_{POC} \approx -37‰$ and $\delta^{13}\text{C}_{POC} \approx -55.15‰$

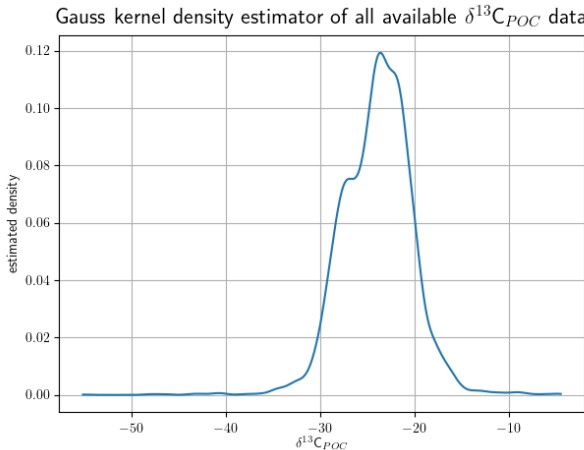

**Figure 1.** The density function of all individual $\delta^{13}C_{POC}$ measurements approximated by Gaussian kernel density estimation: values of the estimated density are drawn on the y-axis, the $\delta^{13}C_{POC}$ values run on the x-axis. The higher the value of the estimated density is, the more $\delta^{13}C_{POC}$ points have been measured around this value.

as well as between $\delta^{13}C_{POC} \approx -14‰$ and $\delta^{13}C_{POC} \approx -4.5‰$ the KDE closely aligns to the x-axis, which indicates very little data points lie in this range.

Below $\delta^{13}C_{POC} = -37‰$ we find 17 data points ranging down to $\delta^{13}C_{POC} = -55.15‰$. Down to $\delta^{13}C_{POC} = -48‰$ these were all taken from Lein and Ivanov (2009) and Lein et al. (2006), measured in September or October 2003, around the location $10°$ N, $104°$ W and below 2500 m depth in the vicinity a hydrothermal field close to the Pacific coast of middle America. The lowest outlier at $\delta^{13}C_{POC} = -55.15‰$ was taken from Altabet and Francois (2003a) from November 1996 and at $62.52°$ S, $169.99°$ E at the ocean surface south from New Zealand.

Above $\delta^{13}C_{POC} = -10‰$ we find 15 data points ranging up to $\delta^{13}C_{POC} = -4.5‰$. Three of them were taken from Lein et al. (2007) and measured at 800 m depth at a hydrothermal vent located $30.125°$ N, $42.117°$ W in the middle north Atlantic. Ten were taken from Calvert and Soon (2013b, c, a). All of these were measured between 636 and 901 m depth around $49°$ N, $130°$ W close to the American coast of the Pacific and all of them in February or May, but one in August. The final two were part of the Lorrain data set. Both were measured at the ocean surface in the south Pacific, in July at $5.3°$ S, $164.9°$ E, and December at $20.9°$ S, $159.6°$ E.

Since more than 98 % of the data (4668 of the 4732 data points) have values that lie between $\delta^{13}C_{POC} = -35‰$ and $\delta^{13}C_{POC} = -15‰$, we will focus on this range in our following analyses.

We tested the robustness of our KDE approach in a subsampling experiment. We considered 500 random subsets of 20 % of the original data over the range with the highest data density $[-35, -15]$ and visualize their KDEs in Figure 2. They show peaks at $\delta^{13}C_{POC} \approx -23‰$ fitting the maximum and the second smaller mode right from it, and at $\delta^{13}C_{POC} \approx -27.5‰$. Outside $[-27, -22]$ the KDEs are closely aligned. Mean and standard variation of the KDE ensemble also show the highest variability around the two modes at $\delta^{13}C_{POC} = -23‰$ and $\delta^{13}C_{POC} = -27.5‰$.

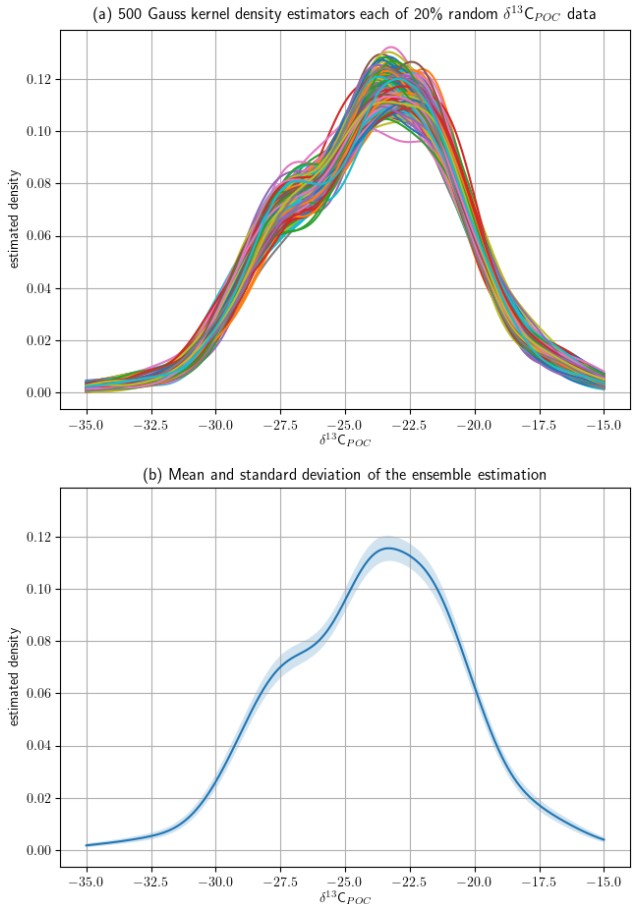

**Figure 2.** A random sample of 20% of the $\delta^{13}C_{POC}$ data was taken from the full data set for 500 times to generate an ensemble of subsets. Their densities were approximated with a Gaussian kernel density estimator. (a) shows all 500 estimated densities by individual lines. (b) shows the mean and the variance of the full ensemble of densities by a graph and the shaded area around it, respectively.

## 4.2 Sampling methods

Various sampling methods were involved in obtaining the $\delta^{13}C_{POC}$ data. Around 67 % of the data had associated sampling method information, which were contributed by eighteen different sampling methods. In principle, all eighteen methods could be grouped into five main observational types: bottles, intake, nets, traps and diverse. "Bottle" data include samples taken from Niskin bottles, PEP bottles, CTDs and samples collected via Seabird submersible pumps. By "intake" we refer to all versions of pumps, underway cruise track measurements, as well as Multiple Unit Large Volume Filtration System (MULVFS). "Net" data represent all occurring versions of plankton nets and traps all represented sediment traps and moorings. Finally, the deep sea manned submersible (MIR2) is not classified to any of these groups and was assigned to a cluster that we refer to as "diverse".

260

All sample devices provided data over all sample depths. Deeper samples were mainly taken from traps and pump systems, the upper from bottle and net data. Most data sampled deeper than 2600 m were collected by sediment traps. At 3800 m there were several trap contributions by Calvert (e.g. Calvert, 2002), mostly from the late 1980s. Data sampled by a deep-sea manned submersible were given at locations down to 2520 m (Lein and Ivanov, 2009).

For resolving differences between sampling methods we chose data from the Atlantic Ocean, which comprise all four major methods (with data embracing a region between $45°$ S and $80°$ N and $70°$ W and $20°$ E). In addition, data were distinguished between tropical, temperate, and polar subregions. By crudely sorting the data according to their sampling locations, we gain some insight to methodological variability within a subregion and may relate these to variations between the three subregions (Figure 3). Overall, we do not find any severe bias with respect to any particular method. Bottle data seem to cover most of the lower $\delta^{13}C_{POC}$ values that typically range between $-28‰$ and $-21‰$, which could be due to samples collected at greater depths. Intake and net measurements are rather restricted to the upper ocean layers and these methods often yield $\delta^{13}C_{POC}$ larger than $-25‰$, with some polar net measurements being a notable exception (Figure 3d). For the tropical Atlantic ($30°$ S - $30°$ N) the net and intake measurements vary around $-21‰$, with 95 % confidence limits between $-24‰$ and $-18‰$ (see Table A2 in the Appendix). According to our comparison, we could not identify any method that yields much greater variance of $\delta^{13}C_{POC}$ values than others. The spatio-temporal variations of the $\delta^{13}C_{POC}$ compare well amongst different methods, but we advise caution when comparing bottle measurements with data of other methods because of potential differences in the depth range covered.

In the full Atlantic Ocean, densities of intake and net data are most representative of the maximum full $\delta^{13}C_{POC}$ sample. From the intake data shown here, $\approx 80$ % were sampled within $30°$ S and $30°$ N. When restricting to this area, net data better resemble the full data. Net sample data were by $\approx 80$ % collected between $30°$ N and $60°$ N, where they fit the overall $\delta^{13}C_{POC}$ density best, followed by trap data. Trap and bottle data deliver lowest $\delta^{13}C_{POC}$ measurements in the Atlantic Ocean. Both data kinds were with $\approx 74$ to 85 % sampled north from $60°$ N. A restriction to this area shows trap and bottle samples being closely aligned to the full data in this region.

The variance of the intake and trap data is $\approx 3‰$ and lower than the variance of all $\delta^{13}C_{POC}$ together, which is $\approx 5‰$, the highest value observed here. Bottle and net data both show a variance less than $2‰$. Furthermore, trap, net and full $\delta^{13}C_{POC}$ show a pronounced second mode in their densities, while bottle and net data show a clear individual maximum. Median values of net and intake data are $\approx 1$ to $\approx 2‰$ higher than the one of the full data, respectively. This has a median of $\delta^{13}C_{POC} = 22.46‰$. Bottle and trap data show both a $\approx 2‰$ lower median. Analytical errors and uncertainties are typically $0.2‰$ or lower (Young et al., 2013), and thus are not likely to significantly contribute to the much larger variance in the observations

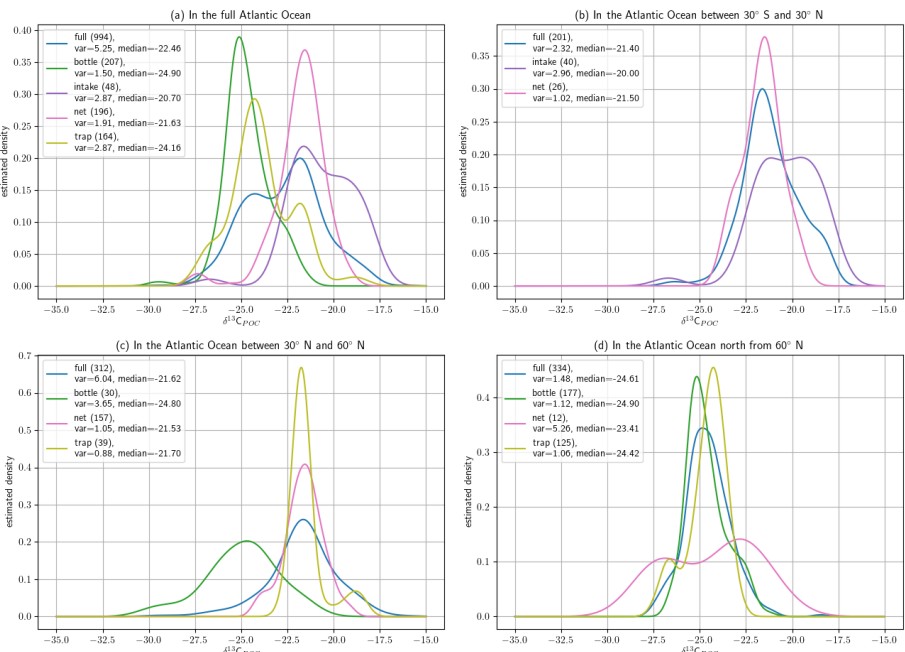

**Figure 3.** Separation of $\delta^{13}C_{POC}$ in the Atlantic Ocean data by four main sample methods: bottle, intake net and trap data. (a) shows the full Atlantic Ocean, (b) the equatorial core of the Atlantic Ocean, (c) the Atlantic between $30°$ S and $30°$ N and (d) its most northern area. In each plot, the density of the $\delta^{13}C_{POC}$ sample groups with enough data was approximated by Gaussian KDEs and drawn with an individual color. An additional graph shows the comparison to the full $\delta^{13}C_{POC}$ data density in the respective area. The numbers of used data points are indicated in each KDE label.

## 5 Spatial distribution

We show the spatial distribution of $\delta^{13}C_{POC}$ measurements across the global ocean surface and depths. Most $\delta^{13}C_{POC}$ data have been measured in the uppermost few ocean meters and best surface coverage is available for the Atlantic Ocean. Changes in $\delta^{13}C_{POC}$ on the ocean surface were evaluated based at the UVic grid.

### 5.1 Vertical distribution of the data set

Depth values are available for more than 80 % of the sample data locating most of them in the upper ocean.The distribution
of depth measurements is shown in Figure 4. An approximation of the depth measurements by Gaussian KDE is visualized in Figure 5 along with the $\delta^{13}C_{POC}$ value distribution over the them in the main ocean basins. The KDE resolves best data coverage for the uppermost $\approx 500$ m of the oceans and a second far smaller maximum at $\approx 3800$ m. The depth ranges presented in Figure 4 correspond to the depth intervals of the UVic grid, only the two uppermost layers are presented in more detail and the last four are combined. Within the first 130 m we observe highest data density and find nearly 2500 measurements of

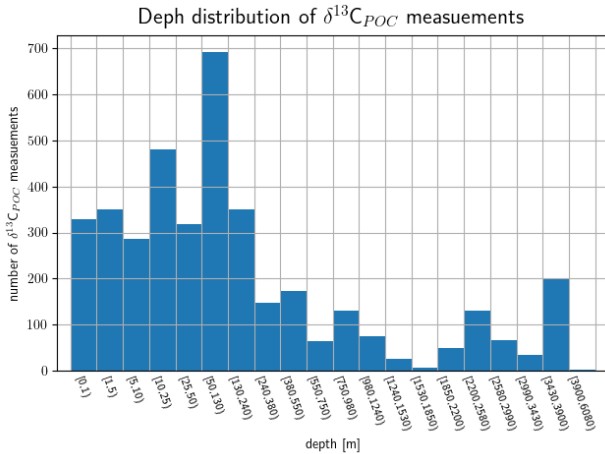

**Figure 4.** Vertical data coverage in depth layers based on the UVic grid: The uppermost 50 m are divided in subranges, below they are according to the UVic grid. The number of $\delta^{13}C_{POC}$ data points available are plotted against their respective depth range.

$\delta^{13}C_{POC}$, where nearly 1000 of them were measured within $[0\,\text{m}, 10\,\text{m}]$. 200 $\delta^{13}C_{POC}$ values were available in the depth interval $[3430\,\text{m}, 3900\,\text{m}]$. The two deepest values were taken from the Fischer (1989) and Altabet and Francois (2003b) and sampled at 4500 and 4850 m depth, respectively.

Values of $\delta^{13}C_{POC}$ are, apart from the North Pacific, closely aligned within the individual ocean basins. The Atlantic, South Pacific and Indian Ocean show values mostly of $-28‰$ to $-19‰$. The $\delta^{13}C_{POC}$ values in the Arctic reach down to $\approx -30‰$ and those in the Southern Ocean even to $\approx -35‰$. The North Pacific shows a wide spread of $\delta^{13}C_{POC}$ values, especially between 50 and 100 m depth and at 2500 m depth. There they either reach down to less than $-40‰$ or up to more than $\approx -10‰$ at a depth of 2500 m.

Measurements in the North Atlantic, North Pacific and Indian Ocean reach down to more than 3500 m. Measurements down to nearly 5000 m were sampled in the Southern Ocean. The South Pacific was sampled down to a depth of 2500 m and the Arctic Ocean and South Atlantic only in the uppermost few hundred m.

## 5.2 Horizontal distribution of the data set

All global oceans are covered with $\delta^{13}C_{POC}$ data. In Figure 6 the horizontal distribution of available data is depicted for both grids. For the UVic grid we show data from the file including all data independent of time, the WOA grid is averaged over all times. In both cases, we averaged data over all depths and also added data without a depth information to best visualize the horizontal coverage. A similar plot, but with a different purpose, is given later in this work in Figure 10 showing only surface data locations.

Many cruises are visible as lines formed by connected grid cells in Figure 6, especially in the Atlantic and Indian Ocean and shorter in the Southern Ocean. Also, smaller sample spots occur, mainly located in the Pacific, Arctic and Southern Ocean. The

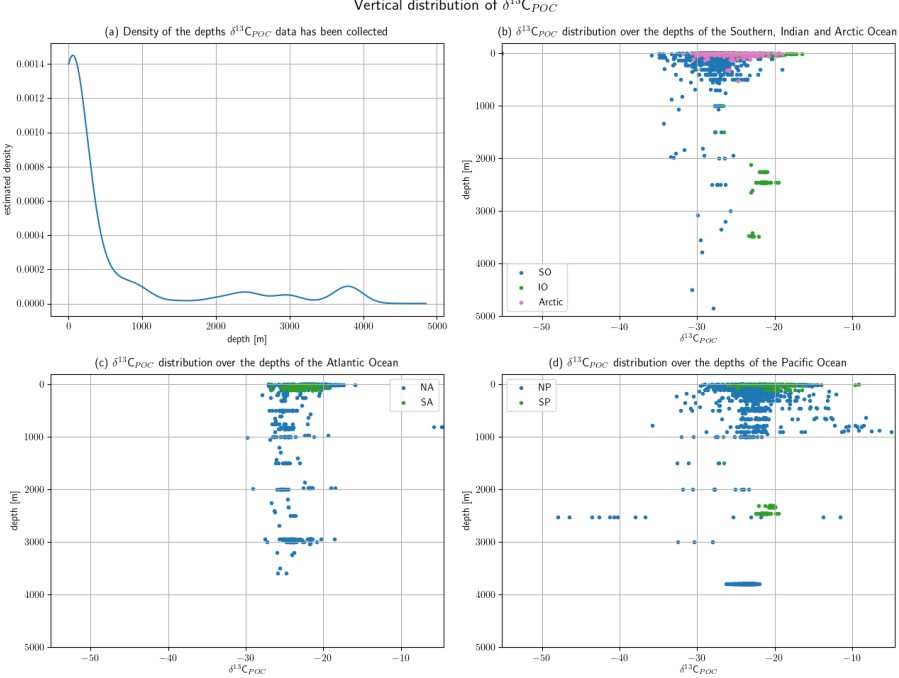

**Figure 5.** The vertical distribution of available $\delta^{13}\text{C}_{POC}$ samples is shown (a) as the approximated density of the measurement depths and (b - d) as measured $\delta^{13}\text{C}_{POC}$ values relative to their respective measurement depth. (a) provides on the y-axis the estimated density of the depth values and on the x-axis the depth in m. The estimation was realized by a Gaussian KDE. (b) resolves the measurements of the Southern, Indian and Arctic Ocean, (c) the North and South Atlantic and (d) the North and South Pacific. The last three panels show on the y-axis the depth in m and on the x-axis the measured $\delta^{13}\text{C}_{POC}$ value. Different colors are used to mark different ocean basins.

Atlantic Ocean provides best data coverage. Following, the Southern and Indian oceans contain the next best coverage with the
northern Pacific having the sparsest.

Highest $\delta^{13}\text{C}_{POC}$ values are evident in low latitude regions. In the Atlantic Ocean highest values were measured between 0-30° N and 30-60° W as well as close to the western coast of France reaching up to $\geq -17\permil$. The Indian Ocean shows generally high values of $\approx -20\permil$. In the Pacific Ocean highest values are close to the Peruvian coast and Papua New Guinea. We also find high values in the Bering Strait and and the northern edge of the Southern Ocean around 65° E.

Lowest $\delta^{13}\text{C}_{POC}$ values are mostly found in the Southern Ocean. Nearly all measured grid cells here belong to $\delta^{13}\text{C}_{POC}$ values lower than around $-28\permil$. The Arctic Ocean shows low values as well, for instance in the Kara Sea. Lowest values in the Pacific Ocean occur in the Southern Ocean at high latitudes.

## 5.3  Meridional trend of $\delta^{13}\text{C}_{POC}$ values

We show the north-south trend of $\delta^{13}\text{C}_{POC}$ over the Atlantic Ocean based on the time-independent UVic grid and restricted to
the uppermost 130 m, which resemble the euphotic zone in the UVic model. We chose this section due to its best data coverage.

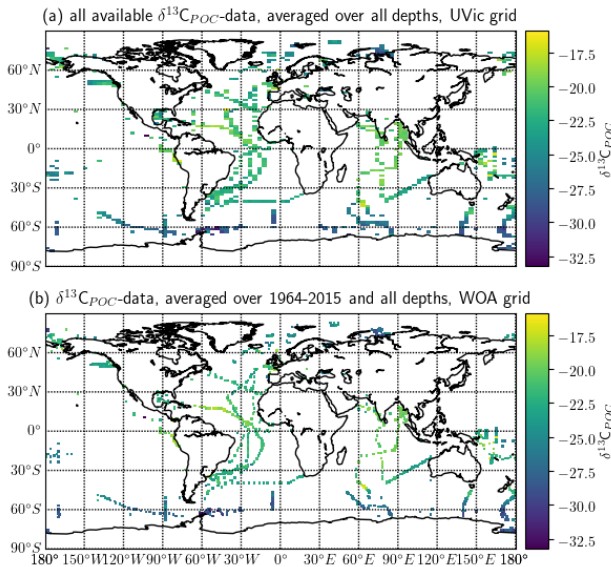

**Figure 6.** Global distribution of the $\delta^{13}C_{POC}$ data is visualized based on (a) the UVic grid and (b) WOA grid. The data used for (a) are independent of time and include all available measurements with latitude and longitude information. The data shown in (b) include only data with complete temporal metadata and are averaged over the years 1964 - 2015. Both data are averaged over all measurements including data with missing depth information. Each colored square refers to a grid cell with available $\delta^{13}C_{POC}$ measurements. The colors indicate the $\delta^{13}C_{POC}$ value in the respective grid cell.

A biome mask according to Fay and McKinley (2014) was applied to the gridded data, thereby defining latitudinal zones in the entire Atlantic Ocean. Distributions of $\delta^{13}C_{POC}$ within the biomes are shown in Figure 7 (see also Table A3 in the Appendix).

The biomes derived by Fay and McKinley (2014) are areas with consistent biological and ecological properties. The chosen biomes cover the Atlantic Ocean and extend to the Arctic Sea and parts of the Southern Ocean. The biomes are numbered 9 to 17, excluding 14. The biomes 15 to 17 are representing parts of the Southern Ocean and were restricted to 70° W and 20° E. Their locations are shown in Figure 7.

Observations by the biomes are consistent with the ones from Figure 6. The two biomes showing the lowest $\delta^{13}C_{POC}$ values from -28 to -29 ‰ are those two located farthest south. The biome located farthest north contains the next lowest values of about -24 ‰. The biomes with more positive $\delta^{13}C_{POC}$ are in the lower latitudes and show similarly higher values from -23 to $-21$‰.

## 6 Temporal distribution of the data set

The full $\delta^{13}C_{POC}$ data cover a time period of around 50 years over 1964-2015 and all twelve months. The number of samples measured during individual decades varies considerably with most measurements in the 1990s. Coverage within the months is quite comparable, only winter months on both hemispheres exhibit less data.

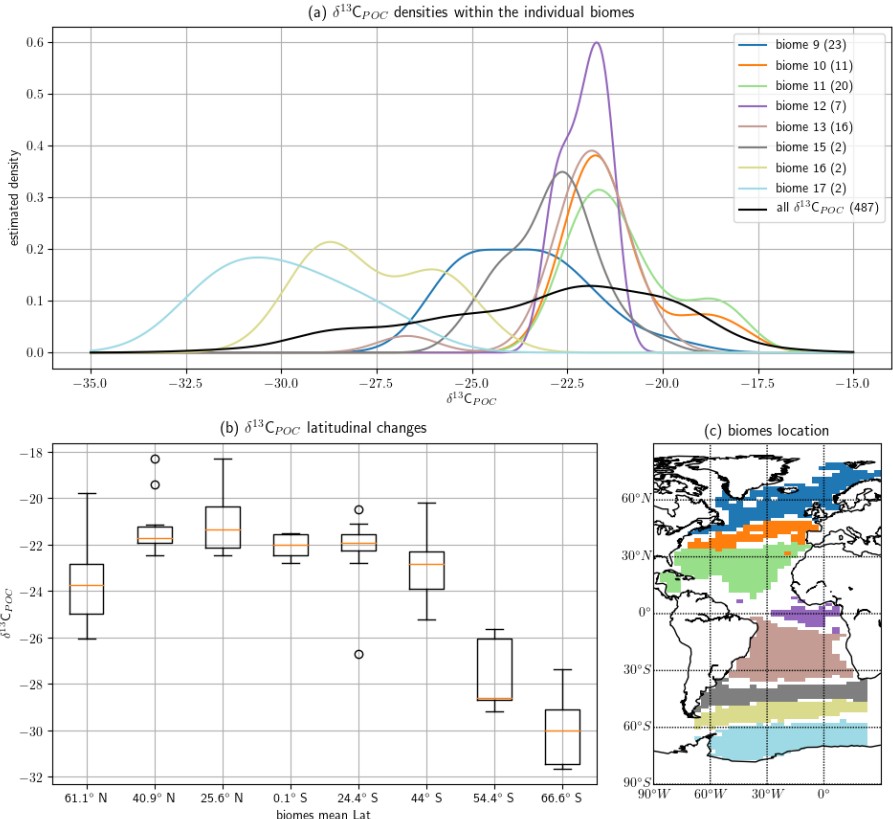

**Figure 7.** North-south trend of sampled $\delta^{13}C_{POC}$ values is visualized by a cross section over the Atlantic ocean. Biomes (Fay and McKinley, 2014) define the latitudinal bands of the interpolated data set. (a) presents for each biome a Gaussian KDE approximating the density of the contained $\delta^{13}C_{POC}$ data. Different colors mark the individual biomes and a black line shows the general global $\delta^{13}C_{POC}$ distribution. The number in brackets in each KDE label counts the number of $\delta^{13}C_{POC}$ measurements used for the respective graph. (b) shows in a box plot the steep decline of $\delta^{13}C_{POC}$ values from the tropical biomes towards the higher latitudes. The x-axis provides the mean latitudes of the biomes introduced in (a). The y-axis measures the $\delta^{13}C_{POC}$ value. (c) shows the biomes locations. Each biome is drawn in the color of its corresponding density estimate in (a) above. The biome numbers increase from the north to the south.

The distribution of $\delta^{13}C_{POC}$ samples over the years is resolved in Table 4 and visually approximated by Gaussian KDE in Figure 8. The 1990s show best data coverage. More than half of the data points are associated to a year in this decade, which are visible by a pronounced maximum in the estimated density. Sparsest data are found in the 1960s, where only 74 data points were sampled. All other decades come with between around 300 and 600 $\delta^{13}C_{POC}$ data points. The latest data are mostly from Lorrain, MacKenzie et al. (2019) and Kaiser et al. (2019). The oldest data were taken from the data sets by Tuerena, Degens

et al. (1968) and Eadie and Jeffrey (1973).

**Table 4.** Data coverage within the available decades: the first column lists the available decades, the second column the number of sampled $\delta^{13}C_{POC}$ data points within this time frame.

| decade | $\delta^{13}C_{POC}$ values available |
|--------|--------------------------------------|
| 1960s  | 74   |
| 1970s  | 321  |
| 1980s  | 463  |
| 1990s  | 2403 |
| 2000s  | 614  |
| 2010s  | 589  |

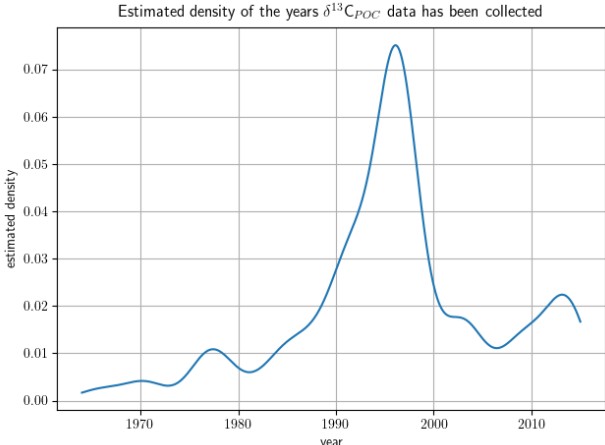

**Figure 8.** The distribution of $\delta^{13}C_{POC}$ data samples over the years approximated by Gaussian KDE. On the y-axis the density is drawn, on the x-axis the sample year. Higher altitude of the graph indicates years with more available data.

## 6.1 Monthly variations

Monthly clustered data of northern and southern hemisphere show monthly variations, but more observations are required to demonstrate robust seasonality within different regions. Since more than 50 % of the available $\delta^{13}C_{POC}$ data originate in the 1990s, we selected data from this decade to exclude changes that might be introduced by longer term trends. Furthermore, we restricted our data to the uppermost 130 m, which resembles in the UVic model the euphotic zone. In Figure 9 we displayed all months with enough data points by a KDE and indicate same months by same colors. We excluded July, November and December on the northern hemisphere from this KDE representation, because these provided three or less data points within them, which resulted in a KDE that overgrew the others by magnitudes and made their visual comparison difficult. The KDEs are supported by comparison of the median values of the individual months in Table 5.

**Table 5.** Monthly median change of $\delta^{13}C_{POC}$. Due to their best data coverage, the analyses were carried out within the 1990s and in the uppermost 130 m.

| hemisphere | Jan | Feb | Mar | Apr | May | Jun | Jul | Aug | Sep | Oct | Nov | Dec |
|---|---|---|---|---|---|---|---|---|---|---|---|---|
| north | | -24.815 | -24.12 | -20. | -24.06 | -24.7 | -21.746 | -23.67 | -22.83 | -21.4 | -23.5455 | -23.368 |
| south | -26.45 | -26.41 | -23.34 | -28.2 | | | | | -28.65 | -27.95 | -27.9 | -26.08 |

The monthly resolved variations of $\delta^{13}C_{POC}$ do not reveal any significant seasonal pattern (Figure 9, see also Table A4 in the Appendix). In general we find highest $\delta^{13}C_{POC}$ values in the northern hemisphere, with median $\delta^{13}C_{POC} = -20.4‰$ in April and $\delta^{13}C_{POC} = -21.5‰$ in October, which are typical months with enhanced primary production (northern hemisphere spring and autumn blooms). Similarly high median $\delta^{13}C_{POC}$ values cannot be ascertained for any month with data of the southern hemisphere, where values of $\delta^{13}C_{POC}$ above $-20‰$ have rarely been observed at any time of the year. In fact, there is an

overall tendency towards low $\delta^{13}C_{POC}$ values for the southern hemisphere, which becomes well expressed during the months April and September, with medians of $\delta^{13}C_{POC} = -28.1‰$ and $\delta^{13}C_{POC} = -28.5‰$ respectively. However, interpretations of this north-south trend should be treated with caution, because the apparent tendency is likely conditioned by some imbalance in the number of high-latitudinal data points. Compared to the number of data points from the Southern Ocean, samples from the Arctic Ocean are considerably underrepresented (see also Figure 10). Furthermore, the discrimination between data of the

northern and southern hemisphere is crude and we encourage to use our data collection for more advanced analyses of seasonal, monthly-based, changes in the $\delta^{13}C_{POC}$ signal.

## 6.2   Decadal variations

The decadal UVic grid NetCDF files are basis for showing long term changes in the $\delta^{13}C_{POC}$ data. An overview of where the data within the individual decades were sampled is given in Figure 10. This shows that sparsest coverage was obtained in

the 1960s, closely located to central American continent. Most data in the Indian Ocean were sampled in the 1970s. A cruise across the southern part of the Atlantic Ocean up to 30° N and some samples close to Iceland were also measured in this decade. 1980s are similarly sparse in spatial coverage as the 1960s. Measurements of the 1980s were taken at locations in the Southern Ocean and in the Arctic and the Atlantic close to the equator. 1990s are best covering most ocean basins. Most Southern Ocean data were sampled within them. The 2000s provide a good coverage of the Arctic Ocean. Finally, the 2010s

data were mostly sampled in the southern hemisphere in the open Pacific and Atlantic. Some smaller Eurasian continental sea data were also part of the 2010s samples.

We show the changes in $\delta^{13}C_{POC}$ values over the available decades by density estimates in Figure 11 (see also Table A5 in the Appendix) and by their median in Figure 12. The first visualizes the sparse coverage of the Southern Ocean outside of the 1990s, which is why it is not part of any further discussion here. The Southern Ocean is defined as the ocean area south of 45°

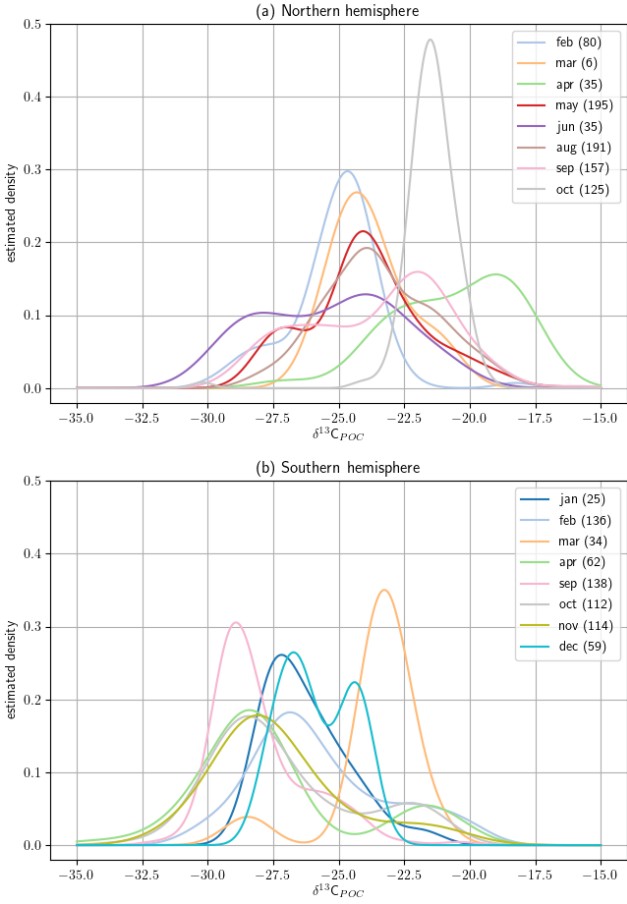

**Figure 9.** Monthly variations are split up by hemisphere in the northern in (a) and southern in (b). Due to their best data coverage, the analyses are carried out within the 1990s and in the uppermost 130 m. The $\delta^{13}C_{POC}$ is split up by sample month and for every month with enough available data points (here more than three) a Gaussian KDE approximate their density. The used data points are given in each KDE label. For each hemisphere the densities are drawn all together, each month indicated by an individual color.

S. All presented analyses were restricted to the euphotic zone, i.e. the uppermost 130 m resembling the two first layers of the UVic grid.

A clear decrease in $\delta^{13}C_{POC}$ densities in Figure 11 can be identified for the global ocean outside of the Southern Ocean. All, but the 1980s show one clear maximum in their approximated densities. The 1980s show a second expressed density maximum at lower values. The main maximum shift from the 1960s at $\delta^{13}C_{POC} \approx -19.9‰$ to the 2010s at $\delta^{13}C_{POC} \approx -23‰$. This
decrease is also clearly visible in the comparison of the decadal medians Figure 12. The Southern Ocean provides far worse data coverage. Only the 1980s and 1990s include enough data to construct a comparable KDE. Due to this very little available data, all of this results must be seen with highest caution.

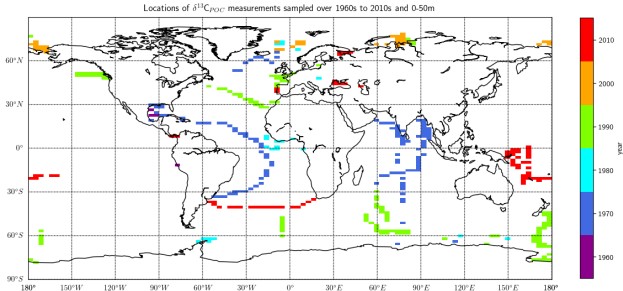

**Figure 10.** Grid locations of the $\delta^{13}C_{POC}$ data, colored by sampling decades. Only data of the uppermost layer are considered in this plot. The different colors indicate the different sample decades and were plotted increasing in time above each other.

## 7    Conclusions

The aim of this work was to construct the largest publicly accessible $\delta^{13}C_{POC}$ data set. The starting point of our collection and analyses was the readily available data collection of Goericke (1994), which comprised 467 data points. Our primary objective was to elaborate this set of data by adding useful meta-information from the original publications and by introducing additional $\delta^{13}C_{POC}$ measurements, as recorded in the world ocean data base PANGAEA and made available by Robyn Tuerena and Anne Lorrain. This way we could expand the data collection substantially, from the original 467 to 4732 data points. This new $\delta^{13}C_{POC}$ data set provides the best coverage to date that will be a useful tool to help constrain many marine carbon cycling processes and pathways from ocean-atmosphere exchange to marine ecosystems, as well as to better understand observations and validate models. To ensure a dynamic growth of our data collection the corresponding author will provide annual updates of the data set. Furthermore, he may be contacted by any interested researcher, who would like to add their data to this collection.

The data are provided in a csv structure and interpolated onto two different global grids as NetCDF format. The csv file contains the $\delta^{13}C_{POC}$, their anomalies to their mean and all available meta information. The interpolations are provided on a coarse $1.8° \times 3.6°$ grid of a $\delta^{13}C_{POC}$ simulating model and a finer $1° \times 1°$ grid by the World Ocean Atlas. We provided a detailed description of our data collection procedure, all added meta information and their coverage as well of the interpolation procedure carried out. We took highest care to make all data coherent, comparable and back trackable and all adjustments transparent. Assumptions, changes and deletions of the used data sets are described in detail.

We described the general spatial and temporal trends of the sampled $\delta^{13}C_{POC}$ data by the raw data file. Distributions were always approximated by Gaussian kernel density estimators. The data range from 1964 - 2015 with far best coverage in the 1990s. Sample locations reach down to a depth of nearly 5000 m and best covers the uppermost 10 m, especially the Atlantic and Indian Ocean. We were able to show our $\delta^{13}C_{POC}$ data values are mostly located between $\delta^{13}C_{POC} = -15$‰ and $\delta^{13}C_{POC} = -35$‰ with two maxima at around $\delta^{13}C_{POC} = -27$‰ and $\delta^{13}C_{POC} = -23$‰, the latter one being the more pronounced. A comparison of the main sample methods showed consistent results when compared with regions. $\delta^{13}C_{POC}$ data separated by months indicate counteracting seasonal trends on both hemispheres, but more data is required to demonstrate robust seasonality.

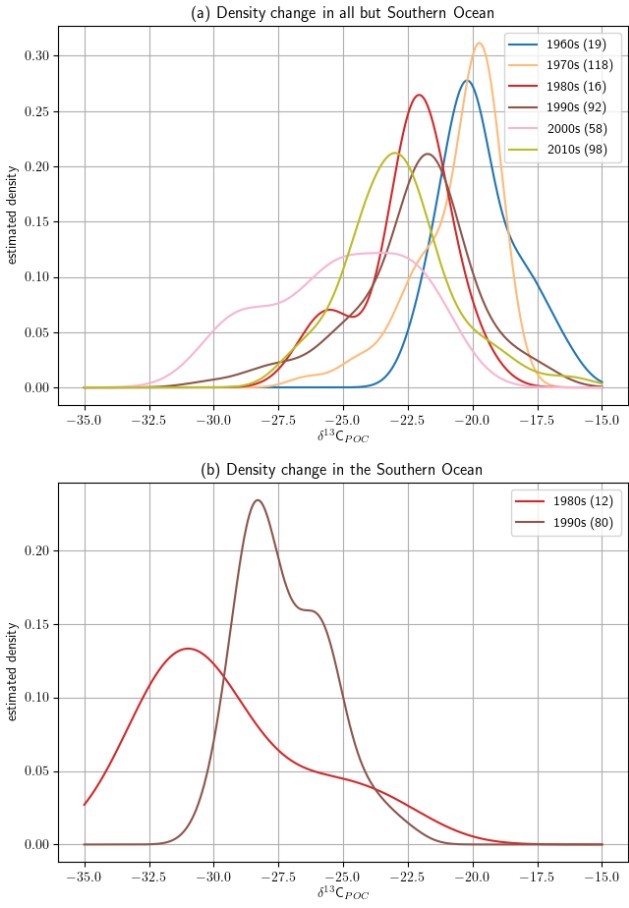

**Figure 11.** The decadal shift of $\delta^{13}C_{POC}$ values for all, but the Southern Ocean (a) and only the Southern Ocean (b) shown by estimated densities of $\delta^{13}C_{POC}$ values. The differently colored graphs refer to the individual decades. Southern Ocean data are sparsely covered and does not provide enough data for a reasonable comparison.

The interpolated data provide insights in geographical behavior of the sampled $\delta^{13}C_{POC}$ data. We showed a good general coverage of all global oceans by $\delta^{13}C_{POC}$, but observed a lack of data in PANGAEA that cover northern Pacific regions. Since the Atlantic Ocean provides the best coverage, corresponding data were used for a north-south trend analysis, where we observed that lowest values ($< \approx -28\text{‰}$) can be found in the Southern Ocean whereas highest ($> \approx -22\text{‰}$) are restricted to low latitudinal regions. This might also have influenced the observed lower $\delta^{13}C_{POC}$ values on the southern hemisphere compared to the northern, due to the relatively good coverage of the Souther Ocean. Finally, we showed the sample locations and value development of $\delta^{13}C_{POC}$ over the observed decades. Since the Southern Ocean data were mainly sampled in the 1990s, a significant multi-decadal trend could not be detected there. In all other oceans our $\delta^{13}C_{POC}$ data show a decrease by about $3\text{‰}$ over the observed timeframe, which is about the double rate of the known Suess effect (Keeling, 1979) on aqueous

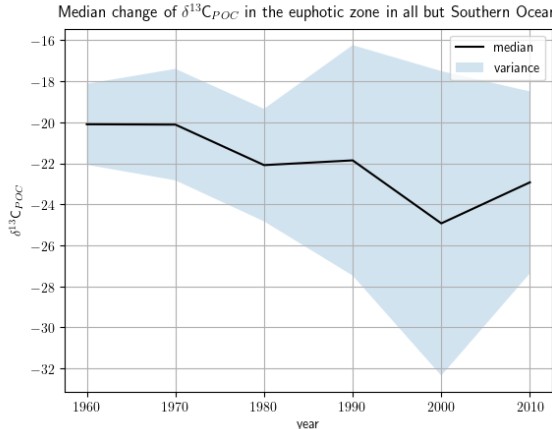

**Figure 12.** The decadal shift of $\delta^{13}C_{POC}$ values in the uppermost 130 m for all, but the Southern Ocean: $\delta^{13}C_{POC}$ decadal median against the decades. The shaded area around the graph marks the variance of the respective decade in each direction.

**Table A1.** Statistical properties of the KDE derived for Figure 1 evaluated on an equidistant grid over $[-55.15, -4.5]$ with 1001 grid points: the first column indicates the respective KDE, the two following list its modes, the fourth the median and the fifth the 95 % confidence interval of the respective KDE. All values are given in ‰.

| $\delta^{13}C_{POC}$ KDE | dominant mode | second mode | median | 95 % confidence interval |
|---|---|---|---|---|
| Figure 1 | $-23.6$ | $-26.9$ | $-23.8$ | $[-30.9, -17.0]$ |

$\delta^{13}CO_2$ (Young et al., 2013). This corroborates an increase in phytoplankton carbon fractionation that may be associated with a change in phytoplankton communities as previously suggested (Lorrain et al., 2020; Young et al., 2013). The data set shows promise to better understand, constrain and predict carbon cycling as it provides a validation tool for mechanistic models and supports separation of non-spatial components in $\delta^{13}C_{POC}$ variations.

*Data availability.* The described $\delta^{13}C_{POC}$ data by Verwega et al. (2021) are available at https://doi.org/10.1594/PANGAEA.929931

## Appendix A: Statistical properties of $\delta^{13}C_{POC}$ kernel density estimates

In Table A1, Table A2, Table A3, Table A4 and Table A5 we present the modes, medians and confidence limits of the KDEs derived in Figure 1, Figure 3, Figure 7, Figure 9 and Figure 11, respectively.

**Table A2.** Statistical properties of the KDEs derived for Figure 3 evaluated on an equidistant grid over $[-35, -15]$ with 1001 grid points: the first column indicates the respective KDE, the two following list its modes, the fourth the median and the fifth the 95 % confidence interval of the respective KDE. All values are given in ‰.

| $\delta^{13}C_{POC}$ KDE | dominant mode | second mode | median | 95 % confidence interval |
|---|---|---|---|---|
| Figure 3a, full | $-21.8$ | $-24.3$ | $-24.3$ | $[-26.8, -18.3]$ |
| Figure 3a, bottle | $-25.1$ | $-$ | $-24.8$ | $[-26.9, -22.0]$ |
| Figure 3a, intake | $-21.6$ | $-$ | $-20.7$ | $[-24.0, -17.4]$ |
| Figure 3a, net | $-21.6$ | $-27.4$ | $-21.7$ | $[-26.4, -19.5]$ |
| Figure 3a, trap | $-24.3$ | $-21.6$ | $-24.1$ | $[-27.2, -20.0]$ |
| Figure 3b, full | $-21.6$ | $-$ | $-21.3$ | $[-24.2, -18.0]$ |
| Figure 3b, intake | $-19.5$ | $-21.1$ | $-20.3$ | $[-24.8, -17.2]$ |
| Figure 3b, net | $-21.5$ | $-$ | $-21.6$ | $[-23.9, -19.4]$ |
| Figure 3c, full | $-21.6$ | $-$ | $-21.6$ | $[-26.4, -17.6]$ |
| Figure 3c, bottle | $-24.7$ | $-$ | $-24.9$ | $[-29.8, -21.1]$ |
| Figure 3c, net | $-21.6$ | $-$ | $-21.6$ | $[-24.0, -19.5]$ |
| Figure 3c, trap | $-21.8$ | $-18.8$ | $-21.7$ | $[-22.8, -18.5]$ |
| Figure 3d, full | $-24.9$ | $-$ | $-24.6$ | $[-27.1, -21.9]$ |
| Figure 3d, bottle | $-25.2$ | $-$ | $-24.8$ | $[-26.5, -22.1]$ |
| Figure 3d, net | $-22.8$ | $-26.9$ | $-24.2$ | $[-29.4, -19.7]$ |
| Figure 3d, trap | $-24.3$ | $-26.6$ | $-24.5$ | $[-27.2, -22.9]$ |

**Table A3.** Statistical properties of the KDEs derived for Figure 7 evaluated on an equidistant grid over $[-35, -15]$ with 1001 grid points: the first column indicates the respective KDE, the two following list its modes, the fourth the median and the fifth the 95 % confidence interval of the respective KDE. All values are given in ‰.

| $\delta^{13}C_{POC}$ KDE | dominant mode | second mode | median | 95 % confidence interval |
|---|---|---|---|---|
| Figure 7a, all | $-21.8$ | – | $-22.8$ | $[-29.9, -18.1]$ |
| Figure 7a, biome 9 | $-24.0$ | – | $-23.8$ | $[-27.5, -18.5]$ |
| Figure 7a, biome 10 | $-21.7$ | – | $-21.5$ | $[-25.0, -17.9]$ |
| Figure 7a, biome 11 | $-21.6$ | $-21.1$ | $-21.3$ | $[-24.4, -17.7]$ |
| Figure 7a, biome 12 | $-21.7$ | – | $-21.9$ | $[-23.2, -20.8]$ |
| Figure 7a, biome 13 | $-21.9$ | $-24.9$ | $-22.0$ | $[-24.4, -20.4]$ |
| Figure 7a, biome 15 | $-22.7$ | – | $-22.8$ | $[-26.5, -19.2]$ |
| Figure 7a, biome 16 | $-28.7$ | $-26.0$ | $-27.7$ | $[-30.7, -24.1]$ |
| Figure 7a, biome 17 | $-27.8$ | – | $-28.5$ | $[-32.7, -24.9]$ |

**Table A4.** Statistical properties of the KDEs derived for Figure 9 evaluated on an equidistant grid over $[-35, -15]$ with 1001 grid points: the first column indicates the respective KDE, the two following list its modes, the fourth the median and the fifth the 95 % confidence interval of the respective KDE. All values are given in ‰.

| $\delta^{13}C_{POC}$ KDE | dominant mode | second mode | median | 95 % confidence interval |
|---|---|---|---|---|
| Figure 9a, feb | $-24.7$ | – | $-25.0$ | $[-29.2, -22.3]$ |
| Figure 9a, mar | $-24.3$ | – | $-24.0$ | $[-26.6, -20.3]$ |
| Figure 9a, apr | $-19.0$ | – | $-20.4$ | $[-26.2, -16.5]$ |
| Figure 9a, may | $-24.1$ | $-27.0$ | $-24.0$ | $[-28.3, -19.0]$ |
| Figure 9a, jun | $-24.0$ | $-27.9$ | $-25.2$ | $[-30.5, -20.1]$ |
| Figure 9a, aug | $-23.9$ | – | $-23.6$ | $[-27.7, -18.9]$ |
| Figure 9a, sep | $-22.0$ | $-26.0$ | $-23.1$ | $[-28.9, -18.7]$ |
| Figure 9a, oct | $-21.5$ | – | $-21.5$ | $[-23.4, -19.7]$ |
| Figure 9b, jan | $-27.2$ | – | $-26.5$ | $[-28.9, -22.1]$ |
| Figure 9b, feb | $-29.9$ | – | $-26.2$ | $[-30.3, -19.8]$ |
| Figure 9b, mar | $-23.3$ | $-28.5$ | $-23.3$ | $[-29.0, -21.0]$ |
| Figure 9b, apr | $-28.4$ | $-21.6$ | $-28.1$ | $[-32.6, -19.9]$ |
| Figure 9b, sep | $-28.9$ | $-20.4$ | $-28.5$ | $[-30.8, -23.6]$ |
| Figure 9b, oct | $-28.5$ | $-22.3$ | $-27.7$ | $[-31.7, -20.8]$ |
| Figure 9b, nov | $-28.1$ | – | $-27.7$ | $[-31.8, -20.1]$ |
| Figure 9b, dec | $-26.7$ | $-24.4$ | $-26.0$ | $[-28.3, -23.3]$ |

**Table A5.** Statistical properties of the KDEs derived for Figure 11 evaluated on an equidistant grid over $[-35, -15]$ with 1001 grid points: the first column indicates the respective KDE, the two following list its modes, the fourth the median and the fifth the 95 % confidence interval of the respective KDE. All values are given in ‰.

| $\delta^{13}C_{POC}$ KDE | dominant mode | second mode | median | 95 % confidence interval |
|---|---|---|---|---|
| Figure 11a, 1960s | $-20.0$ | – | $-19.9$ | $[-26.8, -16.5]$ |
| Figure 11a, 1970s | $-19.8$ | – | $-20.4$ | $[-25.0, -18.0]$ |
| Figure 11a, 1980s | $-21.7$ | $-25.3$ | $-22.1$ | $[-26.9, -18.5]$ |
| Figure 11a, 1990s | $-21.8$ | $27.3$ | $-22.1$ | $[-27.6, -18.2]$ |
| Figure 11a, 2000s | $-22.4$ | – | $-23.2$ | $[-30.4, -19.2]$ |
| Figure 11a, 2010s | $-23.1$ | – | $-23.3$ | $[-27.4, -17.6]$ |
| Figure 11b, 1960s | $-27.5$ | $-30.3$ | $-27.7$ | $[-31.4, -25.2]$ |
| Figure 11b, 1980s | $-31.0$ | – | $-29.8$ | $[-34.3, -15.0]$ |

*Author contributions.* M. - Th. Verwega collected and merged the data, performed the analyses and set up the manuscript. C. J. Somes
initiated and supported the data collection, conducted the grid interpolations, guided analyses of the data and structured and proofread the manuscript. M. Schartau supported the data collection, guided their analyses and proofread the manuscript. R. E. Tuerena provided additional data and ideas for its analyses and proofread the manuscript. A. Lorrain provided additional data and proofread the manuscript. A. Oschlies guided the analysis of the data and proofread the maniscript. Th. Slawig guided the elaboration of the manuscript, structured and proofread it.

*Competing interests.* The authors declare that they have no conflict of interest.

*Acknowledgements.* The first author is funded through the Helmholtz School for Marine Data Science (MarDATA), Grant No. HIDSS-0005. C. Somes is funded by the Deutsche Forschungsgemeinschaft (DFG, project no. 445549720)

We like to thank Tronje Kemena for providing the basic global biomes masks, used for analyzing the interpolated data sets on the coarse grid.

We thank the referees and editors for their constructive feedback regarding the initial version of the manuscript.

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
