# Peer review of "Description of a global marine particulate organic carbon-13 isotope data set"

_Earth System Science Data, 2021_

## Referee Comment (RC1)

**Referee Comments, 9 juli 2021**

**'Description of a global marine particulate organic carbon-13 isotope data set'**

Verwega et al., 2021 (ESSD)

(https://doi.org/10.5194/essd-2021-159)

**General Comments**

Dear Maria-Theresia Verwega and colleagues,

Thank you for your thorough and well-written manuscript. You have provided both the observational and modelling communities with a useful and unique d13C_POC compilation that can be applied to a wide range of research questions and technical (model) evaluations. The details on the temporal and spatial distribution of the data are clearly presented and in general supported by informative figures. I first have some general comments to make:

1. Even though your Introduction reads very well, I think it contains relatively few references. Consider adding some more references. A general one like Zeebe & Wolf-Gladrow (2001) for example?

2. You vary a bit in naming the two grids: The 'coarse grid', the 'UVic grid', 'the main dataset'. I think it is clearest if you refer to them as the 1x1 grid and the 1.8x3.6 grid, and only mention UVic/coarse/fine in the general introduction to the data at the beginning of Sect 3. If you want to present the 1.8x3.6 degree dataset as your main dataset, clarify this early on. In any case, choose a uniform naming.

3. Related to comment (2), why do you choose to present the data in the 1x1 and 1.8x3.6 grids? The 1x1 is commonly used and in WOA format so this one I understand well. As a modeler not working with the UVic model, which I expect most of your dataset users will be, I would be interested in using either the raw data or a gridded 1x1 dataset with the time axis preserved (that is, Year and Month info). Splitting the 1x1 dataset up in decades like you did for the UVic grid could then also be helpful for users. Why do you focus on the UVic grid as the main dataset? As you might understand, I expect the 1x1 grid to have more potential to be used by the broader community.

4. For e.g. your presentation in Sect. 5.2, Sect. 5.3 and Sect. 6, unless d13C_POC at the surface is similar to d13C_POC at depth (unlike d13C_DIC), a depth-average might not be meaningful everywhere. I would then assume taking the surface values only is best. Adding a figure on d13C_POC vs depth in your section on vertical distribution of the data would clarify this. In P20, l325 you indeed state that you only took the euphotic zone – why here and not in other places?

5. Is it possible to give an educated guess on the uncertainty of d13C_POC? This may vary per decade / sampling method / cruise, and I can imagine the source data do not give such estimates themselves. But your experience may give the reader an estimate of the uncertainty, which is valuable for any further analysis.

6. Last, I found that the dataset, even though in practical NetCDF format, does not contain enough information and does not follow conventions well enough to be worked with easily. Please see my comments in a separate Section 'The dataset'.

See also my Specific Comments below, as well as a few Technical Comments.
I am happy to recommend your manuscript for publication after you have clarified my comments and hope they are useful for improving the manuscript.

Best regards,

Anne Morée

**Specific Comments**

P1, l8-9: the consistency with observations is a bit inherent as you compile all available observations? Also, barely any comparison is made in the text with older literature/data for consistency.

P2, l32: This is an example of a location where more and more relevant references would be in place (Rocha and Passow, 2014 only is limited to refer to for the reader to understand the role of the biological pump in sequestering C).

P2, l43-49: In this section you describe the fractionation during photosynthesis. I think it is important to mention somewhere in your introduction that there are three reactions where the C isotopes fractionate: calcification, photosynthesis and air-sea gas exchange and include relevant references. And that the relative importance of these processes depends on location: e.g. (Gruber et al., 1999; Morée et al., 2018; Schmittner et al., 2013).

P2, l54: Underline your statement with some references from both the land biosphere and marine realm. For example, the Suess effect is visible in the ocean in d13Cof DIC (Eide et al., 2017).

P3, l57-60: To what extent are changes in the other fractionation pathway, air-sea gas exchange relevant to your study? The temperature dependence of fractionation during air-sea gas exchange (Zhang et al., 1995) suggests that in a warming world fractionation is weaker over the air-sea interface. Also, Young et al. (2013) reconstructed that the fractionation factor during photosynthesis is changing due to rising CO2 concentrations. If the d13C_DIC in the euphotic zone is different due to the Suess effect, your d13C_POC is affected. In an ESSD article the discussion on this is not necessary, but I think it is important to point the reader to such studies that are relevant for the interpretation of your dataset.

P3, l62: Please make the transition from the previous paragraph to this one more fluent.

P3, l65: When it comes to the implementation of the C isotopes in the ocean component of ESMs, some recent advances could be highlighted here as well: e.g. (Liu et al., 2021; Tjiputra et al., 2020).

P3, l69-74: Please add some references as examples of your statements (especially in line 71).

P3, l80: what do you mean with multilateral here?

P3 l87 – p4, l90: This sentence is inconsistent with/incomplete as compared to p3, l76-78: You say here that you included unpublished data Lorrain and Tuerena but earlier just Lorrain. And you don't cite Tuerena et al. (2019) here which you did before.

P4, l107-110: Repetitive; You mention twice in this section that this is the Tuerena data set.

P5, l129: Wherever … one type ; this sentence is difficult to read, please rephrase. It is also not clear to me how you choose between the similar measurements, and what made them similar (the value?). Are the two following sentences the only two times you have done this? In that case the sentence could end with a ':'.

P5, l141: Why provide it on the UVic grid except for that they have d13C_POC as output? Wouldn't it be more logical to present it on the WOA 1x1 degree grid and provide for example a Ferret/CDO/NCO guide on how to change it to a different grid format (or do you loose too much information in this case, regridding twice)? Also, based on this paragraph I would expect 3 files in the dataset – I think this is a good place to tell the reader how many files you have (and therefore also that you split them up in decades), what their purpose is, what they contain, etc.

P6, Table 1: Do I read it correctly that dataset Degens et al (1968) till Wada et al. (1987) all had their longitude changed? This is not entirely clear because it is empty behind all rows except for Degens'. If the Table should be read such that multiple rows have undergone the same change, I would suggest using curly brackets and centering the change description behind those.

P7, l 156: Why did you leave out the sample day? This does not really confirm your statement that you have taken as many details as possible (p4, l114).

P7, Sect 3.2: Could to the interpolated data description be added what the dimensions are of each dataset (lon, lat, depth, time?) and the size of these dimensions?

P8, Table 3: coverage of Depth, how can that be out of 4754 datapoints? The maximum is 4732, right?

P8, l174: Why did you exclude some data here (because they e.g. lack depth information?)? Do you mean you used the data with full spatial-temporal coverage (thus datapoints that have lon, lat, depth and year and month? – please specify). I assume that if you for example not have the spatio-temporal full metadata information, you could also not add them to the UVic grid dataset (but P9, l186 suggest you did – how?)?

P12, l233-243: 'Overall, after accounting for spatial sampling bias by comparing with regions, the different methods are generally consistent with each other (Figure 3).' I do not conclude that as easily from Fig. 3, how did you account for spatial sampling bias by comparing with regions? (clarify this and check also p20, l355). I think you mean with*in* regions as stated in the abstract, but then still I do not see where you have made a comparison between the sampling methods within several different regions (Fig. 3 is just the Atlantic). Also, why not use the biome regions here for consistency with your other regionally presented data?
And in the paragraph that follows, if e.g. net data make up most of the data of the full Atlantic, then it is no surprise that the full data KDE is similar to the net data? In order to discuss Fig. 3, does one not need a plot of what fraction of the data is coming from what sampling method? E.g. as plot of number of data versus time with contributions from the different sampling methods or something similar? Do these differences between the methods maybe give us an impression of the uncertainty of the d13C_POC values (see also my general comment 5).

P12, l240: Besides discussion the variance, I think it would be interesting to provide the reader with information on differences in the mean/median between the methods (e.g. in a region/the regions in Fig. 3 but also globally compiled).

P12, Sect 5.1.: Here you discuss mostly the density/number of data at a certain depth, take over the global ocean. It would be interesting to hear how this varies with region (so e.g. fewer very deep data in remote locations, etc.) and a plot of d13C_POC versus depth (global mean or region, whatever is more meaningful/informative) as a Fig. 4b for example.

P13, Sect 5.2: How can the coarse resolution dataset be independent of time – clarify how you merged the time dimension? Are the d13C_POC values of depth-averaged lon,lat data meaningful – the value would depend on how many deep measurements are included (or is deep d13C_POC similar to surface d13C_POC?). Why not just show the locations of the data with a black marker in Fig. 5 in order to show their horizontal distribution?

P14, Table 4: I think these data er more logically represented as a histogram. If you think it is important to show the exact values, this could even be added into the histogram.

P14, Sect 5.3: Did you average over all depths or use surface values? Also, I did not understand the first sentence of the Section (in which figure is this shown?): for which decade is Fig. 6 made? In Fig.6, instead of a mean vs. biome wouldn't a mean versus latitude plot contain more information as it then is not discretized into these biome intervals? Or do you need to define zones because of low spatial data coverage?

P16, l291: Maybe help the reader by stating what that means for seasonal availability for each hemisphere?

P17, l 302: define 'enough' in 'enough datapoints' (also in the caption of Fig. 8).

P19, Fig. 8: In the caption you write that b and d are means, but in the title and text (p17, l303) it says median – what is it? I think connecting the mean/median values with a line is a bit confusing especially in d – why not present the values in a small table?

P20, l352: not specific enough, why not specify which areas?

**Technical corrections**
P1, l2: They have *for example* been used to?
P1, l16: via *its* atmospheric form
P2, l 40:  …- 1) : 1000 and remove the '.' at the end
P2, l55: reference?
P3, l69: improve our understanding *of* marine carbon
P4, l11: *and* KH13.
P4, l114: Table 3 referred to before table 1 and 2?
P5, l140: remover the '.' in front of (Verwega et al., 2021).
P9, l190: refer to Figure 1?
P9, l194: refer to Figure?
P9, l201: 'what indicates very little data points lying' should be 'which indicates that very few data points lie'
P16, l 285-287: change /permil to ‰
P18, l310: 'highest maximum' seems double, reformulate?
P20, l344: change to 'two different global grids', the word resolution here reads not well in this position. Or rephrase.
P20, l345: *relative* to their mean? As *an anoma*ly to their mean?
P23, l383-386: 2003a and 2003b instead of 2003an and 2003av?

**The dataset**
1. Only after opening the dataset it became clear to me that the decadal files have all data of that decade saved together in one file (right? Or is the mean for each location?). The poc13_univ_i1.nc file was also unexpected based on the text, and seems to contain all data merged over time, but not over depth. This relates to my comment on P5, l141 – tell the readers how many files they will find and what they contain.
2. The TANN dimension has no description (ie netcdf attributes) but 'TANN:axis = "T" ', please add more information: The netcdf files state that the files follow the CF-1.6 convention. However, when checking this (https://pumatest.nerc.ac.uk/cgi-bin/cf-checker.pl), this seems not to be true for TANN. There is also TYR in the poc13_year_month_woa_c1.0.nc file with the same issue. The dimension 'record' in the last file also confused me.
3. I think the naming of the x and y axis (which represent lon and lat) is not very intuitive or standard – why not use lon or long and lat?
4. Global&variable attributes: If your files end up being saved on a computer somewhere, based on the file name one would not be able to know what they exactly are, who made them, how they were made, etc. I think the absence of this information in the files themselves is important and should be addressed. Try to describe them in such detail that if none of the authors can be contacted for extra information and help, that the dataset is still usable. E.g., in the global attributes, include references to all

sources (Table 1 and 2), reference to this article, a title, datetime_start, datetime_stop, size, contact, license (CC-BY 4.0?), method/how the dataset was made, description, and any other attributes of potential use. For all attributes and dimensions make sure to give them at least where applicable units, long_name, standard_name, _FillValue, missing_value, description. Also, DEPTH has a strange missing_value.

5. In the poc13_year_month_woa_c1.0.nc file, is dimension 'record' the month and TYR the year? Why are these not in one time dimension? A 5 dimensional variable, which it is now, is difficult to work with (for example, CDO can't do it).

**References**

Eide, M., Olsen, A., Ninnemann, U. S., & Eldevik, T. (2017). A global estimate of the full oceanic 13C Suess effect since the preindustrial. *Global Biogeochemical Cycles*, *31*(3), 492-514. https://doi.org/10.1002/2016GB005472

Gruber, N., Keeling, C. D., Bacastow, R. B., Guenther, P. R., Lueker, T. J., Wahlen, M., Meijer, H. A. J., Mook, W. G., & Stocker, T. F. (1999). Spatiotemporal patterns of carbon-13 in the global surface oceans and the oceanic suess effect. *Global Biogeochemical Cycles*, *13*(2), 307-335. https://doi.org/doi:10.1029/1999GB900019

Liu, B., Six, K. D., & Ilyina, T. (2021). Incorporating the stable carbon isotope 13C in the ocean biogeochemical component of the Max Planck Institute Earth System Model. *Biogeosciences Discuss.*, *2021*, 1-44. https://doi.org/10.5194/bg-2021-32

Morée, A. L., Schwinger, J., & Heinze, C. (2018). Southern Ocean controls of the vertical marine δ13C gradient – a modelling study. *Biogeosciences*, *15*(23), 7205-7223. https://doi.org/10.5194/bg-15-7205-2018

Schmittner, A., Gruber, N., Mix, A. C., Key, R. M., Tagliabue, A., & Westberry, T. K. (2013). Biology and air-sea gas exchange controls on the distribution of carbon isotope ratios (δ13C) in the ocean. *Biogeosciences*, *10*(9), 5793-5816. https://doi.org/10.5194/bg-10-5793-2013

Tjiputra, J. F., Schwinger, J., Bentsen, M., Morée, A. L., Gao, S., Bethke, I., Heinze, C., Goris, N., Gupta, A., He, Y. C., Olivié, D., Seland, Ø., & Schulz, M. (2020). Ocean biogeochemistry in the Norwegian Earth System Model version 2 (NorESM2). *Geosci. Model Dev.*, *13*(5), 2393-2431. https://doi.org/10.5194/gmd-13-2393-2020

Young, J. N., Bruggeman, J., Rickaby, R. E. M., Erez, J., & Conte, M. (2013). Evidence for changes in carbon isotopic fractionation by phytoplankton between 1960 and 2010. *Global Biogeochemical Cycles*, *27*(2), 505-515. https://doi.org/10.1002/gbc.20045

Zeebe, R., & Wolf-Gladrow, D. (2001). *CO2 in Seawater: Equilibrium, Kinetics, Isotopes* (Vol. 65). Elsevier Science B.V.

Zhang, J., Quay, P. D., & Wilbur, D. O. (1995). Carbon isotope fractionation during gas-water exchange and dissolution of CO2. *Geochimica et Cosmochimica Acta*, *59*(1), 107-114. https://doi.org/http://dx.doi.org/10.1016/0016-7037(95)91550-D

---

## Community Comment (CC1)

General comments:

I would like to thank the authors for their thorough manuscript. I applaud them for their effort to compile such an extensive dataset of all available d13C POC measurements across the global ocean over multiple depth layers and monthly and decadal time-scales. Datasets as such are highly valuable and find application in a variety of research and technical fields and are useful to calibrate/validate process-based, mechanistic isotope-enabled models. The dataset and its components are clearly presented and so are major patterns in d13C POC values across space and time. It would be also interesting to see if there are any trends with depth, and how these may vary among ocean areas.

I provide a few – hopefully constructive – general and specific comments below and recommend this manuscript for publication after my comments are addressed or discussed.

Issues that I noticed throughout the manuscript:
-data is plurar, datum is singular. So, please make sure to use plural forms when speaking about data
-be sure to use present/past tenses and active/passive forms consistently
-be consistent with terminology related to spatial distributions: coarse/fine grid/interpolation.

Specific comments:
Abstract
Line 1: I think that the correct terminology here is just 'marine particulate organic carbon stable isotope ratios (d13C POC)', without -13 here, as by definition the isotope ratio is given by the ratio of the heavy (carbon-13) to the light (carbon-12) isotopes.
 Line 13: need commas for statement regarding the Southern Ocean: 'except for the Southern Ocean, which shows a weaker trend, but contains...'.

Introduction
Line 16: Consider changing 'it is regulating' with 'it regulates'
Line 47: there are, not their are
Lines 47-48: I don't particularly agree with this statement as I think that the factors and processes underlying fractionation during photosynthesis are fairly well-understood, and fractionation can be predicted with confidence from [CO2aq], phytoplankton growth rate and community composition (due to different cell sizes and geometries of different taxa). Furthermore, these factors are all governed, to an extent, by temperature, which makes temperature a key predictor for fractionation and d13C POC patterns. Perhaps you should mention that here.
Line 58: concentration of CO2aq is also temperature dependent, and so is the distribution of phytoplankton communities. That is, all factors that exert a direct influence on photosynthetic fractionation are ultimately controlled by temperature, which is a major control on fractionation during photosynthesis by phytoplankton. I would stress this a little more in the Introduction.
Lines 59-61: again, I think we have a fairly good understanding of the processes causing variation in photosynthetic fractionation, but a dataset with extensive spatio-temporal coverage is certainly needed to investigate how trends change across space and time and the mechanisms underlying these changes as well as for calibrations/validations of process-based/mechanistic models with no data component.
Line 65: please add citation Magozzi et al. 2017 Ecosphere here. This study models the C isotope fractionation during photosynthesis as a function of a suite of variables provided by the ocean biogeochemical model NEMO-MEDUSA and predicts spatial and temporal patterns in d13C POC across the global ocean over seasonal to decadal time-scales.

Line 66: calibration AND validation. A major issue associated with isotope-enabled biogeochemistry models for the global ocean is the lack of reliable validation datasets, with sufficient spatio-temporal coverage to allow proper validation (sometimes, datasets are so scarce or so scarcely comparable that it almost makes more sense to 'trust' the mechanistic model, based on fairly well-known and understood processes, rather than calibrating the model to the available data)

Line 80: what does 'multilateral' mean?

Line 83: don't need a capital W for we after the semi-column. Please fix this here and throughout the text, as well as in figure captions.

2 Data acquisition

Line 89: rephrase this as 'the adjustments that we conducted are described in the following sections' or something.

2.2 Adjustments made

Line 122: Isn't this Sackett et al. 1966? With a double t?

3 Content and structure of the data set

Line 140: don't need a full stop before reference.

Line 141: why are data presented as anomalies with respect to the global mean d13C POC value? I think that the authors explained this in lines 149-150 but it is not clear to me what they mean.

3.1 Raw data file

Lines 149-150: what do the authors mean by 'anomalies contain all relevant information...during first steps of model calibration'?

3.2 Interpolated data sets

Please make sure that you use the terms coarse/fine and grid/interpolation consistently throughout the manuscript.

Line 165; '...seven different files, where six files contained an individual decade each' or something.

Lines 165-168: here you should maybe say that: on the coarse grid, data were interpolated independent on time, averaged across depths; interpolation on the fine grid only included data with complete spatio-temporal information, averaged across times and depths.

Lines 168-175: why did you interpolated data onto two different grids? Couldn't you just interpolate on the fine grid and resample if you needed values interpolated on a coarser grid?

Line 176: would be helpful if you could please add a statement to say, in poor words, what the Ferret interpolation does and how that works; essentially a sentence that explains the eqns., briefly.

4 Main data set characteristics

Heading: you always use data set, not dataset. Make sure to be consistent.

4.1 Range and outlier values

Line 199: before and after the two outer modes, respectively? Density declines to 0 at d13C POC values < than the more negative mode, and at values > than the more positive mode. Is that what you mean here, right?

Lines 204-205: it would be helpful to mention where these locations are. Also, please change 'the smallest outlier' with the 'most negative/the lowest outlier' or something, as it is not straightforward what you mean here with smallest outlier

Lines 206-2010: again, I think it would be helpful if you could please add where these coordinates are

**4.2 Sampling method**

Line 220: What does it mean that different sampling method could be attributed to 67% of the data as meta information? That 67% of the data had associated sampling method information?

Line 229: double brackets in reference

Lines 232-233: how did you account for spatial sampling bias? What do you mean here with 'by comparing with regions'? Simply that you compared d13C POC data obtained with different method within each region (Figs. 3b-d)?

Line 233: are they? It looks to me that in some cases different sampling methods provide different d13C POC values, e.g., between 30-60 N, bottle values are lower than values obtained with the other sampling methods

Line 236-237: please rephrase this sentence as it is very complex and it is not clear to me what you mean

Line 239: closely aligned

In general, how do read from Fig. 3 the % of data collected with each method in each area?

Line 240: rephrase with 'the variance... is approx. 3 per mil lower than the variance of all d13C POC values, which is approx. 5 per mil, the highest value observed here', or something.

Line 242: show a pronounced, remove clearly. Also second is repeated twice. Also show a clear individual maximum, remove mostly.

**5 Spatial distribution**

Line 245: please consider rephrasing this, e.g., 'we show the spatial distribution of d13C POC measurements across the global ocean surface and depths'. Data is plural, therefore 'most d13C POC data have been measured', please make sure to be consistent with use of plural for data throughout the manuscript.

**5.1 Vertical distribution of the data set**

Line 250: if 80% of the data have associated depth info, depth is a fairly well-recovered metadatum, isn't it? Does that mean that most datapoints don't have associated T and sampling method info?

Line 254: 'within the first 130 m'

Line 255: remove already

Line 255-256: '200 d13C POC values are available in the depth interval [...]'

Line 257: add respectively at the end of this sentence

In addition to how many data points there are for each depth layer etc, it would be interesting to see a plot showing trends in d13C POC values with depth, similar to Figs. 6, 8 and 10 for biomes, months and years...

**5.2 Horizontal distribution of the data set**

Out of curiosity: does the dataset include any d13C POC data for the Mediterranean Sea?

Line 260: I tend to prefer the use of 'grid' over 'interpolation', as the interpolation is essentially the spatial/horizontal distribution of values which can be done over a grid, isn't it?

Line 261: to set context for next sentence, mention here that data are averaged across all depths'.

Line 263, don't need a full stop after Figure 5, but comma.
Line 264: not sure what 'also, data locations of … occur' means.
Line 267: lowest?
Line 269: substitute 'with' with 'of' XX per mil

5.3 Meridional trend of d13C POC values
Line 276: again, make sure to be consistent with use of coarse/fine grid/interpolation.
Line 279-280: description of colors and lines should be given in figure caption, not in the main text.
Line 283: what does 'but 14' mean? Biomes were numbered from 9 to 17, where 15-17 had to be cut to the given lateral range. Also, consider using longitudinal rather than lateral here.
Line 284: I think there is a mistake here, location of biomes in the Atlantic is shown in Fig. 6c, not Fig. 10.
Lines 285-287: please insert per mil symbol. Also consider changing 'the final biomes' with 'the biomes with more positive d13C POC values' or something.

6 Temporal distribution of the dataset
Line 293: Fig. 7 here, not 5.
Line 295-296: latest data are (plural), and again in the following line

6.1 Seasonal trends
You don't describe Figs. 8b,d but I think they're informative as they show the seasonal trend in d13C POC values.
Also, please make sure the distinction between winter/summer is clear for the /S hemisphere in this paragraph.

6.2 Multi-decadal trends
Line 328-329: remove 'both'. Also second is repeated twice.
Lines 329: main maximum shift or the shift in main maxima. Remove 'with every decade lower'.
Lines 334-339: I would remove these lines, if you really want to keep Figs. 10c,d in. Or you could also remove the figures and just say that there are not enough data in the SO to investigate multi-decadal trends.

7 Conclusions
It would be nice to have a paragraph in Conclusions with examples of research and technical questions that could be tackled/answered with datasets as such. These applications should link back to themes presented in the Introduction.
Additionally – and this my reflect my own research interests – I think that the authors should stress the importance of their dataset for calibration/validation of process-based, mechanistic models.  A major issue related with the application of these models in ecology, for instance, has been the lack of suitable calibration/validation datasets, resulting in large and mostly unknown uncertainties (models trusted more than data, as they're based on fairly well-understood mechanisms whereas data are scarce and often incomparable). Datasets like this one provide a validation tool for mechanistic model, and potential for the development of data-based models of the spatio-temporal distributions of stable isotopes in marine ecosystems. An approach that has been successfully used to develop data-based isoscapes is the INLA method (St John Glew et al. 2019 MEE, St John Glew et al. 2020 ESSOAr), which allows separating spatial from non-spatial components of isotope variance when predicting spatial isotope patterns. This dataset could be suitable for such approach as it contains some meta information (e.g., sampling method, depth,

month, decade, etc.) which can be included as factor to estimate non-spatial variance when predicting spatial variation from environmental covariate sets.

Tables

Table 1 & others: don't need a capital letter after semi-column. The second column lists in which ..., without comma after lists. The third and fourth columns, plural; also unnecessary comma between show and from; also show from what values to which values (or something).

Table4: change inspired with based on, or something. Not sure what the sentence starting with 'below 50 m...' means, why only below 50 m? In the last sentence, depth range not depths range.

Figures

Figure 1: in caption don't need capital V for values after semi-column. Please fix this here and in other figs' captions and throughout text.

Figure 6b: can you plot mean lat for each biome on the x-axis, rather than biome number? Or at least an arrow N to S below the x-axis? You need to make this fig as much self-explanatory as possible.
Also, b can you plot some confidence intervals around means in panel b, given by variance of KDE? Alternatively, you could plot boxplots of for each KDE, without black line connecting means.

Figure 8b,d: Similar comments to Fig. 6b.

Figure 9: in caption, 'grid-locations of d13C POC data, colored by sampling decades' or something. Find a clear way to say that the grids of sample locations are shown here, colored by the decades in which the samples were collected. Aren't there any grids with multiple samples collected in different decades?

Figure 10b,d: Similar comments to Figs. 6b and 8b,d.
Also, wouldn't show panels c,d for Southern ocean, but just mention in the text that the SO was excluded from analysis as available data are sufficient to derive KDE for only three decades.  If you really want to keep panels c,d in for consistency and as justification for insufficient data in the SO, then don't describe patterns in the main text.
Panel b: why does the y-axis go down to -30 per mil, when the minimum mean d13C POC value > -24 per mil?

Dataset
I have seen that dataset is stored in Pangaea; do you also plan to submit it to Isobank?

---

## Author Comment (AC1)

**Dear Anne Morée**

thank you very much for your detailed and helpful review!

We have elaborated answers to all points below, suggesting changes we want to realize in a following re-submission. Please find our responses directly alternating with your suggestions, which we greyed for better readability. Moreover, we numbered the comments of all reviewers to allow for cross-referencing.

**General Comments**

**RC1 – 1:** Thank you for your thorough and well-written manuscript. You have provided both the observational and modelling communities with a useful and unique d13C\_POC compilation that can be applied to a wide range of research questions and technical (model) evaluations. The details on the temporal and spatial distribution of the data are clearly presented and in general supported by informative figures.

**Reply RC1 - 1:** Thank you for your appreciation of our work.**

**RC1 – 2:** Even though your Introduction reads very well, I think it contains relatively few references. Consider adding some more references. A general one like Zeebe & Wolf- Gladrow (2001) for example?

**Reply RC1 – 2:** We will add this reference to the paragraph introducing the carbon isotopes as: "Carbon isotopes provide additional insights into the cycling of carbon in the Earth system (Zeebe and Wolf-Gladrow, 2001)."

**Furthermore, we are will add ten more references (see **Reply RC2 - 14**) in the introduction.**

**RC1 – 3:** You vary a bit in naming the two grids: The 'coarse grid', the 'UVic grid', 'the main dataset'. I think it is clearest if you refer to them as the 1x1 grid and the 1.8x3.6 grid, and only mention UVic/coarse/fine in the general introduction to the data at the beginning of Sect 3. If you want to present the 1.8x3.6 degree dataset as your main dataset, clarify this early on. In any case, choose a uniform naming.

**Reply RC1 – 3:** We follow your suggestion and will now refer to "Uvic grid" and "WOA grid" exlusively. The notation shall be given in Sect. 3 as: "[...] a  $1.8 \circ \times 3.6 \circ$  -resolution and 19 depth layers from a model that simulates  $\delta^{13}C_{POC}$  (e.g. Schmittner and Somes, 2016), in the following referred to as the UVic grid, and the  $1 \circ \times 1 \circ$ -resolution and 102 depth layer grid of the World Ocean

Atlas (Garcia et al., 2018), in the following referred to as the WOA grid."

**The following additional reference will be included:**

Garcia, H. E., K. Weathers, C. R. Paver, I. Smolyar, T. P. Boyer, R. A. Locarnini, M. M. Zweng, A. V. Mishonov, O. K. Baranova, D. Seidov, and J. R. Reagan, 2018. World Ocean Atlas 2018, Volume 4: Dissolved Inorganic Nutrients (phosphate, nitrate and nitrate+nitrite, silicate). A. Mishonov Technical Ed.; NOAA Atlas NESDIS 84, 35pp.

**RC1 – 4:** Related to comment (2), why do you choose to present the data in the 1x1 and 1.8x3.6 grids? The 1x1 is commonly used and in WOA format so this one I understand well. As a modeler not working with the UVic model, which I expect most of your dataset users will be, I would be interested in using either the raw data or a gridded 1x1 dataset with the time axis preserved (that is, Year and Month info). Splitting the 1x1 dataset up in decades like you did for the UVic grid could then also be helpful for users. Why do you focus on the UVic grid as the main dataset? As you might understand, I expect the 1x1 grid to have more potential to be used by the broader community.

**Reply RC1 - 4:** As you mentioned, the WOA grid is widely used and was taken for this reason. We decided to additionally present the data in the UVic grid, because this model is amongst those frequently applied to simulate  $\delta^{13}C_{POC}$  tracer. The model is currently used by us for data-model comparisons. The WOA data file contains the full spatial-temporal information, which has also been updated to follow the latest WOA18 (Garcia et a., 2018) grid and file structure. Therefore, all decadal averages can be easily obtained by averaging the respective decades in that file. We now individually provide the annual and monthly data including full temporal range (i.e. each temporal increment ranges across each year, i.e. 1964 to 2015). The analysis and visualization in the manuscript was mainly performed on the coarser UVic grid so the data points are easily visible. Since much of the data cover isolated grid points, it is often difficult to see the color of these isolated grid points on the finer 1x1 degree WOA grid in the manuscript. Therefore, we decided to show the spatial distribution on the 1.8x3.6 degree grid.

**RC1 – 5:** For e.g. your presentation in Sect. 5.2, Sect. 5.3 and Sect. 6, unless d13C\_POC at the surface is similar to d13C\_POC at depth (unlike d13C\_DIC), a depth-average might not be meaningful everywhere. I would then assume taking the surface values only is best. Adding a figure on d13C\_POC vs depth in your section on vertical distribution of the data would clarify this. In P20, I325 you indeed state that you only took the euphotic zone – why here and not in other places?

**Reply RC1 – 5:** Figure 5 in section 5.2 was meant to give an impression of where on the globe measurements were included, this is why we decided to include grid cells from every depth.

Indeed the north-south and seasonal trend analyses are also restricted to the euphotic zone. We will add in section 5.3 "[...] and restricted to the uppermost 130 m, which resemble the euphotic zone in the UVic model."

We will add in section 6.1: "Furthermore, we restricted our data to the uppermost 130 m, which resembles in the UVic model the euphotic zone."

We also follow the suggestion and now provide insight with respect to the vertical distribution of the data. For this we will add a vertical scatterplot of the data as a sub-panel of Figure 4. The figure will be split up into the main ocean basins (Arctic, Southern Ocean, North Pacific, South Pacific, ...) to visualize the distribution of the  $\delta^{13}C_{POC}$  data over the depth among the ocean areas.

**RC1 – 6:** Is it possible to give an educated guess on the uncertainty of d13C\_POC? This may vary per decade / sampling method / cruise, and I can imagine the source data do not give such

estimates themselves. But your experience may give the reader an estimate of the uncertainty, which is valuable for any further analysis.

**Reply RC1 - 6:** We agree with the referee's comment that uncertainty estimates would provide useful complementary information. Since we derived probability density estimates of all  $\delta^{13}C_{POC}$  measurements and of specific data subsets, we actually do have valuable information with respect to variability in the data, which includes methodological uncertainties as well as spatio-temporal variations in sampling. Since this comment is meaningful and we decided to add a table to the Appendix that lists statistical properties, such as mode, median, and 95% confidence intervals of the probability density estimates shown in Figures 1, 3, 6, 8, and 10. An explicit consideration of methodological uncertainties in association with the original measurements is difficult and it would require much additional expertise. For this we suggest to the readers to ascertain such details in the individual studies which we refer to. However, listing the derived statistical parameters is convenient, because these values provide information that appear to be more useful than the figures with the kernel density estimates alone.

Furthermore, we propose to add to all calculated KDEs from specific fractions of the data set the used number of available data points to the plot. With such additional information, we can give some intuitive insight in reliability and comparability of the results shown.

**RC1 – 7:** Last, I found that the dataset, even though in practical NetCDF format, does not contain enough information and does not follow conventions well enough to be worked with easily. Please see my comments in a separate Section 'The dataset'.

**Reply RC1 – 7:** The NetCDF files have been updated to follow the latest WOA18 spatial grids, a properly defined time axis. We included more information in many attributes associated with the files (e.g. brief description, reference to data set, and license) (see **Reply RC1 – 57- 61**).**

**RC1 – 8:** See also my Specific Comments below, as well as a few Technical Comments. I am happy to recommend your manuscript for publication after you have clarified my comments and hope they are useful for improving the manuscript.

**Reply RC1 – 8:** We really much appreciate your support and will address your comments in the following point by point.**

**Specific Comments**

**RC1 – 9:** P1, I8-9: the consistency with observations is a bit inherent as you compile all available observations? Also, barely any comparison is made in the text with older literature/data for consistency.

**Reply RC1 – 9:** We will clarify this statement by adding "[...] previously observed large-scale patterns". While we agree this is somewhat inherent, we still think it is important to state that measurements from individual transects added to the previous large-scale compilations have not significantly affected major trends previously observed (e.g. Young et al., 2013).

**RC1 – 10:** P2, I32: This is an example of a location where more and more relevant references would be in place (Rocha and Passow, 2014 only is limited to refer to for the reader to understand the role of the biological pump in sequestering C).

**Reply RC1 – 10:** We propose to add the additional reference McConnaughey et al. (1979) (see below) for this statement. In agreement with another reviewer's point (**RC2 – 17**) we also will replace the reference for the soft tissue pump by Volk and Hoffert (1985) and Banse (1990). The main point of this paragraph is to generally introduce the pathway of the creation and production of particulate organic carbon before more detailed information is provided about the isotope dynamics. We prefer not to introduce too many aspects here. For example, in one of the subsequent paragraphs (third paragraph hereafter) we have more references about isotopic analysis and insights on the biological carbon pump (lines 48-50 of originally submitted manuscript).

**Reference:**

**McConnaughey, T., and C. P. McRoy (1979), Food-web structure and the fractionation of carbon isotopes in the Bering sea, Mar. Biol., 53(3), 257–262.**

**RC1 – 11:** P2, I43-49: In this section you describe the fractionation during photosynthesis. I think it is important to mention somewhere in your introduction that there are three reactions where the C isotopes fractionate: calcification, photosynthesis and air-sea gas exchange and include relevant references. And that the relative importance of these processes depends on location: e.g. (Gruber et al., 1999; Morée et al., 2018; Schmittner et al., 2013).

**Reply RC1 – 11:** We propose to add a paragraph describing these three processes and their local variations: "Distributed along the carbon cycle the fractionation of  $\delta$ 13C is influenced by biological and thermodynamic processes (Gruber et al., 1999). Air-sea gas exchange plays a dominant role at the ocean surface. Phytoplankton photosynthesis and POC remineralization increase their influence in the ocean interior (Gruber et al., 1999; Morée et al., 2018). The processes are depending on circulation and temperature and thus their individual influence vary with geographic location (Gruber et al., 1999; Schmittner et al., 2013)." as well as the suggested references.

**RC1 – 12:** P2, I54: Underline your statement with some references from both the land biosphere and marine realm. For example, the Suess effect is visible in the ocean in d13Cof DIC (Eide et al., 2017).

**Reply RC1 - 12: We propose to add here the references:**

Eide, M., Olsen, A., Ninnemann, U. S., and Johannessen, T.: A global ocean climatology of preindustrial and modern ocean  $\delta$ 13C, Global Biogeochemical Cycles, 31, 515–534, https://doi.org/10.1002/2016gb005473, 2017.

Levin, I., Schuchard, J., Kromer, B., and Münnich, K. O.: The Continental European Suess Effect, Radiocarbon, 31, 431–440, https://doi.org/10.1017/s0033822200012017, 1989.

Ndeye, M., Sène, M., Diop, D., and Saliège, J.-F.: Anthropogenic CO2 in the Dakar (Senegal) Urban Area Deduced from 14C Concentration in Tree Leaves, Radiocarbon, 59, 1009–1019, https://doi.org/10.1017/rdc.2017.48, 2017.

**RC1 – 13:** P3, I57-60: To what extent are changes in the other fractionation pathway, air-sea gas exchange relevant to your study? The temperature dependence of fractionation during air- sea gas exchange (Zhang et al., 1995) suggests that in a warming world fractionation is weaker over the airsea interface. Also, Young et al. (2013) reconstructed that the fractionation factor during photosynthesis is changing due to rising CO2 concentrations. If the d13C\_DIC in the euphotic zone is different due to the Suess effect, your d13C\_POC is affected. In an ESSD article the discussion on this is not necessary, but I think it is important to point the reader to such studies that are relevant for the interpretation of your dataset.

**Reply RC1 – 13:** Changes in fractionation pathways need to be taken into account when analyzing the data in the context of a specific scientific question. A discussion of fractionation pathways is out of scope here, but we agree to mention aspects that are relevant for data interpretation. To state this better, we propose to change this part to: "However, changes in marine  $\delta^{13}C_{POC}$  are also significantly influenced by changes in phytoplankton fractionation due to other anthropogenic controls. For example increasing CO2 [aq] concentrations increase surface  $\delta^{13}$ C fractionation (Young et al., 2013), changing phytoplankton communities and increasing temperature influences phytoplankton growth rates and  $\delta^{13}$ C fractionation over the air-sea interface (Zhang et al., 1995). But determination of the driving processes(es) of  $\delta^{13}C_{POC}$  spatial and temporal trends remains a challenge." as well as the suggested references.

**RC1 - 14:** P3, I62: Please make the transition from the previous paragraph to this one more fluent.

**Reply RC1 – 14:** We propose to adapt the first sentence of the following paragraph to read the transition as follows: "A better understanding of the contributions from all of these effects requires a robust global data set of  $\delta^{13}C_{POC}$ .

Theoretical projection and understanding of changes associated with  $\delta^{13}C_{POC}$  can be executed by models of different scales, which include  $\delta^{13}C_{POC}$  circulation."

**RC1 – 15:** P3, I65: When it comes to the implementation of the C isotopes in the ocean component of ESMs, some recent advances could be highlighted here as well: e.g. (Liu et al., 2021; Tjiputra et al., 2020).

**Reply RC1 – 15:** We propose to add the sentence: "Recent model approaches support long-term past climate projections (Tjiputra et al., 2020) and assess estimations of the Suess effect (Liu et al., 2021)."

**RC1 – 16:** P3, I69-74: Please add some references as examples of your statements (especially in line 71).

**Reply RC1 – 16:** We propose to add here the examples of the incorporated merged data sets by Goericke and Tuerena, which were both set up with a specific research purpose.

RC1 - 17: P3, I80: what do you mean with multilateral here?

**Reply RC1 – 17:** For clarification we propose to rephrase (combine) two sentences: "The metadata comprise information about sampling location, time, depth and method as well as the original source, which makes original raw data values, method, and further technical description easily accessible." **RC1 – 18:** P3 I87 – p4, I90: This sentence is inconsistent with/incomplete as compared to p3, I76-78: You say here that you included unpublished data Lorrain and Tuerena but earlier just Lorrain. And you don't cite Tuerena et al. (2019) here which you did before.

**Reply RC1 – 18:** We propose to change this sentence to: "[...] we included unpublished data provided by Lorrain and the data products from Tuerena et al. (2019), Goericke (1994) [...]"**

**RC1 – 19:** P4, I107-110: Repetitive; You mention twice in this section that this is the Tuerena data set.

**Reply RC1 - 19:** We will exclude the second mention to let the part read as: "[...] to which we will refer to as the Tuerena data set. This contains 595 data points including 501 from Young et al. (2013) and covers samples within the euphotic zone and an observation timeframe of 1964 - 2012."

**RC1 – 20:** P5, I129: Wherever ... one type ; this sentence is difficult to read, please rephrase. It is also not clear to me how you choose between the similar measurements, and what made them similar (the value?). Are the two following sentences the only two times you have done this? In that case the sentence could end with a ':'.

**Reply RC1 – 20:** We will clarify this better as follows: "Wherever multiple  $\delta^{13}C_{POC}$  datasets from a single event (time, place, investigator) where provided by an author within one source, we chose one of them: [...]". This paragraph was supposed to explain the handling of the appearance of multiple datasets from a single event (time, place, investigator). This has happened in the two described cases, where each provided different stages of data processing or measurement types.

**RC1 – 21:** P5, I141: Why provide it on the UVic grid except for that they have d13C\_POC as output? Wouldn't it be more logical to present it on the WOA 1x1 degree grid and provide for example a Ferret/CDO/NCO guide on how to change it to a different grid format (or do you loose too much information in this case, regridding twice)?

**Reply RC1 – 21:** Indeed, switching from the WOA grid to the UVic grid or to any other model grid would be possible, but would also add uncertainty to such double gridded data product. Your suggestion is inspiring, but it would not be straightforward to provide a generic grid-conversion script that could account for the few but grid-specific pitfalls of inter- and extrapolation. Our choice to also provide the data on the UVic grid is pragmatic, as colleagues who work with UVic desired such data set to become available. We want to provide the data for the UVic grid in addition to the WOA gridded data set. Another, although minor, aspect was that global  $\delta^{13}C_{POC}$  data is easier to visualize on the coarse UVic grid.

**RC1 – 22:** Also, based on this paragraph I would expect 3 files in the dataset – I think this is a good place to tell the reader how many files you have (and therefore also that you split them up in decades), what their purpose is, what they contain, etc.

**Reply RC1 – 22:** We propose to give the reader a clearer insight into the availability and content of provided data. These have changed due to the update of the NetCDF files (see **Reply RC1 – 7**, **Reply RC1 – 61**) and we will describe them in the paper as follows: "[...] Interpolation required

availability of full spatial information (latitude, longitude and depth) of included  $\delta^{13}C_{POC}$  data to locate them on the grid.

On the WOA grid we provide thirteen NetCDF files containing only data with full spatio- temporal metadata: One is, averaging all observations from each year together, each year accounting for a time increment on the time axis. The other twelve files are averaging only observations from an individual month with again each year accounting for a time increment on the time axis. These files provide a variety of analysis opportunities, but also limited content of  $\delta^{13}C_{POC}$  data.

On the UVic grid we provide seven individual NetCDF files: Six of them are each representing one of the decades 1960s to 2010s containing all data, which were able to assign to their respective decade. One file contains all available  $\delta^{13}C_{POC}$  data completely independent of their measurement time. This individual provision of data on a decadal and overall time scale increases the fraction of usable  $\delta^{13}C_{POC}$  data for the following analyses."

**RC1 – 23:** P6, Table 1: Do I read it correctly that dataset Degens et al (1968) till Wada et al. (1987) all had their longitude changed? This is not entirely clear because it is empty behind all rows except for Degens'. If the Table should be read such that multiple rows have undergone the same change, I would suggest using curly brackets and centering the change description behind those.

**Reply RC1 – 23:** Yes, this is correct. We propose to just fill the table cell contents even though this results in repetition. Using curly brackets resulted in a very broad table that does not fit on the page. This was done for Tables 1 and 2 for consistency.

**RC1 – 24:** P7, I 156: Why did you leave out the sample day? This does not really confirm your statement that you have taken as many details as possible (p4, I114).

**Reply RC1 – 24:** Most of the historical data did not provide this specific temporal information in the obtained data files. Our main goal here was to create a monthly climatology.

**RC1 – 25:** P7, Sect 3.2: Could to the interpolated data description be added what the dimensions are of each dataset (lon, lat, depth, time?) and the size of these dimensions?

**Reply RC1 – 25:** We will add the dimensions and their sizes as: : "[...] we chose the grid of the UVic model version 2.9, as used e.g. in Schmittner and Somes (2016). Horizontally, it consists of  $100 \times 100$  cells with a resolution of  $1.8^{\circ} \times 3.6^{\circ}$ , arranged from 0 to  $360^{\circ}$  in longitude (LON) and -90 to  $90^{\circ}$  in latitude (LAT). Vertically, it is split up into 19 vertical layers (DEPTH) [...]"

For the WOA grid we will add: "The the WOA grid is based on the  $1^{\circ} \times 1^{\circ}$  grid of the World Ocean Atlas (Garcia et al., 2018). It has a horizontal resolution of 360 arranged from –180 to 180° in longitude (LON) and 180 arranged from –90 to 90° in latitude (LAT) direction. Vertically, it is split up into 102 layers (DEPTH). The time axis (TIME) increments for each year from 1964 to 2015 by one and has a size of 52."

Moreover, we will discard the last sentence of this paragraph.

**RC1 – 26:** P8, Table 3: coverage of Depth, how can that be out of 4754 datapoints? The maximum is 4732, right?

**Reply RC1 - 26: Yes, this was a typo. We will change the depth coverage to be: "3917 / 4732"**

**RC1 – 27:** P8, I174: Why did you exclude some data here (because they e.g. lack depth information?)? Do you mean you used the data with full spatial-temporal coverage (thus datapoints that have lon, lat, depth and year and month? – please specify). I assume that if you for example not have the spatio-temporal full metadata information, you could also not add them to the UVic grid dataset (but P9, I186 suggest you did – how?)?

**Reply RC1 – 27:** Both interpolations require full spatial metadata coverage, to be able to locate the data points on the grid. On the UVic grid we did not include temporal information, hence we were able to also include those data points without month-information. In this case, we created six decadal files, where data was included by matching year-information, and an additional seventh file, where data was included completely independent of their time-information. We propose to clarify this by describing in the introduction of this section: "Interpolation required availability of full spatial information (latitude, longitude and depth) of included  $\delta^{13}C_{poc}$  data to locate them on the grid."

Moreover, we will add there: "On the WOA grid we provide thirteen NetCDF files containing only data with full spatio-temporal metadata: One is, averaging all observations from each year together, each year accounting for a time increment on the time axis. The other twelve files are averaging only observations from an individual month with again each year accounting for a time increment on the time axis."

And in the end of this section by: "[...] focus following descriptions of interpolated data on the coarse grid interpolations. These also includes data without month-information in the six decadal files and even completely without temporal information in the seventh time-independent file."

**RC1 – 28:** P12, I233-243: 'Overall, after accounting for spatial sampling bias by comparing with regions, the different methods are generally consistent with each other (Figure 3).' I do not conclude that as easily from Fig. 3, how did you account for spatial sampling bias by comparing with regions? (clarify this and check also p20, I355). I think you mean with*in* regions as stated in the abstract, but then still I do not see where you have made a comparison between the sampling methods within several different regions (Fig. 3 is just the Atlantic).

**Reply RC1 – 28:** We understand the reviewer's concern and realized that our explanation was vague. We do not resolve any particular spatial sampling bias explicitly. In general, we want to document some of the variability that we find in the data and clarify whether some of the variability could possible attributed to methodological differences. For simplification we wanted to restrict this comparison to data from only one well sampled ocean. Data from the Atlantic are sufficiently available to compare crudely between tropical, temperate, and polar regions, according to the latitudinal bounds. We propose to reformulate this paragraph (former line 230ff):

"For resolving differences between sampling methods we chose data from the Atlantic Ocean, which comprise all four major methods (with data embracing a region between 45° S and 80° N and 70° W and 20° E). In addition, data were distinguished between tropical, temperate, and polar subregions. By crudely sorting the data according to their sampling locations, we gain some insight to methodological variability within a subregion and may relate these to variations between the three subregions (Figure 3). Overall, we do not find any severe bias with respect to any particular method. Bottle data seem to be cover most of the lower  $\delta^{13}C_{POC}$  values that typically range between -28 ‰ and -21 ‰, which could be due to samples collected at greater depths. Intake and net measurements are rather restricted to the upper ocean layers and these methods often yield  $\delta^{13}C_{POC}$  larger than -25 ‰, with some polar net measurements being a notable exception (Figure 3d). For the tropical Atlantic (30° S - 30° N) the net and intake measurements vary around -21 ‰, with 95% confidence limits between -24 ‰ and -18 ‰ (see Table A in the Appendix). According to our comparison, we could not identify any method that yields much greater variance of  $\delta^{13}C_{POC}$  values than others. The spatio-temporal variations of the  $\delta^{13}C_{POC}$  compare well amongst different methods, but we advise caution when comparing bottle measurements with data of other methods because of potential differences in the depth range covered."

**RC1 – 29:** Also, why not use the biome regions here for consistency with your other regionally presented data?

**Reply RC1 – 29:** The sample type information is only available in the csv-file version of the data and not part of the interpolations. Hence, it is not straightforward possible applying the biome grid masks on data selected by sample type. Furthermore, there are gaps between the core biomes of Fay and McKinley (2014) that are subject to temporal variations (no clear distinction between neighboring biomes). We wanted to exploit all data available for the Atlantic Ocean. The splitting into only three subregions turned out to provide sufficient data in order to come up with a meaningful comparison with just enough differentiation between methods.

**RC1 – 30:** And in the paragraph that follows, if e.g. net data make up most of the data of the full Atlantic, then it is no surprise that the full data KDE is similar to the net data? In order to discuss Fig. 3, does one not need a plot of what fraction of the data is coming from what sampling method? E.g. as plot of number of data versus time with contributions from the different sampling methods or something similar?

**Reply RC1 – 30:** We propose to add the number of used data points to the plot for each created KDE. This shall give a better insight in how much data was used from which source and how (un)certain some results might be. We will do the same for Fig. 6, 8 and 10 for consistency.

**RC1 – 31:** Do these differences between the methods maybe give us an impression of the uncertainty of the d13C\_POC values (see also my general comment 5).

**Reply RC1 – 31:** Yes, these differences could provide some insight to uncertainties and spatiotemporal variations. Unfortunately, these alone do not give a quantitative estimate of uncertainty because of the high spatial and temporal variance in the observations. Furthermore, since these observations were not collected at the exact same time and location, it remains difficult to draw any quantitative conclusions about uncertainty from the different methods since they all fall within the variance of the observations. We reformulated the entire paragraph about the methodological differences (see **Reply RC1 – 28**).

**RC1 – 32:** P12, I240: Besides discussion the variance, I think it would be interesting to provide the reader with information on differences in the mean/median between the methods (e.g. in a region/the regions in Fig. 3 but also globally compiled).

**Reply RC1 – 32:** We like this suggestion and decided to add more details about the differences between the methods. Respective statistical parameters will be depicted in a Table in the Appendix, including median, major modes, and upper and lower 95% confidence limits (see **Reply RC1 - 6**). Furthermore, we propose to shorten the displayed variances in the figure to two digits after the floating point and add for each KDE their median as well.

**RC1 – 33:** P12, Sect 5.1.: Here you discuss mostly the density/number of data at a certain depth, take over the global ocean. It would be interesting to hear how this varies with region (so e.g. fewer very deep data in remote locations, etc.) and a plot of d13C\_POC versus depth (global mean or region, whatever is more meaningful/informative) as a Fig. 4b for example.

**Reply RC1 – 33:** To present an overview of the vertical distribution of the data, we will add a vertical scatterplot of the data (see last paragraph of **Reply RC1 – 5**).**

**RC1 – 34:** P13, Sect 5.2: How can the coarse resolution dataset be independent of time – clarify how you merged the time dimension? Are the d13C\_POC values of depth-averaged lon,lat data meaningful – the value would depend on how many deep measurements are included (or is deep d13C\_POC similar to surface d13C\_POC?). Why not just show the locations of the data with a black marker in Fig. 5 in order to show their horizontal distribution?

**Reply RC1 – 34:** For this plot we used the UVic grid, where we included all  $\delta^{13}C_{POC}$  data with spatial information and disregarded all eventual time information. This results in its independence of time. We propose to clarify this in the text as: "For the UVic grid we show data from the file including all data independent of time, the WOA grid is averaged over all times." We chose this kind of plot to include global locations as well as magnitude of  $\delta^{13}C_{POC}$  values.

**RC1 – 35:** P14, Table 4: I think these data er more logically represented as a histogram. If you think it is important to show the exact values, this could even be added into the histogram.

**Reply RC1 - 35: We will replace Table 4 by a histogram.**

**RC1 – 36:** P14, Sect 5.3: Did you average over all depths or use surface values? Also, I did not understand the first sentence of the Section (in which figure is this shown?): for which decade is Fig. 6 made? In Fig.6, instead of a mean vs. biome wouldn't a mean versus latitude plot contain more information as it then is not discretized into these biome intervals? Or do you need to define zones because of low spatial data coverage?

**Reply RC1 – 36:** We used the two uppermost layer (down to 130 m) and the time-independent UVic grid. All results are presented in Fig. 6. We will clarify this as: "We show the north-south trend of  $\delta^{13}C_{POC}$  over the Atlantic Ocean based on the time-independent UVic grid and restricted to the uppermost 130 m, which resemble the euphotic zone in the UVic model." We chose the biomes for this presentation to directly compare regions with mostly consistent biological and ecological properties. Of course the coverage is a factor and we are able to compare and present our desired regions well in this way.

**RC1 – 37:** P16, I291: Maybe help the reader by stating what that means for seasonal availability for each hemisphere?

**Reply RC1 - 37:** We will clarify this as: "[...] only winter months on both hemispheres exhibit less data."

RC1 - 38: P17, I 302: define 'enough' in 'enough datapoints' (also in the caption of Fig. 8).

**Reply RC1 – 38:** Mathematically, a KDE can be calculated from a single data point (which would just be a single kernel). But the fewer (or the narrower the range of the) data points are available, the narrower, steeper and larger (in terms of y-axis values) the KDE becomes. For July, November and December on the northern hemisphere there were only 3 or less data points available, which caused a KDE by magnitudes larger than the others. Hence, we decided to exclude them from the plot for better comparability of the others. None of these restrictions were applied in the calculation of means. We propose to better clarify this as: "[...] displayed all months with enough data points by a KDE and indicate same months by same colors. We excluded July, November and December on the northern hemisphere from this KDE representation, because these provided three or less data points within them, which resulted in a KDE that overgrew the others by magnitudes and made their visual comparison difficult." and in the caption of Fig. 8: "Not all months include enough (here more than three) data points for a comparable density estimation." Also, we would add the used numbers of data points here, like proposed for Fig. 6 (see **RC1 - 30**).

**RC1 – 39:** P19, Fig. 8: In the caption you write that b and d are means, but in the title and text (p17, I303) it says median – what is it? I think connecting the mean/median values with a line is a bit confusing especially in d – why not present the values in a small table?

**Reply RC1 - 39:** It is the median, as always. We will discard panels (b) and (d) and add a 2-rows table presenting the medians instead.**

RC1 - 40: P20, I352: not specific enough, why not specify which areas?

**Reply RC1 - 40: We will change this part to: "[...], especially the Atlantic and Indian Ocean."**

**Technical corrections**

RC1 - 41: P1, I2: They have for example been used to?

Reply RC1 - 41: We will change the sentence to: "They have for example been used to [...]"

RC1 - 42: P1, I16: via its atmospheric form

**Reply RC1 - 42:** We will change the sentence to: "[...] regulating climate via its atmospheric form [...]"

RC1 - 43: P2, I 40: ...- 1) · 1000 and remove the '.' at the end

**Reply RC1 - 43: We will remove the full stop.**

RC1 - 44: P2, I55: reference?

Reply RC1 - 44: We will add the additional references to this paragraph:

Rubino, M., Etheridge, D. M., Trudinger, C. M., Allison, C. E., Battle, M. O., Langenfelds, R. L., Steele, L. P., Curran, M., Bender, M., White, J. W. C., Jenk, T. M., Blunier, T., and Francey, R. J.: A revised

1000 year atmospheric  $\delta$ 13C-CO2record from Law Dome and South Pole, Antarctica, Journal of Geophysical Research: Atmospheres, 118, 8482–8499, https://doi.org/10.1002/jgrd.50668, 2013.

And we will change this sentence accordingly to: "Atmospheric  $\delta$ 13CCO2 has decreased from -6.5 ‰ in preindustrial times to -8.4 ‰ presently (Rubino et al., 2013)."

RC1 - 45: P3, I69: improve our understanding of marine carbon

**Reply RC1 – 45:** We will change the sentence to: "[...] improve our understanding of marine carbon [...]"

**RC1 – 46:** P4, I11: and KH13.

Reply RC1 - 46: We will change the sentence to: "NECTALIS 3 and 4 and KH13"

RC1 - 47: P4, I114: Table 3 referred to before table 1 and 2?

**Reply RC1 - 47: This will be changed.**

**RC1 - 48:** P5, I140: remover the '.' in front of (Verwega et al., 2021).

Reply RC1 - 48: We will remove the full stop.

RC1 - 49: P9, I190: refer to Figure 1?

**Reply RC1 - 49:** We will change the sentence to: "[...] by Gaussian kernel density estimation (KDE) in Figure 1."

RC1 - 50: P9, 1194: refer to Figure?

**Reply RC1 – 50:** We will change the sentence to: "[...] different measurement method applied (Figure 3)."

**RC1 – 51:** P9, I201: 'what indicates very little data points lying' should be 'which indicates that very few data points lie'

**Reply RC1 – 51:** We will change the sentence to: "[...] which indicates very little data points lie in this range."

RC1 - 52: P16, I 285-287: change /permil to ‰

**Reply RC1 – 52:** We will format the sentences to be: "[...] from -28 to -29 ‰ are those two located farthest south. The biome located farthest north contain the next lowest value at about -24 ‰."

RC1 - 53: P18, I310: 'highest maximum' seems double, reformulate?

Reply RC1 - 53: We will change the sentence to: "[...] A strong pronounced maximum in the [...]"

**RC1 – 54:** P20, I344: change to 'two different global grids', the word resolution here reads not well in this position. Or rephrase.

Reply RC1 - 54: We will change the sentence to: "[...] two different global grids [...]"

**RC1 – 55:** P20, l345: *relative* to their mean? As *an anomaly* to their mean?

**Reply RC1 – 55:** In agreement with the updated structure of the csv-file (see **Reply RC2 – 3**), we will change this part to; "[...] the  $\delta^{13}C_{POC}$ , their anomalies to their mean and all available [...]"

RC1 - 56: P23, I383-386: 2003a and 2003b instead of 2003an and 2003av?

**Reply RC1 - 56: We will change the references to be indicated by their years as:**

Altabet, M. A. and Francois, R.: Natural nitrogen and carbon stable isotopic composition in surface water at cruise NBP96-05, https://doi.org/10.1594/PANGAEA.128266, 2003a.

Altabet, M. A. and Francois, R.: Natural nitrogen and carbon stable isotopic composition of station NBP96-05-06-4, https://doi.org/10.1594/PANGAEA.128229, 2003b.

**The dataset**

**RC1 – 57:** Only after opening the dataset it became clear to me that the decadal files have all data of that decade saved together in one file (right? Or is the mean for each location?). The poc13\_univ\_i1.nc file was also unexpected based on the text, and seems to contain all data merged over time, but not over depth. This relates to my comment on P5, l141 – tell the readers how many files they will find and what they contain.

**Reply RC1 – 57:** These aspects of the data files were briefly described next to the data files on the PANGAEA data site. We have improved their description in the text and on the data site so these aspects are clearer.**

**RC1 – 58:** The TANN dimension has no description (ie netcdf attributes) but 'TANN:axis = "T" ', please add more information: The netcdf files state that the files follow the CF-1.6 convention. However, when checking this (https://pumatest.nerc.ac.uk/cgi-bin/cf- checker.pl), this seems not to be true for TANN. There is also TYR in the poc13\_year\_month\_woa\_c1.0.nc file with the same issue. The dimension 'record' in the last file also confused me.

**Reply RC1 – 58:** We have updated the WOA data files so they are easily accessible with a 4D variable and a properly defined the time axis.**

**RC1 – 59:** I think the naming of the x and y axis (which represent lon and lat) is not very intuitive or standard – why not use lon or long and lat?

**Reply RC1 - 59:** We have updated the data files with these improvements for axes names.**

**RC1 – 60:** Global&variable attributes: If your files end up being saved on a computer somewhere, based on the file name one would not be able to know what they exactly are, who made them, how they were made, etc. I think the absence of this information in the files themselves is important and should be addressed. Try to describe them in such detail that if none of the authors can be contacted for extra information and help, that the dataset is still usable. E.g., in the global attributes, include references to all sources (Table 1 and 2), reference to this article, a title, datetime\_start, datetime\_stop, size, contact, license (CC-BY 4.0?), method/how the dataset was made, description, and any other attributes of potential use. For all attributes and dimensions

make sure to give them at least where applicable units, long\_name, standard\_name, \_FillValue, missing\_value, description. Also, DEPTH has a strange missing\_value.

**Reply RC1 – 60:** We have updated the attributes in the data files to include this additional information.**

**RC1 – 61:** In the poc13\_year\_month\_woa\_c1.0.nc file, is dimension 'record' the month and TYR the year? Why are these not in one time dimension? A 5 dimensional variable, which it is now, is difficult to work with (for example, CDO can't do it).

**Reply RC1 – 61:** We have updated the data files with the finer WOA grid. We now provide one annual and 12 monthly files, each of which containing a 4-D variable containing a time axis with a yearly increment. We have verified that CDO can operate on these updated data files.**

**References**

Eide, M., Olsen, A., Ninnemann, U. S., & Eldevik, T. (2017). A global estimate of the full oceanic 13C Suess effect since the preindustrial. *Global Biogeochemical Cycles*, 31(3), 492-514. https://doi.org/10.1002/2016GB005472

Gruber, N., Keeling, C. D., Bacastow, R. B., Guenther, P. R., Lueker, T. J., Wahlen, M., Meijer, H. A. J., Mook, W. G., & Stocker, T. F. (1999). Spatiotemporal patterns of carbon–13 in the global surface oceans and the oceanic suess effect. *Global Biogeochemical Cycles*, *13*(2), 307-335. https://doi.org/doi:10.1029/1999GB900019

Liu, B., Six, K. D., & Ilyina, T. (2021). Incorporating the stable carbon isotope 13C in the ocean biogeochemical component of the Max Planck Institute Earth System Model. *Biogeosciences Discuss.*, 2021, 1-44. https://doi.org/10.5194/bg-2021-32

Morée, A. L., Schwinger, J., & Heinze, C. (2018). Southern Ocean controls of the vertical marine δ13C gradient – a modelling study. *Biogeosciences*, 15(23), 7205-7223. https://doi.org/10.5194/bg-15-7205-2018

Schmittner, A., Gruber, N., Mix, A. C., Key, R. M., Tagliabue, A., & Westberry, T. K. (2013). Biology and air-sea gas exchange controls on the distribution of carbon isotope ratios ( $\delta$ 13C) in the ocean. *Biogeosciences*, 10(9), 5793-5816. https://doi.org/10.5194/bg- 10-5793-2013

Tjiputra, J. F., Schwinger, J., Bentsen, M., Morée, A. L., Gao, S., Bethke, I., Heinze, C., Goris, N., Gupta, A., He, Y. C., Olivié, D., Seland, Ø., & Schulz, M. (2020). Ocean biogeochemistry in the Norwegian Earth System Model version 2 (NorESM2). *Geosci. Model Dev.*, *13*(5), 2393-2431. https://doi.org/10.5194/gmd-13-2393-2020

Young, J. N., Bruggeman, J., Rickaby, R. E. M., Erez, J., & Conte, M. (2013). Evidence for changes in carbon isotopic fractionation by phytoplankton between 1960 and 2010. *Global Biogeochemical Cycles*, *27*(2), 505-515. https://doi.org/10.1002/gbc.20045

Zeebe, R., & Wolf-Gladrow, D. (2001). CO2 in Seawater: Equilibrium, Kinetics, Isotopes (Vol. 65). Elsevier Science B.V.

Zhang, J., Quay, P. D., & Wilbur, D. O. (1995). Carbon isotope fractionation during gas-water exchange and dissolution of CO2. *Geochimica et Cosmochimica Acta*, *59*(1), 107-114. https://doi.org/http://dx.doi.org/10.1016/0016-7037(95)91550-D

**Dear Reviewer2**

thank you very much for your detailed and helpful review!

We have elaborated answers to all points below, suggesting changes we want to realize in a following re-submission. Please find them directly alternating with your suggestions, which we greyed for better readability.

**General Comments:**

**RC2** – 1: The authors present a new compilation of global measurements of POC  $\delta$ 13C values and a description and overview of the data set. This is an important goal: such a data set is currently lacking and could be an important tool for observing temporal changes, validating current ocean biogeochemical models that incorporate  $\delta$ 13C-POC, and generally exploring ocean carbon dynamics in the particulate phase.

**Reply RC2 – 1:** We appreciate the reviewer's general comment, pointing out the importance of $\delta^{13}C_{POC}$ data collections. The reviewer's comments were addressed point by point, as listed below.**

**RC2 – 2:** After downloading and considering the data set in addition to the summary manuscript, this effort leaves some questions. The creation of such a data set should be forward- thinking and demonstrate a clear vision for how it will grow. Some improvements are needed for this data set to be truly useful and forward-thinking.

**Reply RC2 – 2:** We have updated the csv file as well as the NetCDF files (see **Reply RC2 – 3, Reply RC1 – 7, Reply RC1 – 57-61**). The contact author of this paper (Christopher Somes) is planning to provide annual updates of the data set. We will include this vision in the conclusion part of the manuscript as presented in **Reply RC2 – 6**.**

Specific Comments:

**RC2 – 3:** Presentation of data as anomalies from a mean does not seem logical. As more data are added, the mean will change; thus, the data set needs to be presented as actual values.

**Reply RC2 – 3:** The insertion of anomalies instead of the original values turned out to be a misunderstanding between us and the data management of PANGAEA, which could eventually be clarified and has now been corrected. Data submissions at PANGAEA are usually original data, measured and analyzed by their authors. When these data are used, the data sets can and have to be cited. For a compilation of data, it is important not to get duplicate data into the PANGAEA database. However, if the collected data are used to derive statistical values, it is possible to provide these values in an additional file. The file now contains both, the anomalies as well as the original values.

**RC2 – 4:** In addition, it is unclear why anomalies are reported to so many decimal places. The original data are likely all reported to only one decimal place, possibly two, which is the maximum practical precision of typical isotope ratio instrumentation. The mean itself and the anomalies should not be presented to a precision exceeding 1-2 decimal places, depending on calculation of uncertainty (see detailed comments below).

**Reply RC2 – 4:** This escaped our attention and we are thankful for this comment. We revised the csv file accordingly and have decreased the decimal precision to 2.**

**RC2 – 5:** Many currently available sources of data are not included. I refer the authors to Close and Henderson (2020) for one example of a different list of publications containing oceanic POC  $\delta$ 13C data, which they incorporated in a recent depth-resolved global POC  $\delta$ 13C assessment, as well as a list of publications which they did not include for reasons specific to their assessment. They included more than 300 data points from Pedrosa- Pamies et al. 2018; Bishop et al. 1977; Jeffrey et al. 1983; Hurley et al. 2019; O'Leary et al. 2001; Trull et al. 2008; Saino 1992; Minagawa et al. 2001; Hernes and Benner 2002; Druffel et al. 2003. Additional data sources they did not include but listed and collectively contain hundreds more data points: Williams and Gordon 1970; Eadie and Jeffrey 1973; Druffel et al. 1996; Benner et al. 1997; Trull and Armand 2001; Hernes and Benner 2006; Close et al. 2014; Krishna et al. 2018; Liu et al. 2018; Griffith et al. 2012; Xiang and Lam 2020. Most of these data sources are not included in the data set presented here, and most of them are not archived in PANGAEA.

**Reply RC2 – 5:** We understand the reviewer's concern. The data collection of Close and Henderson (2020) was made available at a time after we had already collected, sorted and analyzed the 4732 data points presented herein. It is regrettable to have missed this opportunity to add these 303 data points, but additional credit should be given to the data collection provided by Close and Henderson (2020), which we now want to point out explicitly (see added changes below, in agreement with **Reply CC1 - 68**). We are explicit about all our data sources and we clarify in the text that our data collection is neither exclusive nor all-embracing. Our goal was to come up with a global data set that is informative and has sufficient potential for meaningful data-model comparisons on global scale.

We propose to add a paragraph at the end of section 2.1 as:"A recent collection of 303 measurements of  $\delta^{13}C_{POC}$  has been provided by Close and Henderson (2020), largely based on data gathered from individual publications referenced therein. Since our analyses originally relied on data sources that differed from those of Close and Henderson (2020) we find our collection to be yet incomplete. Especially measurements from national databases might provide a huge future benefit."

And we will change the first paragraphs of the discussion to: "[...] The starting point of our collection and analyses was the readily available data collection of Goericke (1994), which comprised 467 data points. Our primary objective was to elaborate this set of data by adding useful meta-information from the original publications and by introducing additional  $\delta^{13}C_{POC}$  measurements, as recorded in the world ocean data base PANGAEA and made available by Robyn Tuerena and Anne Lorrain. This way we could expand the data collection substantially, from the original 467 to 4732 data points. This new  $\delta^{13}C_{POC}$  data set provides the best coverage to date that will be a useful tool to help constrain many marine carbon cycling processes and pathways from ocean-atmosphere exchange to marine ecosystems, as well as to better understand observations and validate models. To ensure a dynamic growth of our data collection the corresponding author will provide annual updates of the data set. Furthermore, he may be contacted by any interested researcher, who would like to add their data to this collection."

We compared the global mean of the now available data collection of Close and Henderson (2020) with our data collection. The bias (difference between global means) is 0.3 ‰.

**RC2 – 6:** There are currently many different databases currently used for isotope data in the oceans, particularly across different national research agencies such as BCO-DMO in the U.S., National Institute of Oceanography in India, JAMSTEC's time-series data sets, Japan Oceanographic Data Center, etc.

**Reply RC2 – 6:** We attempted to rely on a world data base that archives data provided by many research groups internationally. We have to admit that we do not know how many data points were missed because we did not stretch our search to national data collections. We decided to add information about additional measurements that can be derived from different data sources and how we plan to account for them in the future (see **Reply RC2 – 5**).

**RC2 – 7:** Importantly, the current manuscript mentions a lack of data from the northern Pacific, but it has missed some publications containing such data, perhaps because European/Atlantic research results are disproportionately represented in PANGAEA.

**Reply RC2 – 7:** In our opinion, it is unclear whether a bias in data archiving exists or how severe such bias is. Since we do not know, we will refine our statement in the Conclusion to: "[...], but observed a lack of data in PANGAEA that cover northern Pacific region."**

**RC2 – 8:** Seeing as this initial data set is missing much existing data, how do the authors propose to keep the data set up to date as new data are produced and published, or not published but instead entered into other databases? In addition to the publications listed above, there are new data sources since 2020, such as from the Arabian Sea by Silori et al., from the Arctic by Xiang and Lam, and South China Sea by Yang et al.

**Reply RC2 – 8:** The contact author of this paper (Christopher Somes) is planning to provide annual updates of the data set. Furthermore, he may be contacted by interested researchers who would like to incorporate their data in the data set (see **Reply RC2 – 6**).**

Some additional major questions:

**RC2 – 9:** What is the reasoning behind including data without 3-dimensional spatial coordinates (latitude, longitude, depth)? 128 data points lack lat/lon data, and 837 data points lack depth data. Of what use are data points without 3-dimensional spatial information? There is an opportunity missed here to also include details of analytical methods that would serve as a quality control measure. Namely, did the original data sources describe acidifying the samples (i.e., can we be sure the data are POC rather than total PC?), using what acidification technique, and did they include a blank correction? Older data may have been produced using closed-tube combustion/dual inlet IRMS, whereas newer data were likely produced using EA-IRMS. Because sampling method is included, the lack of analytical method is notable.

**Reply RC2 – 9:** Our goal was to build upon the previously compiled data sets from Goericke and Young. Since the data files from these compilations included data with missing spatial values, we did not exclude them in case it may be possible to retrieve this additional information in the future. We did not receive a reply when contacting the corresponding authors from these respective papers. However, we did not want to exclude these data from our study. Since analytical errors are substantially smaller than the standard deviation of observations, and thus do not likely represent a significant source of uncertainty and variability in the observations, we did not document the analytical methodology for each of the over 4,000 observations. We thank the reviewer for raising this important issue. Unfortunately, it is not feasible for us to undergo such a time-intensive task within journal deadlines in this review process. We will incorporate this additional analytical information in future updates of the database.

**RC2 – 10:** Defining POC: For some of the sampling methods, a size fraction is not specified. For instance, in situ pump and MULVFS samples are often size fractionated, but there is no data field specified here for size fraction. Similarly, there are zooplankton net results in the current data set. Do the contents of a zooplankton net belong in a data set of POC? Many of the other collection methods listed here exclude zooplankton as components of passive POC, such as pre-screening of sediment trap samples through a 250-350 micron mesh to exclude zooplankton.

**Reply RC2 – 10:** Since there are many potential applications of this data set, we decided to include all data defined as POC by the original publication in PANGAEA. Note that we have provided the publication reference link for the observations in the CSV data file along with sampling methods, so this information is easily accessible for researchers specifically concerned with this issue. We decided not to exclude any data from our search in case it might be relevant for more specific analysis and applications of this data. Both the Goericke and Young data sets also included zooplankton in their analysis, and noted individual studies examining the difference between  $\delta^{13}$ C in POC and zooplankton was not significant (Brodie et al., 2011; Lorrain et al., 2003; McConnaughey and McRoy, 1979; Voß, 1991; Schell 1992 (personal communication to Goericke 1994)).

**References:**

Brodie, C. R., M. J. Leng, J. S. L. Casford, C. P. Kendrick, J. M. Lloyd, Z. Yongqiang, and M. I. Bird (2011), Evidence for bias in C and N concentrations and d13C composition of terrestrial and aquatic organic materials due to pre-analysis acid preparation methods, Chem. Geol., 282(3–4), 67–83.

Lorrain, A., N. Savoye, L. Chauvaud, Y. M. Paulet, and N. Naulet (2003), Decarbonation and preservation method for the analysis of organic C and N contents and stable isotope ratios of low-carbonated suspended par-ticulate material, Anal. Chim. Acta, 491(8), 125–133.

McConnaughey, T., and C. P. McRoy (1979), Food-web structure and the fractionation of carbon isotopes in the Bering sea, Mar. Biol., 53(3), 257–262.

Technical Corrections: Data set---

**RC2 – 11:** Analytical uncertainties are not reproduced here but should be. They should have been included in the original data sources. Often the uncertainty for bulk measurements is consistent across samples so may be mentioned only once in the source texts rather than being tabulated for each data point.

**Reply RC2 – 11:** We will add a statement noting the highest analytical uncertainties we found in our compilation in the text. We note that this is substantially lower than the standard deviation of

the observations, and thus not a significant contributor to variability in the observations. We propose to add this to the end of section 4 as: "Analytical errors and uncertainties are typically 0.2 per mil or lower (Young et al., 2013), and thus are not likely to significantly contribute to the much larger variance in the observations."

**RC2 – 12:** Underway data is reported as 0 m depth, but ship seawater intakes are usually several meters below the surface. For those interested in the surface microlayer, the distinction between 0 m and 5 m would be an important one.

**Reply RC2 – 12:** We are certain that the data does not include surface micro layer measurements, but only intake data, but we did not wanted to change the information given by the original publication. For better clarification we will add to the Raw data file section: "Data having been published as measured at 0 m were included as this, while no surface micro layer measurements were included."

RC2 - 13: What is "biological sample"? This does not sound like POC.

**Reply RC2 – 13:** This is the description from the obtained data file, noting that the variable name was POC. We prefer to refrain from including lengthy descriptions in the data file, and note that the reference and link to each original paper is included in the CSV data file.

RC2 - 13: How are CTD/rosette, CTD, bottle, and Niskin bottle different methods?

We clustered all of these as "bottle" data, since we assume they have been measured under most similar conditions. The above given different namings are those given by the original publishers of the data, which we did not want to change in the detailed data description given along in the csvfile. We would like to clarify this in the Sampling methods section as: ""Bottle" data include samples taken from Niskin bottles, PEP bottles, CTDs and samples collected via Seabird submersible pumps."

Text---

RC2 - 14: Introduction: contains unsupported claims: please include references.

**Reply RC2 – 14:** In agreement with the other reviewers we will add the following references throughout the introduction:

Volk, T., and M. I. Hoffert, Ocean carbon pumps: Analysis of relative strengths and efficiencies in ocean-driven atmospheric CO2 changes, in The Carbon Cycle and Atmospheric CO2: Natural Variations Archean to Present, The Carbon Cycle and Atmospheric CO2: Natural Variations Archean to Present, Geophys. Monogr. Ser., vol. 32, edited by E. T. Sundquist, and W. S. Broecker, pp. 99–110, AGU, Washington, D. C., 1985. (see **Reply RC2 – 17**)

Banse, K.: New views on the degradation and disposition of organic particles as collected by sediment traps in the open sea, Deep Sea Research Part A. Oceanographic Research Papers, 37, 1177– 1195, https://doi.org/10.1016/0198-0149(90)90058-4, https://www.sciencedirect.com/science/article/pii/0198014990900584, 1990. (see **Reply RC2 – 17**)

Zeebe, R. E. and Wolf-Gladrow, D.: CO2 in Seawater: Equilibrium, Kinetics, Isotopes, Elsevier Science B.V., Elsevier Oceanography Series, 65, 2001. (see **Reply RC1 – 2**)

Eide, M., Olsen, A., Ninnemann, U. S., and Johannessen, T.: A global ocean climatology of preindustrial and modern ocean  $\delta$ 13C, Global Biogeochemical Cycles, 31, 515–534, https://doi.org/10.1002/2016gb005473, 2017. (see **Reply RC1 – 12**)

Levin, I., Schuchard, J., Kromer, B., and Münnich, K. O.: The Continental European Suess Effect, Radiocarbon, 31, 431–440, https://doi.org/10.1017/s0033822200012017, 1989. (see **Reply RC1 – 12**)

Ndeye, M., Sène, M., Diop, D., and Saliège, J.-F.: Anthropogenic CO2 in the Dakar (Senegal) Urban Area Deduced from 14C Concentration in Tree Leaves, Radiocarbon, 59, 1009–1019, https://doi.org/10.1017/rdc.2017.48, 2017. (see **Reply RC1 – 12**)

Tjiputra, J. F., Schwinger, J., Bentsen, M., Morée, A. L., Gao, S., Bethke, I., Heinze, C., Goris, N., Gupta, A., He, Y.-C., Olivié, D., Seland, Ø., and Schulz, M.: Ocean biogeochemistry in the Norwegian Earth System Model version 2 (NorESM2), Geoscientific Model Development, 13, 2393–2431, https://doi.org/10.5194/gmd-13-2393-2020, 2020. (see **Reply RC1 – 15**)

Liu, B., Six, K. D., and Ilyina, T.: Incorporating the stable carbon isotope 13C in the ocean biogeochemical component of the Max Planck Institute Earth System Model, https://doi.org/10.5194/bg-2021-32, 2021. (see **Reply RC1 - 15**)

Rubino, M., Etheridge, D. M., Trudinger, C. M., Allison, C. E., Battle, M. O., Langenfelds, R. L., Steele, L. P., Curran, M., Bender, M., White, J. W. C., Jenk, T. M., Blunier, T., and Francey, R. J.: A revised 1000 year atmospheric  $\delta$ 13C-CO2record from Law Dome and South Pole, Antarctica, Journal of Geophysical Research: Atmospheres, 118, 8482–8499, https://doi.org/10.1002/jgrd.50668, 2013. (see **Reply RC1 - 44**)

McConnaughey, T., and C. P. McRoy (1979), Food-web structure and the fractionation of carbon isotopes in the Bering sea, Mar. Biol., 53(3), 257–262. (see **Reply RC1 – 10**)

Gruber, N., Keeling, C. D., Bacastow, R. B., Guenther, P. R., Lueker, T. J., Wahlen, M., Meijer, H. A. J., Mook, W. G., and Stocker, T. F.: Spa- tiotemporal patterns of carbon-13 in the global surface oceans and the oceanic suess effect, Global Biogeochemical Cycles, 13, 307–335, https://doi.org/https://doi.org/10.1029/1999GB900019, https://agupubs.onlinelibrary.wiley.com/ doi/abs/10.1029/1999GB900019, 1999. (see **Reply RC1 – 11**)

Schmittner, A., Gruber, N., C, A., Key, M. R. M., Tagliabue, A., and Westberry, T. K.: Biology and airsea gas exchange controls on the distribution of carbon isotope ratios (13C) in the ocean, Biogeosciences, pp. 5793–5816, https://doi.org/10.5194/bg-10-5793-2013, 2013. (see **Reply RC1 – 11**)

Zhang, J., Quay, P., and Wilbur, D.: Carbon isotope fractionation during gas-water exchange and dissolution of CO2, Geochimica et Cosmochimica Acta, 59, 107–114, https://doi.org/10.1016/0016-7037(95)91550-d, https://doi.org/10.1016/0016-7037(95)91550-d, 1995. (see **Reply RC1 - 13**)

RC2 - 15: Line 10: The reason for the focus on the Atlantic should be clarified (data coverage)

**Reply RC2 – 15:** We will include the superior data coverage of the Atlantic as follows: "An analysis of 1990s mean $\delta^{13}C_{POC}$ values in an meridional section across the best covered Atlantic Ocean shows [...]"**

**RC2 – 16:** Line 30: awkward phrase. Maybe "some particulate organic carbon" or "parts of the particulate organic carbon pool"

**Reply RC2 - 16: We will rephrased this part to: "Some particulate organic carbon (POC) sinks [...]"**

RC2 - 17: Line 32: please use an earlier reference for the soft tissue pump

**Reply RC2 – 17:** We propose to exchange the reference with Volk and Hoffert (1985) and Banse (1990) (see **Reply RC2 – 14**).**

**RC2 – 18:** Line 40: omit the factor of 1000 to adhere to generally accepted d13C terminology (see Coplen TB. 2011. Guidelines and recommended terms for expression of stable-isotope- ratio and gas-ratio measurement results. Rapid Commun Mass Spectrom. 25: 2538-2560).

**Reply RC2 - 18: We will do this in the next manuscript version.**

RC2 - 19: Lines 108 and 110: repetitive about referring to this as the Tuerena dataset.

**Reply RC2 - 19:** We will exclude the second mentioning to let the part read as: "[...] to which we will refer as the Tuerena data set. This contains 595 data points including 501 from Young et al. (2013) and covers samples within the euphotic zone and an observation timeframe of 1964 - 2012."

RC2 - 20: Line 115: this sentence about depth intervals & timeframes is confusing

**Reply RC2 – 20:** We will rephrase it to: "Spatial coordinates originally given as depth intervals were replaced by their respective mid points. Time intervals were not changed in this way. If they contained just one month or year this was taken, otherwise the time information was omitted."

RC2 - 21: Line 116: Longitude values were converted

Reply RC2 - 21: We will change the sentence to: "Longitude values were converted to [...]"

RC2 - 22: Line 123-124: The last part of this sentence is awkward

**Reply RC2 – 22:** We will rephrase the sentence to: "According to Table 3 we complemented the data with month, year, depth, sample method, cruise, trap duration and a references, wherever available."

RC2 - 23: Line 125: Need to elaborate on why some values were considered suspicious.

**Reply RC2 – 23:** We will better clarify this as: "Rounded values were adjusted to their source values as well as data with interchanged longitudinal information, which is in detail shown in Table 1."

RC2 - 24: Line 127: Run on sentence (comma splice)

**Reply RC2 – 24:** We will discard this paragraph, since all information is discussed before (see **Reply RC2 – 20**) and move the sentence "Sample depth given as "surface" was denoted as 1 m." there, too.

**RC2 – 25:** Line 127-128: Confusing sentence. An example would be helpful. What was done if this criteria was not met?

**Reply RC2 - 25: We decided to discard this paragraph (see Reply RC2 - 24).**

**RC2 – 26:** Lines 135-137: This information might fit better where longitude transformation was first discussed.

**Reply RC2 – 26:** We will shorten this part to: "[...] Longitude was converted to [-180°, 180°] from a [0°, 360°] format by Equation 2" and include the equation in the first paragraph of this section as: "[...] Longitude values were converted to the format [-180°, 180°] by the transformation [Equation 2]"

RC2 - 27: Lines 142-143: How did these datasets & interpolations address depth?

**Reply RC2 – 27:** We will better describe the depth levels of the two grids as: "[...] a  $1.8^{\circ} \times 3.6^{\circ}$  - resolution and 19 depth layers from a model that simulates  $\delta^{13}C_{POC}$  (e.g. Schmittner and Somes, 2016), in the following referred to as the UVic grid, and the  $1^{\circ} \times 1^{\circ}$ -resolution and 102 depth layer grid of the World Ocean Atlas (Garcia et al., 2018) [...]"

**See also Reply RC1 - 3 or Reply CC1 - 23.**

**RC2 – 28:** Equation 3: A mean should not be this much more precise than the precision of the individual measurements. The analytical uncertainty in individual measurements is likely between 0.05-0.2 per mil, and the original data were likely presented to a precision of 1-2 decimal places (per mil notation). The propagated uncertainty in the mean would likely be somewhere around 0.1-0.3 per mil. Therefore, the precision of the mean likely should not exceed 1-2 decimal places, purely from a standpoint of analytical realities.

**Reply RC2 – 28:** After re-clarification with the PANGAEA editor, we are now allowed to include the measurement values obtained in the data files so we don not have to rely exclusively on including these anomalies relative to the mean. See also **Reply RC2 – 4**.

RC2 - 29: Line 153: unnecessary second comma

**Reply RC2 - 29: Omitted.**

RC2 - 30: Line 154: Meaning of first sentence is unclear

**Reply RC2 – 30:** We propose to discard this sentence. Details are explained in the following paragraph.**

**RC2 – 31:** Line 172: The depth of the euphotic zone varies by location, so this statement is not wholly accurate.

**Reply RC2 – 31:** We propose to change this sentence to: "The two uppermost layers reach down to depths of 50 and 130 m respectively, and they are supposed to comprise the upper ocean's euphotic zone."

**RC2 – 32:** Lines 202-205: would be useful to mention at what depths these outlier data points were collected. Is there any context to these locations or depths that would hint at the reason for these low values?

**Reply RC2 – 32:** Yes, most of them are located at great depths in the vicinity of hydrothermal fields. Only one data point is from the ocean surface. We propose to add this (in agreement with **Reply CC1 - 31**)as: "[...] taken from Lein and Ivanov (2009) and Lein et al. (2006), measured in September or October 2003, around the location 10° N, 104° W and below 2500 m depth in the vicinity of a hydrothermal field close to the Pacific coast of middle America. The lowest outlier at  $\delta^{13}C_{POC} = -55.15\%$  is taken from Altabet and Francois (2003a) from November 1996 and at 62.52° S, 169.99° E at the ocean surface south from New Zeeland."

**RC2 – 33:** Line 206-209: Values higher than -10 per mil at depths between 636-901 m are very strange for a station where the total water depth is around 3000 m (this appears to be Line P, somewhere near Station 11-13). This may be a case where checking the acidification method and/ or contacting the authors would be appropriate.

**Reply RC2 – 33:** Although we agree this seems strange, as stated above, we decided not to exclude any already published observations found in our search to remain as consistent and objective as possible.

RC2 - 34: Line 242: extra "second"

Reply RC2 - 34: According to (Reply RC1 - 28) we will rephrase the whole paragraph.

RC2 - 35: Line 263-264: Not a full sentence; could combine with previous.

**Reply RC2 – 35:** We propose to combine them to: "Many cruises are visible as lines formed by connected grid cells in Figure 5, especially in the Atlantic and Indian Ocean and shorter in the Southern Ocean."

**RC2 – 36:** Line 267: Confusing sentence: discussing high values but mentioning explicitly low values, <30 per mil.

**Reply RC2 – 36:** We will change the sentence to: "Highest  $\delta^{13}C_{POC}$  values are evident in low latitude regions."

RC2 - 37: Line 283-284: Difficult to understand this sentence

**Reply RC2 – 37:** In agreement with another reviewer's point (**CC1 – 58**), we will change the end of this paragraph to: "[...] cover the Atlantic Ocean and extend to the Arctic Sea and parts of the Southern Ocean. The biomes are numbered 9 to 17, excluding 14. The biomes 15 to 17 are representing parts of the Southern Ocean and were restricted to 70° W and 20° E. Their locations are shown in Figure 6."

RC2 - 38: Line 293: Figure 7

**Reply RC2 - 38: We will correct this.**

RC2 - 39: Section 6.1: Somewhat confusing. Grouping data by season may help.

**Reply RC2 – 39:** We resolved the data according to months and therefore will correct the title, now referring to "month" rather than "season". For coherence, we will also change all mentioning of "season" to "month" and rename the following section to "Decadal variations".

RC2 - 40: Lines 300-301: Specifying the percentage of data is in this decade would be clearer

**Reply RC2 – 40:** We will change the sentence to: "Since more than 50 % of the available  $\delta^{13}C_{POC}$  data [...]"

RC2 - 41: Lines 301-302: How many data points is enough? How do you test this?

**Reply RC2 – 41:** Mathematically, a KDE can be calculated from a single data point (which would just be a single kernel). But the fewer (or the narrower the range of the) data points are available, the narrower and steeper and larger (in terms of y-axis values) the KDE becomes. For July, November and December on the northern hemisphere there were only 3 or less data points available, which caused a KDE by magnitudes larger than the others. Hence, we decided to exclude them from the plot for better comparability of the others. None of these restrictions were applied in the calculation of means. We tried to better clarify this as: "[...] displayed all months with enough data points by a KDE and indicate same months by same colors. We excluded July, November and December on the northern hemisphere from this KDE representation, because these provided three or less data points within them, which resulted in a KDE that overgrew the others by magnitudes and made their visual comparison difficult." and in the caption of Fig. 8: "Not all months include enough (here more than three) data points for a comparable density estimation."

**RC2 – 42:** Line 304: Possible data coverage issue here. The Southern hemisphere could have lower values due to more data coverage in the high-latitude Southern Ocean compared to the high-latitude Arctic.

**Reply RC2 – 42:** This is a good point, which we have not thought of. We agree and think that this should be mentioned in the text. We propose to change / add this thought to the Conclusion: "[...], where we observed that lowest values (< $\approx$  -28 ‰) can be found in the Southern Ocean whereas highest (> $\approx$  -22 ‰) are restricted to low latitudinal regions. This might also have influenced the observed lower  $\delta^{13}C_{POC}$  values on the southern hemisphere compared to the northern, due to the relatively good coverage of the Southern Ocean."

**See also Reply RC2 - 44.**

RC2 - 43: Line 305: How is this determined? Doesn't July have high values?

Reply RC2 - 43: We will rephrase the whole paragraph according to Reply RC2 - 44.

RC2 - 44: Line 306: Winter in which hemisphere?

**Reply RC2 – 44:** We will discuss the point more precisely and want to rewrite the whole paragraph as: "The monthly resolved variations of  $\delta^{13}C_{POC}$  do not reveal any significant seasonal pattern (Fig. 8, see also Table A in the Appendix (for this see **Reply RC1 – 32, Reply RC1 – 6**)). In general we find highest  $\delta^{13}C_{POC}$  values in the northern hemisphere, with median  $\delta^{13}C_{POC} = -20.4 \%$  in April and

 $\delta^{13}C_{POC} = -21.5 \%$  in October, which are typical months with enhanced primary production (northern hemisphere spring and autumn blooms). Similarly high median  $\delta^{13}C_{POC}$  values cannot be ascertained for any month with data of the southern hemisphere, where values of  $\delta^{13}C_{POC}$  above -20 ‰ have rarely been observed at any time of the year. In fact, there is an overall tendency towards low  $\delta^{13}C_{POC}$  values for the southern hemisphere, which becomes well expressed during the months April and September, with medians of  $\delta^{13}C_{POC} = -28.1 \%$  and  $\delta^{13}C_{POC} = -28.5 \%$  respectively. However, interpretations of this north-south trend should be treated with caution, because the apparent tendency is likely conditioned by some imbalance in the number of high-latitudinal data points. Compared to the number of data points from the Southern Ocean, samples from the Arctic Ocean are considerably underrepresented (see also Fig. 9). Furthermore, the discrimination between data of the northern and southern hemisphere is crude and we encourage to use our data collection for more advanced analyses of seasonal, monthly-based, changes in the  $\delta^{13}C_{POC}$ signal."

RC2 - 45: Line 309: Lowest what?

**Reply RC2 – 45:** We mean the lowest shown measurement values. We hope this will be clearer by the changes proposed in **Reply RC2 – 44**.**

**RC2 – 46:** Line 316: The phrase about the data points in 1960s and 1980s suggests that there might be similar numbers of data points; however, the 1980s have much more data but the spatial coverage is similarly poor.

**Reply RC2 - 46:** We will better clarify this as: "1980s are similarly sparse in spatial coverage as the 1960s."**

RC2 - 47: Line 333: why is the median for the 2000s not shown on the figure?

**Reply RC2 – 47:** For the Southern Ocean there were no data available to calculate a 2000s median. In agreement with **RC1 – 39** we will move the available medians to a table.**

RC2 - 48: Line 375: interest

**Reply RC2 - 48: We will correct this.**

Tables----

RC2 - 49: Table 1 caption: unnecessary comma in second sentence.

**Reply RC2 - 49: We will exclude this.**

RC2 - 50: Table 1 Eadie and Jeffrey entry under column "to": is this added from the source itself?

**Reply RC2 – 50:** Yes. In agreement with **RC1 – 23** we will repeat the cell contents even if they are repeated.**

**RC2** – **51:** Table 2 footnote 2: how did you address this for sample durations *not* entirely within an explicit month and year?

**Reply RC2 – 51:** We will clarify this in agreement with **Reply RC2 – 20** in the text as: "Time intervals were not changed in this way. If they contained just one month or year this was taken, otherwise the time information was omitted." and also add to the footnote: "Only for sample durations

entirely within an explicit month and year, otherwise information of time frames has been discarded."

RC2 - 52: Table 3: entry for depth under "coverage": typo in the total number of data points

Reply RC2 - 52: We will correct this to: "3917 / 4732"

**Dear Sarah Magozzi**

thank you very much for your detailed and helpful review!

We have elaborated answers to all points below, suggesting changes we want to realize in a following re-submission. Please find them directly alternating with your suggestions, which we greyed for better readability. Moreover, we numbered the comments of all reviewers to allow for cross-referencing.

**General comments**

**CC1 – 1:** I would like to thank the authors for their thorough manuscript. I applaud them for their effort to compile such an extensive dataset of all available d13C POC measurements across the global ocean over multiple depth layers and monthly and decadal time-scales. Datasets as such are highly valuable and find application in a variety of research and technical fields and are useful to calibrate/validate process-based, mechanistic isotope-enabled models. The dataset and its components are clearly presented and so are major patterns in d13C POC values across space and time.

**Reply CC1 – 1:** We are happy about your appreciation of our work and effort. We will address your comments in the following point by point.**

**CC1 – 2:** It would be also interesting to see if there are any trends with depth, and how these may vary among ocean areas.

**Reply CC1 – 2:** Yes, we agree and also see the benefit of presenting the vertical distribution of the data. A vertical scatterplot of the data will be included in our second submission of the manuscript as a sub-panel of Figure 4. This will be split up into the main ocean basins (Arctic, Southern Ocean, North Pacific, South Pacific, ...) to visualize the distribution of the  $\delta^{13}C_{POC}$  data over the depth among the ocean areas.

**CC1 – 3:** I provide a few – hopefully constructive – general and specific comments below and recommend this manuscript for publication after my comments are addressed or discussed.

**Reply CC1 - 3:** We appreciate this support very much.**

Issues that I noticed throughout the manuscript:

**CC1 – 4:** -data is plurar, datum is singular. So, please make sure to use plural forms when speaking about data

**Reply CC1 – 4:** We somehow missed to treat the plural form (of data) consistently throughout the entire manuscript and we will change this.**

CC1 - 5: -be sure to use present/past tenses and active/passive forms consistently

**Reply CC1 – 5**: We will have a detailed check through the manuscript before re-submission.**

**CC1 – 6:** -be consistent with terminology related to spatial distributions: coarse/fine grid/interpolation.

**Reply CC1 – 6:** We will consequently use the namings "UVic" and "WOA" grid (see also **Reply CC1 –** 23).**

**Specific comments**

Abstract

**CC1 – 7:** Line 1: I think that the correct terminology here is just 'marine particulate organic carbon stable isotope ratios (d13C POC)', without -13 here, as by definition the isotope ratio is given by the ratio of the heavy (carbon-13) to the light (carbon-12) isotopes.

**Reply CC1 – 7:** We propose to change the first sentence to: "Marine particulate organic carbon stable isotope ratios [...]"**

**CC1 – 8:** Line 13: need commas for statement regarding the Southern Ocean: 'except for the Southern Ocean, which shows a weaker trend, but contains...'.

**Reply CC1 - 8:** We will change this part to: "[...] Southern Ocean, which shows a weaker trend, but [...]"**

Introduction

CC1 - 9: Line 16: Consider changing 'it is regulating' with 'it regulates'

**Reply CC1 - 9: We will change this.**

**CC1 - 10:** Line 47: there are, not there are

**Reply CC1 - 10: We will change this.**

**CC1 – 11:** Lines 47-48: I don't particularly agree with this statement as I think that the factors and processes underlying fractionation during photosynthesis are fairly well-understood, and fractionation can be predicted with confidence from [CO2aq], phytoplankton growth rate and community composition (due to different cell sizes and geometries of different taxa). Furthermore, these factors are all governed, to an extent, by temperature, which makes temperature a key predictor for fractionation and d13C POC patterns. Perhaps you should mention that here.

**Reply CC1 – 11:** We generally agree and did not intend to give the impression that these processes are poorly understood. We will modify this sentence: "[...], although uncertainties remain regarding the quantification of the specific processes and mechanisms [...]"**

**CC1 – 12:** Line 58: concentration of CO2aq is also temperature dependent, and so is the distribution of phytoplankton communities. That is, all factors that exert a direct influence on photosynthetic fractionation are ultimately controlled by temperature, which is a major control on fractionation during photosynthesis by phytoplankton. I would stress this a little more in the Introduction.

**Reply CC1 – 12:** We agree and will add a statement temperature has a strong influence on almost all processes that directly govern fractionation: "We also stress that all of these processes are sensitive to temperature changes which adds additional complexity to understanding how fractionation may change in space and time."

**CC1 – 13:** Lines 59-61: again, I think we have a fairly good understanding of the processes causing variation in photosynthetic fractionation, but a dataset with extensive spatio-temporal coverage is certainly needed to investigate how trends change across space and time and the mechanisms underlying these changes as well as for calibrations/validations of process-based/mechanistic models with no data component.

**Reply CC1 – 13:** Here we have stated that the predicting spatial and temporal trends "remains a challenge" We think this is a fair statement given the many different processes listed above that can alter fractionation and have different sensitivities to environmental change.**

**CC1 – 14:** Line 65: please add citation Magozzi et al. 2017 Ecosphere here. This study models the C isotope fractionation during photosynthesis as a function of a suite of variables provided by the ocean biogeochemical model NEMO-MEDUSA and predicts spatial and temporal patterns in d13C POC across the global ocean over seasonal to decadal time-scales.

**Reply CC1 - 14: We will add this.**

**CC1 – 15:** Line 66: calibration AND validation. A major issue associated with isotope-enabled biogeochemistry models for the global ocean is the lack of reliable validation datasets, with sufficient spatio-temporal coverage to allow proper validation (sometimes, datasets are so scarce or so scarcely comparable that it almost makes more sense to 'trust' the mechanistic model, based on fairly well-known and understood processes, rather than calibrating the model to the available data)

**Reply CC1 - 15:** Definitely, we agree and will add "validation" to this sentence.**

CC1 - 16: Line 80: what does 'multilateral' mean?

**Reply CC1 – 16:** For clarification we propose to rephrase (combine) two sentences: "The metadata comprise information about sampling location, time, depth and method as well as the original source, which makes original raw data values, method, and further technical description easily accessible."

**CC1 – 17:** Line 83: don't need a capital W for we after the semi-column. Please fix this here and throughout the text, as well as in figure captions.

**Reply CC1 - 17:** We will fix this in this line and the tables and figures captions, wherever applicable.**

**2 Data acquisition**

**CC1 – 18: Line 89:** rephrase this as 'the adjustments that we conducted are described in the following sections' or something.

**Reply CC1 – 18:** We will rephrase this sentence to: "The adjustments that we conducted are described in the following."

**2.2 Adjustments made**

CC1 - 19: Line 122: Isn't this Sackett et al. 1966? With a double t?

**Reply CC1 - 19: Yes, thank you, we will correct this.**

3 Content and structure of the data set

**CC1 – 20:** Line 140: don't need a full stop before reference.

**Reply CC1 - 20: We will remove the full stop.**

**CC1 – 21:** Line 141: why are data presented as anomalies with respect to the global mean d13C POC value? I think that the authors explained this in lines 149-150 but it is not clear to me what they mean.

**Reply CC1 – 21:** This was due to a misunderstanding during the PANGAEA submission process, since duplicated data must not be archived in PANGAEA. This has now been clarified and corrected and we could now also include the raw data values in the data file provided (see **Reply RC2 - 3**).**

3.1 Raw data file

**CC1 – 22:** Lines 149-150: what do the authors mean by 'anomalies contain all relevant information...during first steps of model calibration'?

**Reply CC1 – 22:** We have now included the raw data values as well in the data file so clear up this confusion (see **Reply RC2 - 3**).**

3.2 Interpolated data sets

**CC1 – 23:** Please make sure that you use the terms coarse/fine and grid/interpolation consistently throughout the manuscript.

**Reply CC1 - 23:** We will consequently use the namings "UVic grid" and "WOA grid". The notation shall be given in the introduction of section 3 as: "The interpolated data is provided as NetCDF files on two different global grids: a  $1.8 \circ \times 3.6 \circ$  -resolution and 19 depth layers from a model that simulates  $\delta^{13}C_{POC}$  (e.g. Schmittner and Somes, 2016), in the following referred to as the UVic grid, and the  $1 \circ \times 1 \circ$ -resolution and 102 depth layer grid of the World Ocean Atlas (Garcia et al., 2018), in the following referred to as the WOA grid."

The following additional reference will be included:

Garcia, H. E., K. Weathers, C. R. Paver, I. Smolyar, T. P. Boyer, R. A. Locarnini, M. M. Zweng, A. V. Mishonov, O. K. Baranova, D. Seidov, and J. R. Reagan, 2018. World Ocean

**Atlas 2018, Volume 4: Dissolved Inorganic Nutrients (phosphate, nitrate and nitrate+nitrite, silicate). A. Mishonov Technical Ed.; NOAA Atlas NESDIS 84, 35pp.**

**CC1 – 24:** Line 165; '...seven different files, where six files contained an individual decade each' or something.

**Reply CC1 – 24:** We will change this part to: "[...] seven different files, where six files contained an individual decade each [...]"

**Also, we will update the file description (see **Reply RC1 – 22**) according to the newly updated NetCDF files (see **Reply RC1 – 4**).**

**CC1 – 25:** Lines 165-168: here you should maybe say that: on the coarse grid, data were interpolated independent on time, averaged across depths; interpolation on the fine grid only included data with complete spatio-temporal information, averaged across times and depths.

**Reply CC1 – 25:** We will add here: "[...] seven different files each of them independent of time and averaged over the available spatial information. Six of them contain [...] " and: "measurements with complete spatial-temporal information, averaged across times and space." See also the updated file descriptions in **Reply RC1 – 22**.**

**CC1 – 26:** Lines 168-175: why did you interpolated data onto two different grids? Couldn't you just interpolate on the fine grid and resample if you needed values interpolated on a coarser grid?

**Reply CC1 – 27:** This was a pragmatic decision. In addition to the WOA gridded data we derived the UVic gridded data for our own plans of data-model comparison studies. Since colleagues from the UVic users community stressed the desirability of such data set, we do not want to withhold our data product. Also, a concomitant effect is that the UVic gridded data allow for better visualization on the figures. Many of the isolated grid points were very difficult to see on the global 1x1 degree WOA grid in the manuscript (e.g. Figure 5b). Nevertheless, the full spatial-temporal information is also provided on the finer WOA grid for users that prefer to use it. The large-scale patterns are virtually identical in the two grids.

**CC1 – 28:** Line 176: would be helpful if you could please add a statement to say, in poor words, what the Ferret interpolation does and how that works; essentially a sentence that explains the eqns., briefly.

**Reply CC1 – 28:** We will add the sentence: "These average the irregularly measured data points within the ocean grid to one single data point representing each covered grid cell."**

4 Main data set characteristics

**CC1 – 29:** Heading: you always use data set, not dataset. Make sure to be consistent.

**Reply CC1 - 29: We will consistently use "data set".**

4.1 Range and outlier values

**CC1 – 30:** Line 199: before and after the two outer modes, respectively? Density declines to 0 at d13C POC values < than the more negative mode, and at values > than the more positive mode. Is that what you mean here, right?

**Reply CC1 – 30:** Yes, we will state this more clearly as: "A steep decline to zero is visible outside the two outer modes."**

**CC1 – 31:** Lines 204-205: it would be helpful to mention where these locations are. Also, please change 'the smallest outlier' with the 'most negative/the lowest outlier' or something, as it is not straightforward what you mean here with smallest outlier

**Reply CC1 – 31:** In agreement with another reviewer's point (**RC2 – 32**), we will change this part to: "[...] measured in September or October 2003, around the location 10° N, 104° W and below 2500 m depth in the vicinity of a hydrothermal field close to the Pacific coast of middle America. The lowest outlier at  $\delta^{13}C_{POC}$  =–55.15‰ is taken from Altabet and Francois (2003a) from November 1996 and at 62.52° S, 169.99° E at the ocean surface south from New Zeeland."

**CC1 – 32:** Lines 206-2010: again, I think it would be helpful if you could please add where these coordinates are

**Reply CC1 - 32:** We will add here: "[...] located 30.125° N, 42.117° W in the middle north Atlantic. [...] around 49° N, 130° W close to the American coast of the Pacific [...] Both were measured at the ocean surface in the south Pacific, in July at [...]"**

4.2 Sampling method

**CC1 – 33:** Line 220: What does it mean that different sampling method could be attributed to 67% of the data as meta information? That 67% of the data had associated sampling method information?

**Reply CC1 – 33:** Yes, we will state this more clearly as: "Around 67 % of the data had associated sampling method information, which are contributed by eighteen different sampling methods."**

CC1 - 34: Line 229: double brackets in reference

**Reply CC1 - 34: We will correct this.**

**CC1 – 35:** Lines 232-233: how did you account for spatial sampling bias? What do you mean here with 'by comparing with regions'? Simply that you compared d13C POC data obtained with different method within each region (Figs. 3B-d)?

**Reply CC1 – 35:** That is correct we compared observations only within the different regions as specified in Figure 3. We proposed a clearer description of the analyses in **Reply RC1 – 28**.**

**CC1 – 36:** Line 233: are they? It looks to me that in some cases different sampling methods provide different d13C POC values, e.g., between 30-60 N, bottle values are lower than values obtained with the other sampling methods

**Reply CC1 - 36: Yes. We have rephrased the whole paragraph in Reply RC1 - 28.**

**CC1 – 37:** Line 236-237: please rephrase this sentence as it is very complex and it is not clear to me what you mean

**Reply CC1 - 37: We have rephrased the whole paragraph in Reply RC1 - 28.**

CC1 - 38: Line 239: closely aligned

**Reply CC1 - 38: We have rephrased the whole paragraph in Reply RC1 - 28.**

**CC1 – 39:** In general, how do read from Fig. 3 the % of data collected with each method in each area?

**Reply CC1 – 39:** It is not possible to read this information from the plot. We took it out of the data before we created the plot. This information is given along the plot to give the reader an impression of the ratio of data. We will refine the plot and add the number of data points available for each KDE (see **Reply RC1 – 30**) to make this information easily accessible.

**CC1 – 40:** Line 240: rephrase with 'the variance... is approx. 3 per mil lower than the variance of all d13C POC values, which is approx. 5 per mil, the highest value observed here', or something.

**Reply CC1 – 40:** We propose to rephrase this to: "The variance of the intake and trap data is $\approx 3\%$ and lower than the variance of all $\delta^{13}C_{POC}$ together, which is $\approx 5\%$ , the highest value observed here."**

**CC1 – 41:** Line 242: show a pronounced, remove clearly. Also second is repeated twice. Also show a clear individual maximum, remove mostly.

**Reply CC1 - 41: We have rephrased the whole paragraph in Reply RC1 - 28.**

5 Spatial distribution

**CC1 – 42:** Line 245: please consider rephrasing this, e.g., 'we show the spatial distribution of d13C POC measurements across the global ocean surface and depths'. Data is plural, therefore 'most d13C POC data have been measured', please make sure to be consistent with use of plural for data throughout the manuscript.

**Reply CC1 – 42:** We propose to change this part to: "We show the spatial distribution of $\delta^{13}C_{POC}$ measurements across the global ocean surface and depths. Most $\delta^{13}C_{POC}$ data have been [...]"**

5.1 Vertical distribution of the data set

**CC1 – 43:** Line 250: if 80% of the data have associated depth info, depth is a fairly well-recovered metadatum, isn't it? Does that mean that most datapoints don't have associated T and sampling method info?

**Reply CC1 - 43:** We will remove this sentence to prevent confusion.**

CC1 - 44: Line 254: 'within the first 130 m'

**Reply CC1 - 44: We will correct this to: "Within the first 130 m [...]"**

**Reply CC1 - 45: We will correct this to: "[...] where nearly 1000 of them [...]"**

CC1 - 46: Line 255-256: '200 d13C POC values are available in the depth interval [...)'

**Reply CC1 - 46: We will correct this.**

**CC1 – 47:** Line 257: add respectively at the end of this sentence

**Reply CC1 - 47: We will change this.**

**CC1 – 48:** In addition to how many data points there are for each depth layer etc, it would be interesting to see a plot showing trends in d13C POC values with depth, similar to Figs. 6, 8 and 10 for biomes, months and years...

**Reply CC1 - 48:** To present an isight into the vertical distribution of the data we will add a vertical scatterplot of the data. See **Reply CC1 - 2**.**

5.2 Horizontal distribution of the data set

**CC1 – 49:** Out of curiosity: does the dataset include any d13C POC data for the Mediterranean Sea?

**Reply CC1 – 49:** Yes, it does. In particular, it includes two bottle sampled data points by Carlier et al. (Carlier, A., Le Guilloux, E., Olu, K., Sarrazin, J., Mastrototaro, F., Taviani, M., and Clavier, J.: Stable Isotope composition of particulate organic matter and fauna in a deep Mediterranean coldwater coral bank, https://doi.org/10.1594/PANGAEA.771095, 2011.) from October 2007.

**CC1 – 50:** Line 260: I tend to prefer the use of 'grid' over 'interpolation', as the interpolation is essentially the spatial/horizontal distribution of values which can be done over a grid, isn't it?

**Reply CC1 - 50: Yes, exactly. We will replace "interpolation" here with "grid" (see Reply CC1 - 23).**

**CC1 – 51:** Line 261: to set context for next sentence, mention here that data are averaged across all depths'.

**Reply CC1 - 51: We will add the sentence: "In both cases, we averaged data over all depths."**

**CC1 – 52:** Line 263, don't need a full stop after Figure 5, but comma.

**Reply CC1 - 52: We will correct this..**

**CC1 – 53:** Line 264: not sure what 'also, data locations of ... occur' means.

**Reply CC1 – 53:** We mean that there are also smaller sample spots visible in the plot. We will clarify this as: "Also, smaller sample spots occur, mainly located in the [...]"

CC1 - 54: Line 267: lowest?

**Reply CC1 - 54: We mean that the data coverage is in comparison very sparse in this area.**

CC1 - 55: Line 269: substitute 'with' with 'of' XX per mil

**Reply CC1 – 55:** We will correct this to: "[...] high values of $\approx$ -20 ‰."**

5.3 Meridional trend of d13C POC values

**CC1 – 56:** Line 276: again, make sure to be consistent with use of coarse/fine grid/interpolation.

**Reply CC1 – 56:** We will rephrase this to: "[...] based on the time-independent WOA grid and restricted to [...]"**

**CC1 – 57:** Line 279-280: description of colors and lines should be given in figure caption, not in the main text.

**Reply CC1 - 57: We will move this description to the caption.**

**CC1 – 58:** Line 283: what does 'but 14' mean? Biomes were numbered from 9 to 17, where 15-17 had to be cut to the given lateral range. Also, consider using longitudinal rather than lateral here.

**Reply CC1 – 58:** We will change this part to: "[...] cover the Atlantic Ocean and extend to the Arctic Sea and parts of the Southern Ocean. The biomes are numbered 9 to 17, excluding 14. The biomes 15 to 17 are representing parts of the Southern Ocean and were restricted to 70° W and 20° E. Their locations are shown in Figure 6."

**CC1 – 59:** Line 284: I think there is a mistake here, location of biomes in the Atlantic is shown in Fig. 6c, not Fig. 10.

**Reply CC1 - 59: Yes, indeed. We will correct this.**

**CC1 – 60:** Lines 285-287: please insert per mil symbol. Also consider changing 'the final biomes' with 'the biomes with more positive d13C POC values' or something.

**Reply CC1 – 60:** We will add the per mil symbol and propose to rephrase the last sentence to: "The biomes with more positive $\delta^{13}C_{POC}$ are in the lower latitudes show similarly higher values from -23 to -21‰."**

6 Temporal distribution of the dataset

**CC1 - 61:** Line 293: Fig. 7 here, not 5.

**Reply CC1 - 61: We will correct this in the next version.**

CC1 - 62: Line 295-296: latest data are (plural), and again in the following line

**Reply CC1 - 62: We will correct this in the next version to "are" each.**

6.1 Seasonal trends

**CC1 – 63:** You don't describe Figs. 8b,d but I think they're informative as they show the seasonal trend in d13C POC values.

**Reply CC1 – 63:** In agreement with another reviewer's point **(RC1 – 39)** we will move panels (b) and (d) to a table.**

**CC1 – 64:** Also, please make sure the distinction between winter/summer is clear for the /S hemisphere in this paragraph.

**Reply CC1 – 64:** We agree that this part was a bit confusing. We propose to discuss the point more precise and rewrite the whole paragraph according to **Reply RC2 – 44**.**

6.2 Multi-decadal trends

**CC1 – 65:** Line 328-329: remove 'both'. Also second is repeated twice.

**Reply CC1 – 65**: We will correct this part to: "[...] decades that show a second expressed density maximum [...]"**

**CC1 - 66:** Lines 329: main maximum shift or the shift in main maxima. Remove 'with every decade lower'.

**Reply CC1 – 66:** We will rephrase this sentence to: "The main maximum shift from the 1960s at $\delta^{13}C_{POC} \approx -19.9\%$ to the 2010s at $\delta^{13}C_{POC} \approx -23\%$ ."**

**CC1 – 67:** Lines 334-339: I would remove these lines, if you really want to keep Figs. 10c,d in. Or you could also remove the figures and just say that there are not enough data in the SO to investigate multi- decadal trends.

**Reply CC1 – 67:** We will remove the panels (b) and (d) and only show (a) and (c), where we can use the last one to emphasize the need for more Southern Ocean data. Also, we will add the number of used data points for each KDE (see **Reply RC1 – 30**). We would move (b) to an own figure to present the visible decrease and add a shaded area for the variance around the graph to give an insight in the certainty as you suggested in **CC1 – 81** for a similar plot.

**7 Conclusions**

**CC1 – 68:** It would be nice to have a paragraph in Conclusions with examples of research and technical questions that could be tackled/answered with datasets as such. These applications should link back to themes presented in the Introduction.

**Reply CC1 – 68:** We will add a statement in the Conclusions about some of the broader applications of this dataset. This shall be placed at the end of the first paragraph as: "This new  $\delta^{13}C_{POC}$  data set provides the best coverage to date that will be a useful tool to help constrain many marine carbon cycling processes and pathways from ocean-atmosphere exchange to marine ecosystems, as well as to better understand observations and validate models."

**CC1 – 69:** Additionally – and this my reflect my own research interests – I think that the authors should stress the importance of their dataset for calibration/validation of process-based, mechanistic models. A major issue related with the application of these models in ecology, for instance, has been the lack of suitable calibration/validation datasets, resulting in large and mostly

unknown uncertainties (models trusted more than data, as they're based on fairly well-understood mechanisms whereas data are scarce and often incomparable).

**Reply CC1 - 69:** We propose to add to / change in the introduction the parts: "For reliable calibrations and validations of such processed-based mechanistic models, a spatially and temporally comprehensive data set is essential. [...] But until today, there is a lack of suitable data sets as constraints. This results in large and mostly even unknown uncertainties in model results." and in Section 3.2 Interpolated data sets: "One major aim of this work is to support reliable validation and calibration of  $\delta^{13}C_{POC}$ -simulating models. Hence, we chose for the coarser interpolation the grid of the version 2.9 UVic model, as used e.g. in Schmittner and Somes (2016)."

**CC1 – 70:** Datasets like this one provide a validation tool for mechanistic model, and potential for the development of data-based models of the spatio-temporal distributions of stable isotopes in marine ecosystems. An approach that has been successfully used to develop data-based isoscapes is the INLA method (St John Glew et al. 2019 MEE, St John Glew et al. 2020 ESSOAr), which allows separating spatial from non-spatial components of isotope variance when predicting spatial isotope patterns. This dataset could be suitable for such approach as it contains some meta information (e.g., sampling method, depth, month, decade, etc.) which can be included as factor to estimate non-spatial variance when predicting spatial variation from environmental covariate sets.

**Reply CC1 – 70:** Thank you very much for this idea! We will rephrase the last sentence to emphasize this point as: "The data set shows promise to better understand, constrain and predict carbon cycling as it provides a validation tool for mechanistic models and supports separation of non-spatial components in  $\delta^{13}C_{poc}$  variations."

Tables

**CC1 – 71:** Table 1 & others: don't need a capital letter after semi-column.

**Reply CC1 - 71:** We will correct this in the next version.**

**CC1 – 72:** The second column lists in which ..., without comma after lists.

**Reply CC1 - 72: We will remove this in the next version.**

**CC1 – 73:** The third and fourth columns, plural;

**Reply CC1 - 73: We will correct this in the next version.**

CC1 - 74: also unnecessary comma between show and from;

**Reply CC1 - 74: We will remove this in the next version.**

CC1 - 75: also show from what values to which values (or something).

**Reply CC1 – 75:** We will correct this in the next version to: "[...] from what values to which values [...]"**

CC1 - 76: Table4: change inspired with based on, or something.

**Reply CC1 – 76:** We will change this in the next version to: "[...] based on the coarse grid [...]". Also, we will change this table to a histogram according to **RC1 – 35.**

CC1 - 77: Not sure what the sentence starting with 'below 50 m...' means, why only below 50 m?

**Reply CC1 – 77:** The first 50 m are only one layer in the presented grid. To better resolve the distribution within this, we split it up. We propose to better clarify this as: "The uppermost 50 m are divided in subranges, below they are according to the coarse grid."

**CC1 – 78**: In the last sentence, depth range not depths range.

**Reply CC1 - 78: We will correct this in the next version.**

In general, we will change this table to a histogram in agreement of another reviewer's point (**RC1** – **35**).

Figures

**CC1 – 79**: Figure 1: in caption don't need capital V for values after semi-column. Please fix this here and in other figs' captions and throughout text.

**Reply CC1 - 79: We will fix this here and in the other captions as well.**

**CC1 – 80:** Figure 6b: can you plot mean lat for each biome on the x-axis, rather than biome number? Or at least an arrow N to S below the x-axis? You need to make this fig as much self-explanatory as possible.

**Reply CC1 - 80:** We are planning to add the mean latitudes from the biomes to the labels.**

**CC1 – 81:** Also, b can you plot some confidence intervals around means in panel b, given by variance of KDE? Alternatively, you could plot boxplots of for each KDE, without black line connecting means.

**Reply CC1 – 81:** We are planning to change this panel to a boxplot for the next version of the manuscript, but also appreciate the idea of the shaded variances and will use this for Figure 10.**

**CC1 - 82:** Figure 8b,d: Similar comments to Fig. 6b.

**Reply CC1 – 82:** We will discard panels (b) and (d) and show the values in a 2-rows table following another reviewer's point (see **RC1 – 39**).**

**CC1 – 83:** Figure 9: in caption, 'grid-locations of d13C POC data, colored by sampling decades' or something. Find a clear way to say that the grids of sample locations are shown here, colored by the decades in which the samples were collected.

**Reply CC1 – 83:** We will rephrase this as: "Grid locations of the $\delta^{13}C_{POC}$ data, colored by sampling decades."**

CC1 - 84: Aren't there any grids with multiple samples collected in different decades?

**Reply CC1 – 84**: Yes, this might happen. The plot sequence was 1960s to 2010s. We will better clarify this as: "The different colors indicate the different sample decades and were plotted increasing in time above each other."

**CC1 – 85:** Figure 10b,d: Similar comments to Figs. 6b and 8b,d.

Also, wouldn't show panels c,d for Southern ocean, but just mention in the text that the SO was excluded from analysis as available data are sufficient to derive KDE for only three decades. If you really want to keep panels c,d in for consistency and as justification for insufficient data in the SO, then don't describe patterns in the main text.

**Reply CC1 – 85:** We will remove the panels (b) and (d) and only show (a) and (c), where we can use the last one to emphasize the need for more Southern Ocean data. We will move (b) to an own figure to present the visible decrease. This should then show a shaded area of the variances around the graph as described above (see **Reply CC1 – 67**).

**CC1 - 86:** Panel b: why does the y-axis go down to -30 per mil, when the minimum mean d13C POC value > - 24 per mil?

**Reply CC1 – 86:** We made both rows of this plot share both axis for better comparability. We feared, if we would give the upper row (panel b) a different scale than the lower (panel d), this would lead to false interpretation of the relative magnitude of both changes. But according to **Reply CC1 – 85** we will discard this panel, since it does not contain enough meaningful information.

**Dataset**

CC1 - 87: I have seen that dataset is stored in Pangaea; do you also plan to submit it to Isobank?

**Reply CC1 – 87:** We contacted Brian Hayden many months ago about this possibility, but he told us that Isobank was, at that time, not yet ready. We will follow up on this again after publication.

---

## Author Response (AR1)

**Point-by-point response to reviewer comments**

Please find below the point-by-point responses to all three reviewers' comments. We numbered the comments and their answers to allow for cross-referencing. The referees comments are greyed for better readability. The given line number are referring to the tracked-changes version of the manuscript, which you will find at the end of this document.

**Comments by RC1**

**RC1 – 1:** Thank you for your thorough and well-written manuscript. You have provided both the observational and modelling communities with a useful and unique d13C\_POC compilation that can be applied to a wide range of research questions and technical (model) evaluations. The details on the temporal and spatial distribution of the data are clearly presented and in general supported by informative figures.

**Reply RC1 - 1:** Thank you for your appreciation of our work.**

**RC1 – 2:** Even though your Introduction reads very well, I think it contains relatively few references. Consider adding some more references. A general one like Zeebe & Wolf- Gladrow (2001) for example?

**Reply RC1 – 2:** We added this reference to the paragraph introducing the carbon isotopes. Furthermore, we are added ten more references (see **Reply RC2 – 14**) in the introduction.**

**Changes for RC1 – 2:** Added reference in I. 35: "Carbon isotopes provide additional insights into the cycling of carbon in the Earth system (Zeebe and Wolf-Gladrow, 2001)."**

**RC1 – 3:** You vary a bit in naming the two grids: The 'coarse grid', the 'UVic grid', 'the main dataset'. I think it is clearest if you refer to them as the 1x1 grid and the 1.8x3.6 grid, and only mention UVic/coarse/fine in the general introduction to the data at the beginning of Sect 3. If you want to present the 1.8x3.6 degree dataset as your main dataset, clarify this early on. In any case, choose a uniform naming.

**Reply RC1 – 3:** We follow your suggestion and now refer to "Uvic grid" and "WOA grid" exclusively.**

**Changes for RC1 – 3:** The notation is given in II. 176-179 as: "[...] a  $1.8 \times 3.6 \times 3.6$

The following additional reference was included:

Garcia, H. E., K. Weathers, C. R. Paver, I. Smolyar, T. P. Boyer, R. A. Locarnini, M. M. Zweng, A. V. Mishonov, O. K. Baranova, D. Seidov, and J. R. Reagan, 2018. World Ocean Atlas 2018, Volume 4: Dissolved Inorganic Nutrients (phosphate, nitrate and nitrate+nitrite, silicate). A. Mishonov Technical Ed.; NOAA Atlas NESDIS 84, 35pp.

See also Changes for CC1 - 6.

**RC1 – 4:** Related to comment (2), why do you choose to present the data in the 1x1 and 1.8x3.6 grids? The 1x1 is commonly used and in WOA format so this one I understand well. As a modeler not working with the UVic model, which I expect most of your dataset users will be, I would be interested in using either the raw data or a gridded 1x1 dataset with the time axis preserved (that is, Year and Month info). Splitting the 1x1 dataset up in decades like you did for the UVic grid could then also be helpful for users. Why do you focus on the UVic grid as the main dataset? As you might understand, I expect the 1x1 grid to have more potential to be used by the broader community.

**Reply RC1 - 4:** As you mentioned, the WOA grid is widely used and was taken for this reason. We decided to additionally present the data in the UVic grid, because this model is amongst those frequently applied to simulate  $\delta^{13}C_{POC}$  tracer. The model is currently used by us for data-model comparisons. The WOA data file contains the full spatial-temporal information, which has also been updated to follow the latest WOA18 (Garcia et a., 2018) grid and file structure. Therefore, all decadal averages can be easily obtained by averaging the respective decades in that file. We now individually provide the annual and monthly data including full temporal range (i.e. each temporal increment ranges across each year, i.e. 1964 to 2015). The analysis and visualization in the manuscript was mainly performed on the coarser UVic grid so the data points are easily visible. Since much of the data cover isolated grid points, it is often difficult to see the color of these isolated grid points on the finer 1x1 degree WOA grid in the manuscript. Therefore, we decided to show the spatial distribution on the 1.8x3.6 degree grid.

**RC1 – 5:** For e.g. your presentation in Sect. 5.2, Sect. 5.3 and Sect. 6, unless d13C\_POC at the surface is similar to d13C\_POC at depth (unlike d13C\_DIC), a depth-average might not be meaningful everywhere. I would then assume taking the surface values only is best. Adding a figure on d13C\_POC vs depth in your section on vertical distribution of the data would clarify this. In P20, I325 you indeed state that you only took the euphotic zone – why here and not in other places?

**Reply RC1 – 5:** Figure 5 in section 5.2 was meant to give an impression of where on the globe measurements were included, this is why we decided to include grid cells from every depth.

Indeed the north-south and seasonal trend analyses are also restricted to the euphotic zone.

We also followed the suggestion and now provide insight with respect to the vertical distribution of the data and added a vertical scatterplot of the data as a sub-panel of Figure 4. The figure is split up into the main ocean basins (Arctic, Southern Ocean, North Pacific, South Pacific, ...) to visualize the distribution of the  $\delta^{13}C_{POC}$  data over the depth among the ocean areas.

**Changes for RC1 – 5:** We added in II. 376-377 "[...] and restricted to the uppermost 130 m, which resemble the euphotic zone in the UVic model."

We added in II. 406-407: "Furthermore, we restricted our data to the uppermost 130 m, which resembles in the UVic model the euphotic zone."

We added three subpanels to Figure 5 (former Figure 4) on p. 18 with vertical scatterplots of the main ocean basins: Southern Ocean, Indian Ocean, Arctic Ocean, North and South Atlantic Ocean, and North and South Pacific Ocean and changed the caption to: "The vertical distribution of

available  $\delta$ 13CPOC samples is shown (a) as the approximated density of the measurement depths and (b - d) as measured  $\delta$ 13CPOC values relative to their respective measurement depth.(a) provides on the y-axis the estimated density of the depth values and on the x-axis the depth in m. The estimation was realized by a Gaussian KDE. (b) resolves the measurements of the Southern, Indian and Arctic Ocean, (c) the North and South Atlantic and (d) the North and South Pacific. The last three panels show on the y-axis the depth in m and on the x-axis the measured  $\delta$ 13CP OC value. Different colors are used to mark different ocean basins."

We changed II. 331-333: "The distribution of depth measurements is shown in Figure 4. An approximation of the depth measurements by Gaussian KDE is visualized in Figure 5 along with the d13POC value distribution over them in the main ocean basins."

**RC1 – 6:** Is it possible to give an educated guess on the uncertainty of d13C\_POC? This may vary per decade / sampling method / cruise, and I can imagine the source data do not give such estimates themselves. But your experience may give the reader an estimate of the uncertainty, which is valuable for any further analysis.

**Reply RC1 - 6:** We agree with the referee's comment that uncertainty estimates would provide useful complementary information. Since we derived probability density estimates of all  $\delta^{13}C_{POC}$  measurements and of specific data subsets, we actually do have valuable information with respect to variability in the data, which includes methodological uncertainties as well as spatio-temporal variations in sampling. Since this comment is meaningful and we decided to add a table to the Appendix that lists statistical properties, such as mode, median, and 95% confidence intervals of the probability density estimates shown in Figures 1, 3, 6, 8, and 10. An explicit consideration of methodological uncertainties in association with the original measurements is difficult and it would require much additional expertise. For this we suggest to the readers to ascertain such details in the individual studies which we refer to. However, listing the derived statistical parameters is convenient, because these values provide information that appear to be more useful than the figures with the kernel density estimates alone.

Furthermore, we propose to add to all calculated KDEs from specific fractions of the data set the used number of available data points to the plot. With such additional information, we can give some intuitive insight in reliability and comparability of the results shown.

**Changes for RC1 – 6:** We added the tables A1-A5 to the appendix listing the two most dominant modes, the median and the 95 % confidence intervals for each KDE shown in the Figures 1, 3, 6, 8, and 10.

We added the number of used data points to the labels of the KDEs in Figures 3, 7 (former 6), 9, (former 8) and 11 (former 10).

**We refer to the appendix tables in II. 257-258, 304-305, 380, 413, 443**

**RC1 – 7:** Last, I found that the dataset, even though in practical NetCDF format, does not contain enough information and does not follow conventions well enough to be worked with easily. Please see my comments in a separate Section 'The dataset'.

**Reply RC1 – 7:** The NetCDF files have been updated to follow the latest WOA18 spatial grids, a properly defined time axis. We included more information in many attributes associated with the files (e.g. brief description, reference to data set, and license) (see **Reply RC1 – 57- 61**).

**Changes for RC1 – 7:** Due to the update in out files, the file structure of the WOA grid changed, where we now provide thirteen different files: one averaging over each month and one over the full year.

We changed in I. 215: "[...] thirteen files [...].

We changed in I. 226-230: "This interpolation only includes d13CPOC data with full spatiotemporal metadata coverage, i.e. additional latitude, longitude and depth, we also require and include year and month information."

We changed in the caption of Figure 6: "[...] latitude and longitude information. The data shown in (b) include only data with complete temporal metadata and are averaged over the years 1964 - 2015. Both data are averaged over all measurements including data with missing depth information."

Also, we were able to resolve some issues within the Uvic grid, which resulted in slightly different plot results due to decreased number of data points located in the uppermost ocean layers in the decadal files.

We changed in I. 319: "[...] show a pronounced [...]".

We deleted in I. 320: "mostly".

We added in II. 320-322: "Median values of net and intake data are  $\approx$  1 to  $\approx$  2 ‰ higher than the one of the full data, respectively. This has a median of  $\delta$ 13CPOC = 22.46‰. Bottle and trap data show both a  $\approx$  2‰ lower median."

We added in II. 343-354: "Values of  $\delta$ 13CPOC are, apart from the North Pacific, closely aligned within the individual ocean basins. The Atlantic, South Pacific and Indian Ocean show values mostly of -28% to -19%. The  $\delta$ 13CPOC values in the Arctic reach down to  $\approx$ -30‰ and those in the Souther Ocean even to  $\approx$ -35‰. The North Pacific shows a wide spread of  $\delta$ 13CPOC values, especially between 50 and 100 m depth and at 2500 m depth. There they either reach down to less than -40% or up to more than  $\approx -10\%$  at a depth of 2500 m. Measurements in the North Atlantic, North Pacific and Indian Ocean reach down to more than 3500 m. Measurements down to nearly 5000 m were sampled in the Southern Ocean. The South Pacific was sampled down to a depth of 2500 m and the Arctic Ocean and South Atlantic only in the uppermost few hundred m."

We added in II. 424-429: "However, interpretations of this north-south trend should be treated with caution, because the apparent tendency is likely conditioned by some imbalance in the number of high-latitudinal data points. Compared to the number of data points from the Southern Ocean, samples from the Arctic Ocean are considerably underrepresented (see also Figure 10). Furthermore, the discrimination between data of the northern and southern hemisphere is crude and we encourage to use our data collection for more advanced analyses of seasonal, monthly-based, changes in the  $\delta$ 13CP OC signal."

**We deleted II. 433-434.**

**We changed II. 436-437: "[...] taken at locations in the Arctic Ocean [...]"**

**We deleted in I. 439: "Atlantic and the"**

**RC1 – 8:** See also my Specific Comments below, as well as a few Technical Comments. I am happy to recommend your manuscript for publication after you have clarified my comments and hope they are useful for improving the manuscript.

**Reply RC1 – 8:** We really much appreciate your support and will address your comments in the following point by point.**

**RC1 – 9:** P1, I8-9: the consistency with observations is a bit inherent as you compile all available observations? Also, barely any comparison is made in the text with older literature/data for consistency.

**Reply RC1 – 9:** We clarified this statement. While we agree this is somewhat inherent, we still think it is important to state that measurements from individual transects added to the previous large-scale compilations have not significantly affected major trends previously observed (e.g. Young et al., 2013).**

**Changes for RC1 - 9: We added in I. 9: "[...] previously observed large-scale patterns".**

**RC1 – 10:** P2, I32: This is an example of a location where more and more relevant references would be in place (Rocha and Passow, 2014 only is limited to refer to for the reader to understand the role of the biological pump in sequestering C).

**Reply RC1 – 10:** We added the additional reference McConnaughey et al. (1979) (see below) for this statement. In agreement with another reviewer's point (**RC2 – 17**) we also will replace the reference for the soft tissue pump by Volk and Hoffert (1985) and Banse (1990). The main point of this paragraph is to generally introduce the pathway of the creation and production of particulate organic carbon before more detailed information is provided about the isotope dynamics. We prefer not to introduce too many aspects here. For example, in one of the subsequent paragraphs (third paragraph hereafter) we have more references about isotopic analysis and insights on the biological carbon pump (lines 48-50 of originally submitted manuscript).

**Changes for RC1 – 10:** We added in I. 33 the reference: McConnaughey, T., and C. P. McRoy (1979), Food-web structure and the fractionation of carbon isotopes in the Bering sea, Mar. Biol., 53(3), 257–262.**

**RC1 – 11:** P2, I43-49: In this section you describe the fractionation during photosynthesis. I think it is important to mention somewhere in your introduction that there are three reactions where the C isotopes fractionate: calcification, photosynthesis and air-sea gas exchange and include relevant references. And that the relative importance of these processes depends on location: e.g. (Gruber et al., 1999; Morée et al., 2018; Schmittner et al., 2013).

**Reply RC1 - 11: We added a paragraph describing these three processes and their local variations.**

**Changes for RC1 – 11:** We added II. 44-48: "Distributed along the carbon cycle the fractionation of  $\delta$ 13C is influenced by biological and thermodynamic processes (Gruber et al., 1999). Air-sea gas

exchange plays a dominant role at the ocean surface. Phytoplankton photosynthesis and POC remineralization increase their influence in the ocean interior (Gruber et al., 1999; Morée et al., 2018). The processes are depending on circulation and temperature and thus their individual influence vary with geographic location (Gruber et al., 1999; Schmittner et al., 2013)."

And the suggested references:

Gruber, N., Keeling, C. D., Bacastow, R. B., Guenther, P. R., Lueker, T. J., Wahlen, M., Meijer, H. A. J., Mook, W. G., & Stocker, T. F. (1999). Spatiotemporal patterns of carbon–13 in the global surface oceans and the oceanic suess effect. *Global Biogeochemical Cycles*, *13*(2), 307-335. https://doi.org/doi:10.1029/1999GB900019

Morée, A. L., Schwinger, J., & Heinze, C. (2018). Southern Ocean controls of the vertical marine  $\delta$ 13C gradient – a modelling study. *Biogeosciences*, 15(23), 7205-7223. https://doi.org/10.5194/bg-15-7205-2018

Schmittner, A., Gruber, N., Mix, A. C., Key, R. M., Tagliabue, A., & Westberry, T. K. (2013). Biology and air-sea gas exchange controls on the distribution of carbon isotope ratios ( $\delta$ 13C) in the ocean. *Biogeosciences*, 10(9), 5793-5816. https://doi.org/10.5194/bg- 10-5793-2013

**RC1 – 12:** P2, I54: Underline your statement with some references from both the land biosphere and marine realm. For example, the Suess effect is visible in the ocean in d13Cof DIC (Eide et al., 2017).

Reply RC1 - 12: We added three more references.

Changes for RC1 - 12: We added in I. 61 the references:

Eide, M., Olsen, A., Ninnemann, U. S., and Johannessen, T.: A global ocean climatology of preindustrial and modern ocean  $\delta$ 13C, Global Biogeochemical Cycles, 31, 515–534, https://doi.org/10.1002/2016gb005473, 2017.

Levin, I., Schuchard, J., Kromer, B., and Münnich, K. O.: The Continental European Suess Effect, Radiocarbon, 31, 431–440, https://doi.org/10.1017/s0033822200012017, 1989.

Ndeye, M., Sène, M., Diop, D., and Saliège, J.-F.: Anthropogenic CO2 in the Dakar (Senegal) Urban Area Deduced from 14C Concentration in Tree Leaves, Radiocarbon, 59, 1009–1019, https://doi.org/10.1017/rdc.2017.48, 2017.

**RC1 – 13:** P3, I57-60: To what extent are changes in the other fractionation pathway, air-sea gas exchange relevant to your study? The temperature dependence of fractionation during air- sea gas exchange (Zhang et al., 1995) suggests that in a warming world fractionation is weaker over the air-sea interface. Also, Young et al. (2013) reconstructed that the fractionation factor during photosynthesis is changing due to rising CO2 concentrations. If the d13C\_DIC in the euphotic zone is different due to the Suess effect, your d13C\_POC is affected. In an ESSD article the discussion on this is not necessary, but I think it is important to point the reader to such studies that are relevant for the interpretation of your dataset.

**Reply RC1 – 13:** Changes in fractionation pathways need to be taken into account when analyzing the data in the context of a specific scientific question. A discussion of fractionation pathways is

out of scope here, but we agree to mention aspects that are relevant for data interpretation. To state this better, we propose to change this part to:

**Changes for RC1 – 13:** We changed II. 63-68 to: "However, changes in marine  $\delta^{13}C_{POC}$  are also significantly influenced by changes in phytoplankton fractionation due to other anthropogenic controls. For example increasing CO2 [aq] concentrations increase surface  $\delta^{13}$ C fractionation (Young et al., 2013), changing phytoplankton communities and increasing temperature influences phytoplankton growth rates and  $\delta^{13}$ C fractionation over the air-sea interface (Zhang et al., 1995). But determination of the driving processes(es) of  $\delta^{13}C_{POC}$  spatial and temporal trends remains a challenge."

And we added the suggested references:

Zhang, J., Quay, P. D., & Wilbur, D. O. (1995). Carbon isotope fractionation during gas-water exchange and dissolution of CO2. *Geochimica et Cosmochimica Acta*, *59*(1), 107-114. https://doi.org/http://dx.doi.org/10.1016/0016-7037(95)91550-D

Young, J. N., Bruggeman, J., Rickaby, R. E. M., Erez, J., & Conte, M. (2013). Evidence for changes in carbon isotopic fractionation by phytoplankton between 1960 and 2010. *Global Biogeochemical Cycles*, *27*(2), 505-515. https://doi.org/10.1002/gbc.20045

**RC1 - 14:** P3, I62: Please make the transition from the previous paragraph to this one more fluent.

Reply RC1 - 14: We adapted the first sentence of the following paragraph.

**Changes for RC1 – 14**: We changed I,72 to: "Theoretical projection and understanding of changes associated with  $\delta^{13}C_{POC}$  can be executed by models of different scales, which include  $\delta^{13}C_{POC}$  circulation."

**RC1 – 15:** P3, I65: When it comes to the implementation of the C isotopes in the ocean component of ESMs, some recent advances could be highlighted here as well: e.g. (Liu et al., 2021; Tjiputra et al., 2020).

Reply RC1 - 15: We added the sentence for this.

**Changes for RC1 – 15**: We added in II. 83-84: "Recent model approaches support long-term past climate projections (Tjiputra et al., 2020) and assess estimations of the Suess effect (Liu et al., 2021)."

**RC1 - 16:** P3, I69-74: Please add some references as examples of your statements (especially in line 71).

**Reply RC1 – 16:** We propose to add here the examples of the incorporated merged data sets by Goericke and Tuerena, which were both set up with a specific research purpose.

Changes for RC1 - 16: We added in I. 86: "[...] (e.g. Goericke, 1994; Tuerena et al., 2019)".

RC1 - 17: P3, I80: what do you mean with multilateral here?

Reply RC1 - 17: For clarification we rephrased (combined) two sentences.

**Changes for RC1 – 17**: We changed II. 93-96 to: "The meta-data comprise information about sampling location, time, depth and method as well as the original source, which makes original raw data values, method, and further technical description easily accessible."

**RC1 – 18:** P3 l87 – p4, l90: This sentence is inconsistent with/incomplete as compared to p3, l76-78: You say here that you included unpublished data Lorrain and Tuerena but earlier just Lorrain. And you don't cite Tuerena et al. (2019) here which you did before.

**Reply RC1 - 18: We changed this sentence.**

**Changes for RC1 – 18**: We changed II. 105-107 to: "[...] we included unpublished data provided by Lorrain and the data products from Tuerena et al. (2019), Goericke (1994) [...]"

**RC1 - 19:** P4, I107-110: Repetitive; You mention twice in this section that this is the Tuerena data set.

Reply RC1 - 19: We excluded the second mention.

**Changes for RC1 – 19**: We changed II. 128-130 to: "This contains 595 data points including 501 from Young et al. (2013) and covers samples within the euphotic zone and an observation timeframe of 1964 - 2012."**

**RC1 – 20:** P5, I129: Wherever ... one type ; this sentence is difficult to read, please rephrase. It is also not clear to me how you choose between the similar measurements, and what made them similar (the value?). Are the two following sentences the only two times you have done this? In that case the sentence could end with a ':'.

**Reply RC1 – 20:** This paragraph was supposed to explain the handling of the appearance of multiple datasets from a single event (time, place, investigator). This has happened in the two below described cases, where each provided different stages of data processing or measurement types.

**Changes for RC1 – 20**: We changed II. 156-162 to: "In two cases we identified multiple  $\delta^{13}C_{POC}$  datasets from a single event (time, place, investigator), where the data had been subject to different stages of processing types of measurements: In Westerhausen and Sarnthein (2003), we chose [...]. In Trull and Armand (2013a) and Trull and Armand (20013b), we used [...]".

**RC1 – 21:** P5, l141: Why provide it on the UVic grid except for that they have d13C\_POC as output? Wouldn't it be more logical to present it on the WOA 1x1 degree grid and provide for example a Ferret/CDO/NCO guide on how to change it to a different grid format (or do you loose too much information in this case, regridding twice)?

**Reply RC1 – 21:** Indeed, switching from the WOA grid to the UVic grid or to any other model grid would be possible, but would also add uncertainty to such double gridded data product. Your suggestion is inspiring, but it would not be straightforward to provide a generic grid-conversion script that could account for the few but grid-specific pitfalls of inter- and extrapolation. Our choice to also provide the data on the UVic grid is pragmatic, as colleagues who work with UVic desired such data set to become available. We want to provide the data for the UVic grid in addition to the

WOA gridded data set. Another, although minor, aspect was that global  $\delta^{13}C_{POC}$  data is easier to visualize on the coarse UVic grid.

**RC1 – 22:** Also, based on this paragraph I would expect 3 files in the dataset – I think this is a good place to tell the reader how many files you have (and therefore also that you split them up in decades), what their purpose is, what they contain, etc.

**Reply RC1 – 22:** We gave the reader a clearer insight into the availability and content of provided data. These have changed due to the update of the NetCDF files (see **Reply RC1 – 7, Reply RC1 – 61**)

**Changes for RC1 – 22**: We added II. 179-188: "[...] Interpolation required availability of full spatial information (latitude, longitude and depth) of included  $\delta^{13}C_{POC}$  data to locate them on the grid.

On the WOA grid we provide thirteen NetCDF files containing only data with full spatio- temporal metadata: One is, averaging all observations from each year together, each year accounting for a time increment on the time axis. The other twelve files are averaging only observations from an individual month with again each year accounting for a time increment on the time axis. These files provide a variety of analysis opportunities, but also limited content of  $\delta^{13}C_{POC}$  data.

On the UVic grid we provide seven individual NetCDF files: Six of them are each representing one of the decades 1960s to 2010s containing all data, which were able to assign to their respective decade. One file contains all available  $\delta^{13}C_{POC}$  data completely independent of their measurement time. This individual provision of data on a decadal and overall time scale increases the fraction of usable  $\delta^{13}C_{POC}$  data for the following analyses."

**RC1 – 23:** P6, Table 1: Do I read it correctly that dataset Degens et al (1968) till Wada et al. (1987) all had their longitude changed? This is not entirely clear because it is empty behind all rows except for Degens'. If the Table should be read such that multiple rows have undergone the same change, I would suggest using curly brackets and centering the change description behind those.

**Reply RC1 – 23:** Yes, this is correct. We filled the table cell contents even though this results in repetition. Using curly brackets resulted in a very broad table that does not fit on the page. This was done for Tables 1 and 2 for consistency.

**Changes for RC1 – 23**: We filled the gaps for repeating entries in the tables 1 and 2. Also, we abbreviated Latitude and Longitude by Lat and Lon, respectively.

**RC1 – 24:** P7, I 156: Why did you leave out the sample day? This does not really confirm your statement that you have taken as many details as possible (p4, I114).

**Reply RC1 – 24:** Most of the historical data did not provide this specific temporal information in the obtained data files. Our main goal here was to create a monthly climatology.

**RC1 – 25:** P7, Sect 3.2: Could to the interpolated data description be added what the dimensions are of each dataset (lon, lat, depth, time?) and the size of these dimensions?

**Reply RC1 – 25:** We will add the dimensions and their sizes. Moreover, we will discard the last sentence of this paragraph.

**Changes for RC1 – 25**: We changed II. 219-212 to: "Horizontally, it consists of  $100 \times 100$  cells with a resolution of  $1.8^{\circ} \times 3.6^{\circ}$ , arranged from 0 to  $360^{\circ}$  in longitude (LON) and -90 to  $90^{\circ}$  in latitude (LAT). Vertically, it is split up into 19 vertical layers (DEPTH), [...]"

We changed II. 225-227 to: "It has a horizontal resolution of 360 arranged from –180 to 180° in longitude (LON) and 180 arranged from –90 to 90° in latitude (LAT) direction. Vertically, it is split up into 102 layers (DEPTH). The time axis (TIME) increments for each year from 1964 to 2015 by one and has a size of 52."

**We discarded II. 229-230.**

**RC1 – 26:** P8, Table 3: coverage of Depth, how can that be out of 4754 datapoints? The maximum is 4732, right?

**Reply RC1 - 26: Yes, this was a typo.**

**Changes for RC1 - 26: We changed the depth coverage in Table 3 to: "3917 / 4732"**

**RC1 – 27:** P8, I174: Why did you exclude some data here (because they e.g. lack depth information?)? Do you mean you used the data with full spatial-temporal coverage (thus datapoints that have lon, lat, depth and year and month? – please specify). I assume that if you for example not have the spatio-temporal full metadata information, you could also not add them to the UVic grid dataset (but P9, I186 suggest you did – how?)?

**Reply RC1 – 27:** Both interpolations require full spatial metadata coverage, to be able to locate the data points on the grid. On the UVic grid we did not include temporal information, hence we were able to also include those data points without month-information. In this case, we created six decadal files, where data was included by matching year-information, and an additional seventh file, where data was included completely independent of their time-information. We clarified this.

**Changes for RC1 – 27**: We added in II. 244-246: "These also includes data without monthinformation in the six decadal files and even completely without temporal information in the seventh time-independent file."

**See also Changes for RC1 - 22 for changes in II. 179-188.**

**RC1 – 28:** P12, I233-243: 'Overall, after accounting for spatial sampling bias by comparing with regions, the different methods are generally consistent with each other (Figure 3).' I do not conclude that as easily from Fig. 3, how did you account for spatial sampling bias by comparing with regions? (clarify this and check also p20, I355). I think you mean with*in* regions as stated in the abstract, but then still I do not see where you have made a comparison between the sampling methods within several different regions (Fig. 3 is just the Atlantic).

**Reply RC1 – 28:** We understand the reviewer's concern and realized that our explanation was vague. We do not resolve any particular spatial sampling bias explicitly. In general, we want to document some of the variability that we find in the data and clarify whether some of the variability could possible attributed to methodological differences. For simplification we wanted to restrict this comparison to data from only one well sampled ocean. Data from the Atlantic are

sufficiently available to compare crudely between tropical, temperate, and polar regions, according to the latitudinal bounds. We propose to reformulate this paragraph (former line 230ff).

Changes for RC1 - 28: We reformulated II.294-309: "For resolving differences between sampling methods we chose data from the Atlantic Ocean, which comprise all four major methods (with data embracing a region between 45° S and 80° N and 70° W and 20° E). In addition, data were distinguished between tropical, temperate, and polar subregions. By crudely sorting the data according to their sampling locations, we gain some insight to methodological variability within a subregion and may relate these to variations between the three subregions (Figure 3). Overall, we do not find any severe bias with respect to any particular method. Bottle data seem to be cover most of the lower  $\delta^{13}C_{POC}$  values that typically range between -28 ‰ and -21 ‰, which could be due to samples collected at greater depths. Intake and net measurements are rather restricted to the upper ocean layers and these methods often yield  $\delta^{13}C_{POC}$  larger than -25 ‰, with some polar net measurements being a notable exception (Figure 3d). For the tropical Atlantic (30° S - 30° N) the net and intake measurements vary around -21 ‰, with 95% confidence limits between -24 ‰ and -18 ‰ (see Table A in the Appendix). According to our comparison, we could not identify any method that yields much greater variance of  $\delta^{13}C_{POC}$  values than others. The spatio-temporal variations of the  $\delta^{13}C_{POC}$  compare well amongst different methods, but we advise caution when comparing bottle measurements with data of other methods because of potential differences in the depth range covered."

**RC1 – 29:** Also, why not use the biome regions here for consistency with your other regionally presented data?

**Reply RC1 – 29:** The sample type information is only available in the csv-file version of the data and not part of the interpolations. Hence, it is not straightforward possible applying the biome grid masks on data selected by sample type. Furthermore, there are gaps between the core biomes of Fay and McKinley (2014) that are subject to temporal variations (no clear distinction between neighboring biomes). We wanted to exploit all data available for the Atlantic Ocean. The splitting into only three subregions turned out to provide sufficient data in order to come up with a meaningful comparison with just enough differentiation between methods.

**RC1 – 30:** And in the paragraph that follows, if e.g. net data make up most of the data of the full Atlantic, then it is no surprise that the full data KDE is similar to the net data? In order to discuss Fig. 3, does one not need a plot of what fraction of the data is coming from what sampling method? E.g. as plot of number of data versus time with contributions from the different sampling methods or something similar?

**Reply RC1 – 30:** We added the number of used data points to the plot for each created KDE. This shall give a better insight in how much data was used from which source and how (un)certain some results might be. We did the same for Fig. 7, 9 and 11 (former 6, 8 and 10) for consistency.

**Changes for RC1 – 30**: Figures 3, 7, 9 and 11 were added the number of used data points in the labels of the respective KDEs.**

**RC1 – 31:** Do these differences between the methods maybe give us an impression of the uncertainty of the d13C\_POC values (see also my general comment 5).

**Reply RC1 – 31:** Yes, these differences could provide some insight to uncertainties and spatiotemporal variations. Unfortunately, these alone do not give a quantitative estimate of uncertainty because of the high spatial and temporal variance in the observations. Furthermore, since these observations were not collected at the exact same time and location, it remains difficult to draw any quantitative conclusions about uncertainty from the different methods since they all fall within the variance of the observations. We reformulated the entire paragraph about the methodological differences (see **Reply RC1 – 28**).

**Changes for RC1 - 31: see Changes for RC1 - 28.**

**RC1 – 32:** P12, I240: Besides discussion the variance, I think it would be interesting to provide the reader with information on differences in the mean/median between the methods (e.g. in a region/the regions in Fig. 3 but also globally compiled).

**Reply RC1 - 32:** We like this suggestion and decided to add more details about the differences between the methods. Respective statistical parameters are depicted in tables in the Appendix, including median, major modes, and upper and lower 95% confidence limits (see **Reply RC1 - 6**). Furthermore, we shortened the displayed variances in the figure to two digits after the floating point and add for each KDE their median as well.

**Changes for RC1 – 32**: In Figure 3 we shortened the displayed variances in all panels and added the median for each KDE.

**See also Changes for RC1 - 6.**

**RC1 – 33:** P12, Sect 5.1.: Here you discuss mostly the density/number of data at a certain depth, take over the global ocean. It would be interesting to hear how this varies with region (so e.g. fewer very deep data in remote locations, etc.) and a plot of d13C\_POC versus depth (global mean or region, whatever is more meaningful/informative) as a Fig. 4b for example.

**Reply RC1 – 33:** To present an overview of the vertical distribution of the data, we will add a vertical scatterplot of the data (see last paragraph of **Reply RC1 – 5**).

**Changes for RC1 – 33**: We added three sub-panels to Figure 5 (former 4) and adapted the caption. See **Changes for RC1 – 5**.

**RC1 – 34:** P13, Sect 5.2: How can the coarse resolution dataset be independent of time – clarify how you merged the time dimension? Are the d13C\_POC values of depth-averaged lon,lat data meaningful – the value would depend on how many deep measurements are included (or is deep d13C\_POC similar to surface d13C\_POC?). Why not just show the locations of the data with a black marker in Fig. 5 in order to show their horizontal distribution?

**Reply RC1 - 34:** For this plot we used the UVic grid, where we included all  $\delta^{13}C_{POC}$  data with spatial information and disregarded all eventual time information. This results in its independence of time. We clarified this. We chose this kind of plot to include global locations as well as magnitude of  $\delta^{13}C_{POC}$  values.

**Changes for RC1 – 34:** We added in II. 357-358: "For the UVic grid we show data from the file including all data independent of time, the WOA grid is averaged over all times."

**RC1 – 35:** P14, Table 4: I think these data er more logically represented as a histogram. If you think it is important to show the exact values, this could even be added into the histogram.

**Reply RC1 - 35: We replaced Table 4 by a histogram.**

**Changes for RC1 – 35:** We replaced the table by a histogram and adapted the caption to: "Vertical data coverage in depth layers based on the UVic grid: The uppermost 50 m are divided in subranges, below they are according to the UVic grid. The number of  $\delta$ 13CPOC data points available are plotted against their respective depth range."

**This resulted in an increase of all following figure numbers by one.**

**RC1 – 36:** P14, Sect 5.3: Did you average over all depths or use surface values? Also, I did not understand the first sentence of the Section (in which figure is this shown?): for which decade is Fig. 6 made? In Fig.6, instead of a mean vs. biome wouldn't a mean versus latitude plot contain more information as it then is not discretized into these biome intervals? Or do you need to define zones because of low spatial data coverage?

**Reply RC1 – 36:** We used the two uppermost layer (down to 130 m) and the time-independent UVic grid. All results are presented in Fig. 7 (former 6). We clarified this. We chose the biomes for this presentation to directly compare regions with mostly consistent biological and ecological properties. Of course the coverage is a factor and we are able to compare and present our desired regions well in this way.

**Changes for RC1 – 36:** We reformulated II. 375-377: "We show the north-south trend of  $\delta^{13}C_{POC}$  over the Atlantic Ocean based on the time-independent UVic grid and restricted to the uppermost 130 m, which resemble the euphotic zone in the UVic model."

**RC1 – 37:** P16, I291: Maybe help the reader by stating what that means for seasonal availability for each hemisphere?

Reply RC1 - 37: We clarified this.

**Changes for RC1 – 37:** We reformulated I.394: "[...] only winter months on both hemispheres exhibit less data."

RC1 - 38: P17, I 302: define 'enough' in 'enough datapoints' (also in the caption of Fig. 8).

**Reply RC1 – 38:** Mathematically, a KDE can be calculated from a single data point (which would just be a single kernel). But the fewer (or the narrower the range of the) data points are available, the narrower, steeper and larger (in terms of y-axis values) the KDE becomes. For July, November and December on the northern hemisphere there were only 3 or less data points available, which caused a KDE by magnitudes larger than the others. Hence, we decided to exclude them from the plot for better comparability of the others. None of these restrictions were applied in the calculation of means. We clarified this.

**Changes for RC1 – 38:** We reformulated II. 407-410"[...] displayed all months with enough data points by a KDE and indicate same months by same colors. We excluded July, November and December on the northern hemisphere from this KDE representation, because these provided

three or less data points within them, which resulted in a KDE that overgrew the others by magnitudes and made their visual comparison difficult."

**and in the caption of Fig. 9 (former 8): "[...] enough available data points (here more than three) a Gaussian KDE approximate their density."**

**Also, we added the used numbers of data points here (see Reply RC1 - 30).**

**RC1 – 39:** P19, Fig. 8: In the caption you write that b and d are means, but in the title and text (p17, I303) it says median – what is it? I think connecting the mean/median values with a line is a bit confusing especially in d – why not present the values in a small table?

**Reply RC1 - 39:** It is the median, as always. We discarded panels (b) and (d) and added a 2-rows table presenting the medians instead.

**Changes for RC1 – 39:** In Figure 9 (former 8) we discarded panels (b) and (d) and added the numbers of data points (see **Reply RC1 – 30**).

We changed the caption from Figure 9 (former 8): "Monthly variations are split up by hemisphere in the northern in (a) and southern in (b). Due to their best data coverage, the analyses are carried out within the 1990s and in the uppermost 130 m. The  $\delta$ 13CPOC is split up by sample month and for every month with enough available data points (here more than three) a Gaussian KDE approximate their density. The used data points are given in each KDE label. For each hemisphere the densities are drawn all together, each month indicated by an individual color."

Table 5 is new and presents the median change on both hemispheres.

We changed mean to median in II. 11, 13, 452-453

RC1 - 40: P20, I352: not specific enough, why not specify which areas?

**Reply RC1 - 40:** We changed this part.

Changes for RC1 - 40: We changed II. 483-484 to: "[...], especially the Atlantic and Indian Ocean."

RC1 - 41: P1, I2: They have for example been used to?

Reply RC1 - 41: We changed the sentence.

Changes for RC1 - 41: We added in I. 2: "for example"

RC1 - 42: P1, I16: via its atmospheric form

**Reply RC1 - 42: We changed the sentence.**

**Changes for RC1 – 42:** We changed I. 16 to: "[...] regulating climate via its atmospheric form [...]" **RC1 – 43:** P2,  $| 40: ...- 1 ) \cdot 1000$  and remove the '.' at the end

Reply RC1 - 43: We will remove the full stop.

Changes for RC1 - 43: We removed the full stop in I. 41.

RC1 - 44: P2, I55: reference?

**Reply RC1 – 44:** We added an additional references to this paragraph and we changed this sentence accordingly.

**Changes for RC1 – 44**: We changed I. 61 to: "Atmospheric  $\delta^{13}C_{co2}$  has decreased from -6.5 ‰ in preindustrial times to -8.4 ‰ presently (Rubino et al., 2013)."

We added the reference: Rubino, M., Etheridge, D. M., Trudinger, C. M., Allison, C. E., Battle, M. O., Langenfelds, R. L., Steele, L. P., Curran, M., Bender, M., White, J. W. C., Jenk, T. M., Blunier, T., and Francey, R. J.: A revised 1000 year atmospheric  $\delta$ 13C-CO2record from Law Dome and South Pole, Antarctica, Journal of Geophysical Research: Atmospheres, 118, 8482–8499, https://doi.org/10.1002/jgrd.50668, 2013.

RC1 - 45: P3, I69: improve our understanding of marine carbon

**Reply RC1 - 45:** We changed the sentence.

Changes for RC1 - 45: we changed I. 83 to: "[...] improve our understanding of marine carbon [...]"

**RC1 – 46:** P4, I11: and KH13.

Reply RC1 - 46: We changed the sentence.

Changes for RC1 - 46: We changed I. 132 to: "NECTALIS 3 and 4 and KH13"

RC1 - 47: P4, I114: Table 3 referred to before table 1 and 2?

Reply RC1 - 47: This will be changed.

Changes for RC1 - 47: We excluded the reference to Table 3 from I. 139.

We added in I. 146: "The complete structure is presented in Table 3."

**RC1 - 48:** P5, I140: remover the '.' in front of (Verwega et al., 2021).

Reply RC1 - 48: We will remove the full stop.

Changes for RC1 - 48: We removed the full stop in I. 174.

RC1 - 49: P9, 1190: refer to Figure 1?

Reply RC1 - 49: We changed the sentence.

**Changes for RC1 - 49**: We changed I. 249 to: "[...] by Gaussian kernel density estimation (KDE) in Figure 1."

**RC1 - 50:** P9, I194: refer to Figure?

**Reply RC1 - 50:** We changed the sentence.

**Changes for RC1 – 50**: We changed I. 253 to: "[...] different measurement method applied (Figure 3)."

**RC1 – 51:** P9, I201: 'what indicates very little data points lying' should be 'which indicates that very few data points lie'

**Reply RC1 - 51:** We changed the sentence.

**Changes for RC1 – 51**: We changed II. 260-261 to: "[...] which indicates very little data points lie in this range."

RC1 - 52: P16, I 285-287: change /permil to ‰

**Reply RC1 - 52:** We formatted the sentence.

**Changes for RC1 – 52**: We changed II. 388-389 to: "[...] from -28 to -29 ‰ are those two located farthest south. The biome located farthest north contain the next lowest value at about -24 ‰."

RC1 - 53: P18, I310: 'highest maximum' seems double, reformulate?

Reply RC1 - 53: We excluded this formulation.

**Changes for RC1 – 53**: We reformulated II. 416-424 to: "Similarly high median  $\delta^{13}C_{POC}$  values cannot be ascertained for any month with data of the southern hemisphere, where values of  $\delta^{13}C_{POC}$ above –20‰ have rarely been observed at any time of the year. In fact, there is an overall tendency towards low  $\delta^{13}C_{POC}$  values for the southern hemisphere, which becomes well expressed during the months April and September, with medians of  $\delta^{13}C_{POC} = -28.1\%$  and  $\delta^{13}C_{POC} = -28.5\%$ respectively."

**RC1 – 54:** P20, I344: change to 'two different global grids', the word resolution here reads not well in this position. Or rephrase.

Reply RC1 - 54: We changed the sentence.

Changes for RC1 - 54: we reformulated I. 474 to: "[...] two different global grids [...]"

RC1 - 55: P20, I345: relative to their mean? As an anomaly to their mean?

**Reply RC1 – 55:** In agreement with the updated structure of the csv-file (see **Reply RC2 – 3**) and we changed this part, accordingly.

**Changes for RC1 – 54**: We changed I. 475 to: "[...] the  $\delta^{13}C_{POC}$ , their anomalies to their mean and all available [...]"

RC1 - 56: P23, I383-386: 2003a and 2003b instead of 2003an and 2003av?

Reply RC1 - 56: We changed the references to be indicated by their years.

We changed the following references:

Altabet, M. A. and Francois, R.: Natural nitrogen and carbon stable isotopic composition in surface water at cruise NBP96-05, https://doi.org/10.1594/PANGAEA.128266, 2003a.

Altabet, M. A. and Francois, R.: Natural nitrogen and carbon stable isotopic composition of station NBP96-05-06-4, https://doi.org/10.1594/PANGAEA.128229, 2003b.

**RC1 – 57:** Only after opening the dataset it became clear to me that the decadal files have all data of that decade saved together in one file (right? Or is the mean for each location?). The poc13\_univ\_i1.nc file was also unexpected based on the text, and seems to contain all data merged over time, but not over depth. This relates to my comment on P5, l141 – tell the readers how many files they will find and what they contain.

**Reply RC1 – 57:** These aspects of the data files were briefly described next to the data files on the PANGAEA data site. We have improved their description in the text and on the data site so these aspects are clearer.**

**RC1 – 58:** The TANN dimension has no description (ie netcdf attributes) but 'TANN:axis = "T" ', please add more information: The netcdf files state that the files follow the CF-1.6 convention. However, when checking this (https://pumatest.nerc.ac.uk/cgi-bin/cf- checker.pl), this seems not to be true for TANN. There is also TYR in the poc13\_year\_month\_woa\_c1.0.nc file with the same issue. The dimension 'record' in the last file also confused me.

**Reply RC1 – 58:** We have updated the WOA data files so they are easily accessible with a 4D variable and a properly defined the time axis.**

**RC1 – 59:** I think the naming of the x and y axis (which represent lon and lat) is not very intuitive or standard – why not use lon or long and lat?

**Reply RC1 - 59:** We have updated the data files with these improvements for axes names.**

**RC1 – 60:** Global&variable attributes: If your files end up being saved on a computer somewhere, based on the file name one would not be able to know what they exactly are, who made them, how they were made, etc. I think the absence of this information in the files themselves is important and should be addressed. Try to describe them in such detail that if none of the authors can be contacted for extra information and help, that the dataset is still usable. E.g., in the global attributes, include references to all sources (Table 1 and 2), reference to this article, a title, datetime\_start, datetime\_stop, size, contact, license (CC-BY 4.0?), method/how the dataset was made, description, and any other attributes of potential use. For all attributes and dimensions make sure to give them at least where applicable units, long\_name, standard\_name, \_FillValue, missing\_value, description. Also, DEPTH has a strange missing\_value.

**Reply RC1 – 60:** We have updated the attributes in the data files to include this additional information.**

**RC1 – 61:** In the poc13\_year\_month\_woa\_c1.0.nc file, is dimension 'record' the month and TYR the year? Why are these not in one time dimension? A 5 dimensional variable, which it is now, is difficult to work with (for example, CDO can't do it).

**Reply RC1 – 61:** We have updated the data files with the finer WOA grid. We now provide one annual and 12 monthly files, each of which containing a 4-D variable containing a time axis with a yearly increment. We have verified that CDO can operate on these updated data files.**

**Comments by RC2**

**RC2** – 1: The authors present a new compilation of global measurements of POC  $\delta$ 13C values and a description and overview of the data set. This is an important goal: such a data set is currently lacking and could be an important tool for observing temporal changes, validating current ocean biogeochemical models that incorporate  $\delta$ 13C-POC, and generally exploring ocean carbon dynamics in the particulate phase.

**Reply RC2 – 1:** We appreciate the reviewer's general comment, pointing out the importance of $\delta^{13}C_{POC}$ data collections. The reviewer's comments were addressed point by point, as listed below.**

**RC2 – 2:** After downloading and considering the data set in addition to the summary manuscript, this effort leaves some questions. The creation of such a data set should be forward- thinking and demonstrate a clear vision for how it will grow. Some improvements are needed for this data set to be truly useful and forward-thinking.

**Reply RC2 – 2:** We have updated the csv file as well as the NetCDF files (see **Reply RC2 – 3, Reply RC1 – 7, Reply RC1 – 57-61**). The contact author of this paper (Christopher Somes) is planning to provide annual updates of the data set. We will include this vision in the conclusion part of the manuscript as presented in **Reply RC2 – 6**.

**RC2 – 3:** Presentation of data as anomalies from a mean does not seem logical. As more data are added, the mean will change; thus, the data set needs to be presented as actual values.

**Reply RC2 – 3:** The insertion of anomalies instead of the original values turned out to be a misunderstanding between us and the data management of PANGAEA, which could eventually be clarified and has now been corrected. Data submissions at PANGAEA are usually original data, measured and analyzed by their authors. When these data are used, the data sets can and have to be cited. For a compilation of data, it is important not to get duplicate data into the PANGAEA database. However, if the collected data are used to derive statistical values, it is possible to provide these values in an additional file. The file now contains both, the anomalies as well as the original values.

**Changes for RC2 – 3:** We revised the raw data file, which includes now the raw measurements along with the anomalies.

We added in I. 97: "and their".

We changed I. 175: "[...] d13CPOC measurements, their anomalies [...]".

We added in I. 190: "measurements".

We added a line in Table 3 for the measurements.

**RC2 – 4:** In addition, it is unclear why anomalies are reported to so many decimal places. The original data are likely all reported to only one decimal place, possibly two, which is the maximum practical precision of typical isotope ratio instrumentation. The mean itself and the anomalies should not be presented to a precision exceeding 1-2 decimal places, depending on calculation of uncertainty (see detailed comments below).

**Reply RC2 – 4:** This escaped our attention and we are thankful for this comment. We revised the csv file accordingly and have decreased the decimal precision to 2.

Changes for RC2 - 4: We added to I. 192: "roundedn to two digits after the floating point"

We changed equation 3 to "[...] = -23.96 [...]"

We decreased the precision of the anomalies range in Table 3 to [-31.19, 19.46]

**RC2 – 5:** Many currently available sources of data are not included. I refer the authors to Close and Henderson (2020) for one example of a different list of publications containing oceanic POC  $\delta$ 13C data, which they incorporated in a recent depth-resolved global POC  $\delta$ 13C assessment, as well as a list of publications which they did not include for reasons specific to their assessment. They included more than 300 data points from Pedrosa- Pamies et al. 2018; Bishop et al. 1977; Jeffrey et al. 1983; Hurley et al. 2019; O'Leary et al. 2001; Trull et al. 2008; Saino 1992; Minagawa et al. 2001; Hernes and Benner 2002; Druffel et al. 2003. Additional data sources they did not include but listed and collectively contain hundreds more data points: Williams and Gordon 1970; Eadie and Jeffrey 1973; Druffel et al. 1996; Benner et al. 1997; Trull and Armand 2001; Hernes and Benner 2006; Close et al. 2014; Krishna et al. 2018; Liu et al. 2018; Griffith et al. 2012; Xiang and Lam 2020. Most of these data sources are not included in the data set presented here, and most of them are not archived in PANGAEA.

**Reply RC2 - 5:** We understand the reviewer's concern. The data collection of Close and Henderson (2020) was made available at a time after we had already collected, sorted and analyzed the 4732 data points presented herein. It is regrettable to have missed this opportunity to add these 303 data points, but additional credit should be given to the data collection provided by Close and Henderson (2020), which we now want to point out explicitly (see added changes below, in agreement with **Reply CC1 - 68**). We are explicit about all our data sources and we clarify in the text that our data collection is neither exclusive nor all-embracing. Our goal was to come up with a global data set that is informative and has sufficient potential for meaningful data-model comparisons on global scale. We added a paragraph at the end of section 2.1 and we changed the first paragraphs of the discussion. We compared the global mean of the now available data collection of Close and Henderson (2020) with our data collection. The bias (difference between global means) is 0.3 ‰.

**Changes for RC2 – 5**: We added II. 133-136: "A recent collection of 303 measurements of  $\delta^{13}C_{POC}$  has been provided by Close and Henderson (2020), largely based on data gathered from individual publications referenced therein. Since our analyses originally relied on data sources that differed from those of Close and Henderson (2020) we find our collection to be yet incomplete. Especially measurements from national databases might provide a huge future benefit."

We reformulated II. 462-473: "[...] The starting point of our collection and analyses was the readily available data collection of Goericke (1994), which comprised 467 data points. Our primary objective was to elaborate this set of data by adding useful meta-information from the original publications and by introducing additional  $d^{13}C_{POC}$  measurements, as recorded in the world ocean data base PANGAEA and made available by Robyn Tuerena and Anne Lorrain. This way we could expand the data collection substantially, from the original 467 to 4732 data points. This new  $\delta^{13}C_{POC}$ data set provides the best coverage to date that will be a useful tool to help constrain many marine carbon cycling processes and pathways from ocean-atmosphere exchange to marine ecosystems, as well as to better understand observations and validate models. To ensure a dynamic growth of our data collection the corresponding author will provide annual updates of the data set.

**Furthermore, he may be contacted by any interested researcher, who would like to add their data to this collection."**

**RC2 – 6:** There are currently many different databases currently used for isotope data in the oceans, particularly across different national research agencies such as BCO-DMO in the U.S., National Institute of Oceanography in India, JAMSTEC's time-series data sets, Japan Oceanographic Data Center, etc.

**Reply RC2 – 6:** We attempted to rely on a world data base that archives data provided by many research groups internationally. We have to admit that we do not know how many data points were missed because we did not stretch our search to national data collections. We decided to add information about additional measurements that can be derived from different data sources and how we plan to account for them in the future (see **Reply RC2 – 5**).**

**RC2 – 7:** Importantly, the current manuscript mentions a lack of data from the northern Pacific, but it has missed some publications containing such data, perhaps because European/Atlantic research results are disproportionately represented in PANGAEA.

**Reply RC2 – 7:** In our opinion, it is unclear whether a bias in data archiving exists or how severe such bias is. Since we do not know, we refined our statement in the Conclusion.

**Changes for RC2 – 7**: We reformulated II. 490-491: "[...], but observed a lack of data in PANGAEA that cover northern Pacific region."**

**RC2 – 8:** Seeing as this initial data set is missing much existing data, how do the authors propose to keep the data set up to date as new data are produced and published, or not published but instead entered into other databases? In addition to the publications listed above, there are new data sources since 2020, such as from the Arabian Sea by Silori et al., from the Arctic by Xiang and Lam, and South China Sea by Yang et al.

**Reply RC2 – 8:** The contact author of this paper (Christopher Somes) is planning to provide annual updates of the data set. Furthermore, he may be contacted by interested researchers who would like to incorporate their data in the data set (see **Reply RC2 – 6**).**

**RC2 – 9:** What is the reasoning behind including data without 3-dimensional spatial coordinates (latitude, longitude, depth)? 128 data points lack lat/lon data, and 837 data points lack depth data. Of what use are data points without 3-dimensional spatial information? There is an opportunity missed here to also include details of analytical methods that would serve as a quality control measure. Namely, did the original data sources describe acidifying the samples (i.e., can we be sure the data are POC rather than total PC?), using what acidification technique, and did they include a blank correction? Older data may have been produced using closed-tube combustion/dual inlet IRMS, whereas newer data were likely produced using EA-IRMS. Because sampling method is included, the lack of analytical method is notable.

**Reply RC2 – 9:** Our goal was to build upon the previously compiled data sets from Goericke and Young. Since the data files from these compilations included data with missing spatial values, we did not exclude them in case it may be possible to retrieve this additional information in the

future. We did not receive a reply when contacting the corresponding authors from these respective papers. However, we did not want to exclude these data from our study.

Since analytical errors are substantially smaller than the standard deviation of observations, and thus do not likely represent a significant source of uncertainty and variability in the observations, we did not document the analytical methodology for each of the over 4,000 observations. We thank the reviewer for raising this important issue. Unfortunately, it is not feasible for us to undergo such a time-intensive task within journal deadlines in this review process. We will incorporate this additional analytical information in future updates of the database.

**RC2** – **10:** Defining POC: For some of the sampling methods, a size fraction is not specified. For instance, in situ pump and MULVFS samples are often size fractionated, but there is no data field specified here for size fraction. Similarly, there are zooplankton net results in the current data set. Do the contents of a zooplankton net belong in a data set of POC? Many of the other collection methods listed here exclude zooplankton as components of passive POC, such as pre-screening of sediment trap samples through a 250-350 micron mesh to exclude zooplankton.

**Reply RC2 – 10:** Since there are many potential applications of this data set, we decided to include all data defined as POC by the original publication in PANGAEA. Note that we have provided the publication reference link for the observations in the CSV data file along with sampling methods, so this information is easily accessible for researchers specifically concerned with this issue. We decided not to exclude any data from our search in case it might be relevant for more specific analysis and applications of this data. Both the Goericke and Young data sets also included zooplankton in their analysis, and noted individual studies examining the difference between  $\delta^{13}$ C in POC and zooplankton was not significant (Brodie et al., 2011; Lorrain et al., 2003; McConnaughey and McRoy, 1979; Voß, 1991; Schell 1992 (personal communication to Goericke 1994)).

RC2 - 13: What is "biological sample"? This does not sound like POC.

**Reply RC2 – 13:** This is the description from the obtained data file, noting that the variable name was POC. We prefer to refrain from including lengthy descriptions in the data file, and note that the reference and link to each original paper is included in the CSV data file.

RC2 - 13: How are CTD/rosette, CTD, bottle, and Niskin bottle different methods?

We clustered all of these as "bottle" data, since we assume they have been measured under most similar conditions. The above given different namings are those given by the original publishers of the data, which we did not want to change in the detailed data description given along in the csvfile. We clarified this in the Sampling methods section.

Changes for RC2 - 13: We changed II. 285-286: "Bottle" data include samples taken from Niskin bottles, PEP bottles, CTDs and samples collected via Seabird submersible pumps."

RC2 - 14: Introduction: contains unsupported claims: please include references.

**Reply RC2 – 14:** In agreement with the other reviewers we added the following references throughout the introduction:

Volk, T., and M. I. Hoffert, Ocean carbon pumps: Analysis of relative strengths and efficiencies in ocean-driven atmospheric CO2 changes, in The Carbon Cycle and Atmospheric CO2: Natural Variations Archean to Present, The Carbon Cycle and Atmospheric CO2: Natural Variations Archean to Present, Geophys. Monogr. Ser., vol. 32, edited by E. T. Sundquist, and W. S. Broecker, pp. 99–110, AGU, Washington, D. C., 1985. (see **Reply RC2 – 17**)

Banse, K.: New views on the degradation and disposition of organic particles as collected by sediment traps in the open sea, Deep Sea Research Part A. Oceanographic Research Papers, 37, 1177– 1195, https://doi.org/10.1016/0198-0149(90)90058-4, https://www.sciencedirect.com/science/article/pii/0198014990900584, 1990. (see **Reply RC2 – 17**) Zeebe, R. E. and Wolf-Gladrow, D.: CO2 in Seawater: Equilibrium, Kinetics, Isotopes, Elsevier Science B.V., Elsevier Oceanography Series, 65, 2001. (see **Reply RC1 – 2**)

Eide, M., Olsen, A., Ninnemann, U. S., and Johannessen, T.: A global ocean climatology of preindustrial and modern ocean  $\delta$ 13C, Global Biogeochemical Cycles, 31, 515–534, https://doi.org/10.1002/2016gb005473, 2017. (see **Reply RC1 – 12**)

Levin, I., Schuchard, J., Kromer, B., and Münnich, K. O.: The Continental European Suess Effect, Radiocarbon, 31, 431–440, https://doi.org/10.1017/s0033822200012017, 1989. (see **Reply RC1 – 12**)

Ndeye, M., Sène, M., Diop, D., and Saliège, J.-F.: Anthropogenic CO2 in the Dakar (Senegal) Urban Area Deduced from 14C Concentration in Tree Leaves, Radiocarbon, 59, 1009–1019, https://doi.org/10.1017/rdc.2017.48, 2017. (see **Reply RC1 – 12**)

Tjiputra, J. F., Schwinger, J., Bentsen, M., Morée, A. L., Gao, S., Bethke, I., Heinze, C., Goris, N., Gupta, A., He, Y.-C., Olivié, D., Seland, Ø., and Schulz, M.: Ocean biogeochemistry in the Norwegian Earth System Model version 2 (NorESM2), Geoscientific Model Development, 13, 2393–2431, https://doi.org/10.5194/gmd-13-2393-2020, 2020. (see **Reply RC1 – 15**)

Liu, B., Six, K. D., and Ilyina, T.: Incorporating the stable carbon isotope 13C in the ocean biogeochemical component of the Max Planck Institute Earth System Model, https://doi.org/10.5194/bg-2021-32, 2021. (see **Reply RC1 - 15**)

Rubino, M., Etheridge, D. M., Trudinger, C. M., Allison, C. E., Battle, M. O., Langenfelds, R. L., Steele, L. P., Curran, M., Bender, M., White, J. W. C., Jenk, T. M., Blunier, T., and Francey, R. J.: A revised 1000 year atmospheric  $\delta$ 13C-CO2record from Law Dome and South Pole, Antarctica, Journal of Geophysical Research: Atmospheres, 118, 8482–8499, https://doi.org/10.1002/jgrd.50668, 2013. (see **Reply RC1 - 44**)

McConnaughey, T., and C. P. McRoy (1979), Food-web structure and the fractionation of carbon isotopes in the Bering sea, Mar. Biol., 53(3), 257–262. (see **Reply RC1 – 10**)

Gruber, N., Keeling, C. D., Bacastow, R. B., Guenther, P. R., Lueker, T. J., Wahlen, M., Meijer, H. A. J., Mook, W. G., and Stocker, T. F.: Spa- tiotemporal patterns of carbon-13 in the global surface oceans and the oceanic suess effect, Global Biogeochemical Cycles, 13, 307–335, https://doi.org/ https://doi.org/10.1029/1999GB900019, https://agupubs.onlinelibrary.wiley.com/doi/abs/ 10.1029/1999GB900019, 1999. (see **Reply RC1 – 11**)

Schmittner, A., Gruber, N., C, A., Key, M. R. M., Tagliabue, A., and Westberry, T. K.: Biology and airsea gas exchange controls on the distribution of carbon isotope ratios (13C) in the ocean, Biogeosciences, pp. 5793–5816, https://doi.org/10.5194/bg-10-5793-2013, 2013. (see **Reply RC1 – 11**)

Zhang, J., Quay, P., and Wilbur, D.: Carbon isotope fractionation during gas-water exchange and dissolution of CO2, Geochimica et Cosmochimica Acta, 59, 107–114, https:// doi.org/10.1016/0016-7037(95)91550-d, https://doi.org/10.1016/0016-7037(95)91550-d, 1995. (see **Reply RC1 - 13**) RC2 - 15: Line 10: The reason for the focus on the Atlantic should be clarified (data coverage)

**Reply RC2 - 15:** We included the superior data coverage of the Atlantic.**

**Changes for RC2 – 15**: We changed II. 10-11: "An analysis of 1990s median  $\delta^{13}C_{POC}$  values in an meridional section across the best covered Atlantic Ocean shows [...]"

**RC2 – 16:** Line 30: awkward phrase. Maybe "some particulate organic carbon" or "parts of the particulate organic carbon pool"

**Reply RC2 - 16: We rephrased this part.**

Changes for RC2 - 16: We changed I. 30: "Some particulate organic carbon (POC) sinks [...]"

RC2 - 17: Line 32: please use an earlier reference for the soft tissue pump

**Reply RC2 – 17:** We exchanged the reference with Volk and Hoffert (1985) and Banse (1990) (see **Reply RC2 – 14**).

**Changes for RC2 - 17: The references were exchanged in I. 33.**

**RC2 – 18:** Line 40: omit the factor of 1000 to adhere to generally accepted d13C terminology (see Coplen TB. 2011. Guidelines and recommended terms for expression of stable-isotope- ratio and gas-ratio measurement results. Rapid Commun Mass Spectrom. 25: 2538-2560).

**Reply RC2 - 18:** We did this in the next manuscript version.

Changes for RC2 - 18: We omitted the factor 1000 in l. 41.

RC2 - 19: Lines 108 and 110: repetitive about referring to this as the Tuerena dataset.

Reply RC2 - 19: We excluded the second mentioning.

**Changes for RC2 – 19**: We changed II. 128-130: "[...] to which we will refer to as the Tuerena data set. This contains 595 data points including 501 from Young et al. (2013) and covers samples within the euphotic zone and an observation timeframe of 1964 - 2012."

RC2 - 20: Line 115: this sentence about depth intervals & timeframes is confusing

Reply RC2 - 20: We rephrased it.

**Changes for RC2 – 20**: We rephrased II. 139-142: "Spatial coordinates originally given as depth intervals were replaced by their respective mid points. Time intervals were not changed in this way. If they contained just one month or year this was taken, otherwise the time information was omitted."

RC2 - 21: Line 116: Longitude values were converted

Reply RC2 - 21: We changed this.

Changes for RC2 - 21: We changed I. 143 to: "Longitude values were converted to [...]"

RC2 - 22: Line 123-124: The last part of this sentence is awkward

Reply RC2 - 22: We rephrased the sentence.

**Changes for RC2 – 22**: We changed II. 151-153: "According to Table 3 we complemented the data with month, year, depth, sample method, cruise, trap duration and a references, wherever available."

RC2 - 23: Line 125: Need to elaborate on why some values were considered suspicious.

Reply RC2 - 23: We clarified this.

**Changes for RC2 – 23**: We changed II. 154-155: "Rounded values were adjusted to their source values as well as data with interchanged longitudinal information, which is in detail shown in Table 1."

RC2 - 24: Line 127: Run on sentence (comma splice)

**Reply RC2 – 24:** We discarded this paragraph, since all information is discussed before (see **Reply RC2 – 20**) and move the sentence "Sample depth given as "surface" was denoted as 1 m." there, too.

Changes for RC2 - 24: We added in I. 143: "Sample depth given as "surface" was denoted as 1 m."

The II. 156-157 were discarded.

See also Changes for RC2 - 20.

**RC2 – 25:** Line 127-128: Confusing sentence. An example would be helpful. What was done if this criteria was not met?

Reply RC2 - 25: We decided to discard this paragraph (see Reply RC2 - 24).

Changes for RC2 - 25: The II. 156-157 were discarded.

**RC2 – 26:** Lines 135-137: This information might fit better where longitude transformation was first discussed.

**Reply RC2 - 26:** We will shorten this part and move the equation.

**Changes for RC2 – 26**: We shortened II. 167-170: "[...] Longitude was converted to [–180°, 180°] from a [0°, 360°] format by Equation 2"

We added in II. 142-144: "[...] Longitude values were converted to the format [-180°,180°] by the transformation [Equation 2]"

RC2 - 27: Lines 142-143: How did these datasets & interpolations address depth?

Reply RC2 - 27: We described the depth levels of the two grids in their introduction.

Changes for RC2 – 27: We changed II. 177-178: "[...] a 1.8° × 3.6° -resolution and 19 depth layers

from a model that simulates  $\delta^{13}C_{POC}$  (e.g. Schmittner and Somes, 2016), in the following referred to as the UVic grid, and the 1°×1°-resolution and 102 depth layer grid of the World Ocean

Atlas (Garcia et al., 2018) [...]"

See also Reply RC1 - 3 or Reply CC1 - 23.

**RC2 – 28:** Equation 3: A mean should not be this much more precise than the precision of the individual measurements. The analytical uncertainty in individual measurements is likely between 0.05-0.2 per mil, and the original data were likely presented to a precision of 1-2 decimal places (per mil notation). The propagated uncertainty in the mean would likely be somewhere around 0.1-0.3 per mil. Therefore, the precision of the mean likely should not exceed 1-2 decimal places, purely from a standpoint of analytical realities.

**Reply RC2 – 28:** After re-clarification with the PANGAEA editor, we are now allowed to include the measurement values obtained in the data files so we don not have to rely exclusively on including these anomalies relative to the mean. See also **Reply RC2 – 4**.

RC2 - 29: Line 153: unnecessary second comma

Changes for RC2 - 29: Omitted in I. 199.

RC2 - 30: Line 154: Meaning of first sentence is unclear

Reply RC2 - 30: We discarded this sentence. Details are explained in the following paragraph.

Changes for RC2 - 30: We discarded the respective sentence in I. 200.

**RC2 – 31:** Line 172: The depth of the euphotic zone varies by location, so this statement is not wholly accurate.

Reply RC2 - 31: We changed this sentence.

**Changes for RC2 – 31**: We changed II. 222-223: "The two uppermost layers reach down to depths of 50 and 130 m respectively, and they are supposed to comprise the upper ocean's euphotic zone."

**RC2 – 32:** Lines 202-205: would be useful to mention at what depths these outlier data points were collected. Is there any context to these locations or depths that would hint at the reason for these low values?

**Reply RC2 – 32:** Yes, most of them are located at great depths in the vicinity of hydrothermal fields. Only one data point is from the ocean surface. We added this (in agreement with **Reply CC1 – 31**).

**Changes for RC2 – 32**: We changed II. 263-267: "[...] taken from Lein and Ivanov (2009) and Lein et al. (2006), measured in September or October 2003, around the location 10° N, 104° W and below 2500 m depth in the vicinity of a hydrothermal field close to the Pacific coast of middle America. The lowest outlier at  $\delta^{13}C_{POC} = -55.15\%$  is taken from Altabet and Francois (2003a) from November 1996 and at 62.52° S, 169.99° E at the ocean surface south from New Zealand."

**RC2 – 33:** Line 206-209: Values higher than -10 per mil at depths between 636-901 m are very strange for a station where the total water depth is around 3000 m (this appears to be Line P, somewhere near Station 11-13). This may be a case where checking the acidification method and/ or contacting the authors would be appropriate.

**Reply RC2 – 33:** Although we agree this seems strange, as stated above, we decided not to exclude any already published observations found in our search to remain as consistent and objective as possible.

RC2 - 34: Line 242: extra "second"

**Reply RC2 - 34: According to (Reply RC1 - 28) we rephrased the whole paragraph.**

Changes for RC2 - 34: See Changes for RC1 - 28.

RC2 - 35: Line 263-264: Not a full sentence; could combine with previous.

Reply RC2 - 35: We combined them.

**Changes for RC2 – 35**: We changed I. 361: "Many cruises are visible as lines formed by connected grid cells in Figure 5, especially in the Atlantic and [...]."

**RC2 – 36:** Line 267: Confusing sentence: discussing high values but mentioning explicitly low values, <30 per mil.

Reply RC2 - 36: We changed the sentence.

**Changes for RC2 – 36**: We changed I. 366: "Highest  $\delta^{13}C_{POC}$  values are evident in low latitude regions."

RC2 - 37: Line 283-284: Difficult to understand this sentence

**Reply RC2 – 37:** In agreement with another reviewer's point (**CC1 – 58**), we changed the end of this paragraph.

**Changes for RC2 – 37**: We changed II. 383-386"The biomes are numbered 9 to 17, excluding 14. The biomes 15 to 17 are representing parts of the Southern Ocean and were restricted to 70° W and 20° E. Their locations are shown in Figure 7."

RC2 - 38: Line 293: Figure 7

Reply RC2 - 38: We corrected this.

**Changes for RC2 – 38:** We changed in I. 396: "Figure 8" in agreement with the new numbering of Figures.

RC2 - 39: Section 6.1: Somewhat confusing. Grouping data by season may help.

**Reply RC2 – 39:** We resolved the data according to months and therefore corrected the title, now referring to "month" rather than "season". For coherence, we also changed all mentioning of "season" to "month" and rename the following section to "Decadal variations".

Changes for RC2 - 39: We changed I. 402 to: "6.1 Monthly variations".

We changed I. 403 to: "[...] monthly variations [...]".

We changed I. 430 to: "6.2 Decadal variations".

RC2 - 40: Lines 300-301: Specifying the percentage of data is in this decade would be clearer

Reply RC2 - 40: We changed the sentence.

**Changes for RC2 – 40**: We changed I. 403: "Since more than 50 % of the available  $\delta^{13}C_{POC}$  data [...]"

RC2 - 41: Lines 301-302: How many data points is enough? How do you test this?

**Reply RC2 – 41:** Mathematically, a KDE can be calculated from a single data point (which would just be a single kernel). But the fewer (or the narrower the range of the) data points are available, the narrower and steeper and larger (in terms of y-axis values) the KDE becomes. For July, November and December on the northern hemisphere there were only 3 or less data points available, which caused a KDE by magnitudes larger than the others. Hence, we decided to exclude them from the plot for better comparability of the others. None of these restrictions were applied in the calculation of means. We clarified this.

**Changes for RC2 – 41:** We changed II. 407-410: "[...] displayed all months with enough data points by a KDE and indicate same months by same colors. We excluded July, November and December on the northern hemisphere from this KDE representation, because these provided three or less data points within them, which resulted in a KDE that overgrew the others by magnitudes and made their visual comparison difficult."

We changed in the caption of Fig. 9: "[...] enough available data points (here more than three) a Gaussian KDE approximate [...]".

**RC2 – 42:** Line 304: Possible data coverage issue here. The Southern hemisphere could have lower values due to more data coverage in the high-latitude Southern Ocean compared to the high-latitude Arctic.

**Reply RC2 – 42:** This is a good point, which we have not thought of. We agree and think that this should be mentioned in the text. We added this thought to the Conclusion.

**Changes for RC2 – 42**: We added in II. 492-495: "[...], where we observed that lowest values (< $\approx$  –28 ‰) can be found in the Southern Ocean whereas highest (> $\approx$  –22 ‰) are restricted to low latitudinal regions. This might also have influenced the observed lower  $\delta^{13}C_{POC}$  values on the southern hemisphere compared to the northern, due to the relatively good coverage of the Southern Ocean."

See also Reply RC2 - 44.

RC2 - 43: Line 305: How is this determined? Doesn't July have high values?

Reply RC2 - 43: We rephrase the whole paragraph according to Reply RC2 - 44.

RC2 - 44: Line 306: Winter in which hemisphere?

**Reply RC2 - 44:** We discussed the point more precisely and rewrote the whole paragraph as.

**Changes for RC2 – 44:** We changed II. 412-429: "The monthly resolved variations of  $\delta^{13}C_{POC}$  do not reveal any significant seasonal pattern (Figure 9, see also Table A4 in the Appendix). In general we find highest  $\delta^{13}C_{POC}$  values in the northern hemisphere, with median  $\delta^{13}C_{POC} = -20.4 \%$  in April and  $\delta^{13}C_{POC} = -21.5 \%$  in October, which are typical months with enhanced primary production (northern hemisphere spring and autumn blooms). Similarly high median  $\delta^{13}C_{POC}$  values cannot be ascertained for any month with data of the southern hemisphere, where values of  $\delta^{13}C_{POC}$  above - 20 ‰ have rarely been observed at any time of the year. In fact, there is an overall tendency towards low  $\delta^{13}C_{POC}$  values for the southern hemisphere, which becomes well expressed during the months April and September, with medians of  $\delta^{13}C_{POC} = -28.1 \%$  and  $\delta^{13}C_{POC} = -28.5 \%$  respectively.

However, interpretations of this north-south trend should be treated with caution, because the apparent tendency is likely conditioned by some imbalance in the number of high-latitudinal data points. Compared to the number of data points from the Southern Ocean, samples from the Arctic Ocean are considerably underrepresented (see also Figure 10). Furthermore, the discrimination between data of the northern and southern hemisphere is crude and we encourage to use our data collection for more advanced analyses of seasonal, monthly-based, changes in the  $\delta^{13}C_{POC}$  signal."

RC2 - 45: Line 309: Lowest what?

**Reply RC2 – 45:** We mean the lowest shown measurement values. We hope this is clearer by the changes proposed in **Changes for RC2 – 44**.**

**RC2 – 46:** Line 316: The phrase about the data points in 1960s and 1980s suggests that there might be similar numbers of data points; however, the 1980s have much more data but the spatial coverage is similarly poor.

Reply RC2 - 46: We clarified this.

**Changes for RC2 - 46**: We changed I. 436: "1980s are similarly sparse in spatial coverage as the 1960s."

RC2 - 47: Line 333: why is the median for the 2000s not shown on the figure?

**Reply RC2 – 47:** For the Southern Ocean there were no data available to calculate a 2000s median. In agreement with **Reply CC1 – 85** we discarded panels (b) and (d) in this figure and moved panel (b) to an own figure.

**Changes for RC2 – 47**: We discarded panels (b) and (d) in Figure 11 (former 10). The medians from panel (b) are shown separately in Figure 12.

RC2 - 48: Line 375: interest

Reply RC2 - 48: We corrected this.

Changes for RC2 - 48: We changed I. 513 to "[...] interest."

**RC2 – 49:** Table 1 caption: unnecessary comma in second sentence.

Changes for RC2 - 49: We excluded this.

RC2 - 50: Table 1 Eadie and Jeffrey entry under column "to": is this added from the source itself?

Reply RC2 - 50: Yes.

**Changes for RC2 – 50**: In agreement with **RC1 – 23** we repeated the cell contents even if they are repeated.

**RC2 – 51:** Table 2 footnote 2: how did you address this for sample durations *not* entirely within an explicit month and year?

Reply RC2 - 51: We clarified this in agreement with Reply RC2 - 20.

**Changes for RC2 – 51**: We changed II. 141-142: "Time intervals were not changed in this way. If they contained just one month or year this was taken, otherwise the time information was omitted."

We added to the footnote: "Only for sample durations entirely within an explicit month and year, otherwise information of time frames has been discarded."

RC2 - 52: Table 3: entry for depth under "coverage": typo in the total number of data points

Changes for RC2 - 52: We corrected this to: "3917 / 4732"

**Comments by CC1**

**CC1 – 1:** I would like to thank the authors for their thorough manuscript. I applaud them for their effort to compile such an extensive dataset of all available d13C POC measurements across the global ocean over multiple depth layers and monthly and decadal time-scales. Datasets as such are highly valuable and find application in a variety of research and technical fields and are useful to calibrate/validate process-based, mechanistic isotope-enabled models. The dataset and its components are clearly presented and so are major patterns in d13C POC values across space and time.

**Reply CC1 – 1:** We are happy about your appreciation of our work and effort. We will address your comments in the following point by point.**

**CC1 – 2:** It would be also interesting to see if there are any trends with depth, and how these may vary among ocean areas.

**Reply CC1 – 2:** Yes, we agree and also see the benefit of presenting the vertical distribution of the data. A vertical scatterplot of the data now included in our second submission of the manuscript as sub-panels of Figure 5 (former 4). This is split up into the main ocean basins (Arctic, Southern Ocean, North Pacific, South Pacific, ...) to visualize the distribution of the  $\delta^{13}C_{POC}$  data over the depth among the ocean areas.

**Changes for CC1 – 2:** We added three sub-panels to Figure 5 showing vertical scatterplots of  $\delta^{13}C_{POC}$  data over the depth among the main ocean basins. (b) includes data from the Southern, Indian and Arctic Ocean, (c) shows the Atlantic and (d) the Pacific Ocean. We added the caption: "The vertical distribution of available  $\delta^{13}CPOC$  samples is shown (a) as the approximated density of the measurement depths and (b-d) as measured  $\delta^{13}CPOC$  values relative to their respective measurement depth. (a) provides on the y-axis the estimated density of the depth values and on the x-axis the depth in m. The estimation was realized by a Gaussian KDE. (b) resolves the measurements of the Southern, Indian and Arctic Ocean, (c) the North and South Atlantic and (d) the North and South Pacific. The last three panels show on the y-axis the depth in m and on the x-axis the measured  $\delta^{13}CPOC$  value. Different colors are used to mark different ocean basins."

**CC1 – 3:** I provide a few – hopefully constructive – general and specific comments below and recommend this manuscript for publication after my comments are addressed or discussed.

**Reply CC1 - 3:** We appreciate this support very much.**

**CC1 – 4:** -data is plurar, datum is singular. So, please make sure to use plural forms when speaking about data

**Reply CC1 – 4:** We somehow missed to treat the plural form (of data) consistently throughout the entire manuscript and we will change this.

**Changes for CC1 – 4:** We changed this in Il. 99, 139-140, 145, 175, 176, 239, 270, 272, 291, 313, 397, 398, 400, 405, 438, 440, 474, 483, 496 and in the caption of Figure 6.

**CC1 – 5:** -be sure to use present/past tenses and active/passive forms consistently

**Reply CC1 – 5**: We had a detailed check through the manuscript before re-submission.

**Changes for CC1 – 5:** We corrected the language in II. 79, 153, 203, 205, 207, 208, 223-224, 234, 268-269, 272, 277, 279-280, 284, 290, 312, 326, 334, 336, 368, 396, 397, 433, 446, 448 and caption of Table 1,

**CC1 – 6:** -be consistent with terminology related to spatial distributions: coarse/fine grid/interpolation.

**Reply CC1 – 6:** We consequently used the namings "UVic" and "WOA" grid (see also **Reply CC1 – 23**).

**Changes for CC1 - 6:** We changed the notations in II. 212, 214, 224, 243, 244, 328, 335, 357, 431, 447.

**CC1 – 7:** Line 1: I think that the correct terminology here is just 'marine particulate organic carbon stable isotope ratios (d13C POC)', without -13 here, as by definition the isotope ratio is given by the ratio of the heavy (carbon-13) to the light (carbon-12) isotopes.

**Reply CC1 - 7: We changed this.**

**Changes for CC1 – 7**: We changed I. 1: "Marine particulate organic carbon stable isotope ratios [...]"**

**CC1 – 8:** Line 13: need commas for statement regarding the Southern Ocean: 'except for the Southern Ocean, which shows a weaker trend, but contains...'.

**Reply CC1 - 8: We changed this.**

**Changes for CC1 - 8**: We changed II. 13-14: "[...] Southern Ocean, which shows a weaker trend, but [...]"**

CC1 - 9: Line 16: Consider changing 'it is regulating' with 'it regulates'

**Reply CC1 - 9: We changed this.**

**Changes for CC1 - 9: We changed I. 16 to "[...] regulated via its climate [...]"**

**Reply CC1 - 10: We will changed this in agreement with Reply CC1 - 11.**

**Changes for CC1 – 10:** We changed II. 53-54: "[...] although uncertainties remain regarding the quantification of the specific processes and mechanisms [...]"**

**CC1 – 11:** Lines 47-48: I don't particularly agree with this statement as I think that the factors and processes underlying fractionation during photosynthesis are fairly well-understood, and fractionation can be predicted with confidence from [CO2aq], phytoplankton growth rate and community composition (due to different cell sizes and geometries of different taxa). Furthermore, these factors are all governed, to an extent, by temperature, which makes temperature a key predictor for fractionation and d13C POC patterns. Perhaps you should mention that here.

**Reply CC1 – 11:** We generally agree and did not intend to give the impression that these processes are poorly understood.**

**Changes for CC1 - 11: See Changes for CC1 - 10.**

**CC1 – 12:** Line 58: concentration of CO2aq is also temperature dependent, and so is the distribution of phytoplankton communities. That is, all factors that exert a direct influence on photosynthetic fractionation are ultimately controlled by temperature, which is a major control on fractionation during photosynthesis by phytoplankton. I would stress this a little more in the Introduction.

**Reply CC1 – 12:** We agree and added a statement temperature has a strong influence on almost all processes that directly govern fractionation.

**Changes for CC1 – 12:** We added II. 68-70: "We also stress that all of these processes are sensitive to temperature changes which adds additional complexity to understanding how fractionation may change in space and time."

**CC1 – 13:** Lines 59-61: again, I think we have a fairly good understanding of the processes causing variation in photosynthetic fractionation, but a dataset with extensive spatio-temporal coverage is certainly needed to investigate how trends change across space and time and the mechanisms underlying these changes as well as for calibrations/validations of process-based/mechanistic models with no data component.

**Reply CC1 – 13:** Here we have stated that the predicting spatial and temporal trends "remains a challenge" We think this is a fair statement given the many different processes listed above that can alter fractionation and have different sensitivities to environmental change.**

**CC1 – 14:** Line 65: please add citation Magozzi et al. 2017 Ecosphere here. This study models the C isotope fractionation during photosynthesis as a function of a suite of variables provided by the ocean biogeochemical model NEMO-MEDUSA and predicts spatial and temporal patterns in d13C POC across the global ocean over seasonal to decadal time-scales.

**Changes for CC1 - 14: We added this in I. 76.**

**CC1 – 15:** Line 66: calibration AND validation. A major issue associated with isotope-enabled biogeochemistry models for the global ocean is the lack of reliable validation datasets, with sufficient spatio-temporal coverage to allow proper validation (sometimes, datasets are so scarce or so scarcely comparable that it almost makes more sense to 'trust' the mechanistic model, based on fairly well-known and understood processes, rather than calibrating the model to the available data)

**Reply CC1 - 15: Definitely, we agree and added "validation" to this sentence.**

**Changes for CC1 – 15:** We added in I. 77: "[...] and validation of such process-based mechanistic models [...]"

CC1 - 16: Line 80: what does 'multilateral' mean?

Reply CC1 - 16: For clarification we rephrased (combine) two sentences.

**Changes for CC1 – 16:** We changed II. 93-96: "The meta-data comprise information about sampling location, time, depth and method as well as the original source, which makes original raw data values, method, and further technical description easily accessible."

**CC1 – 17:** Line 83: don't need a capital W for we after the semi-column. Please fix this here and throughout the text, as well as in figure captions.

Reply CC1 - 17: We fixed this in this line and the tables and figures captions, wherever applicable.

Changes for CC1 - 17: We fixed the spelling in II. 99, 235, caption of Table 1, 2, 3, 4, Figure 1.

**CC1 – 18: Line 89:** rephrase this as 'the adjustments that we conducted are described in the following sections' or something.

**Reply CC1 - 18:** We rephrased this sentence.**

**Changes for CC1 – 17**: We changed I. 107: "The adjustments that we conducted are described in the following."**

CC1 - 19: Line 122: Isn't this Sackett et al. 1966? With a double t?

Reply CC1 - 19: Yes, thank you, we will correct this.

**Changes for CC1 - 19: We corrected the reference of Sackett et al. 1966.**

**CC1 – 20:** Line 140: don't need a full stop before reference.

**Changes for CC1 - 20: We removed the full stop in I. 144.**

**CC1 – 21:** Line 141: why are data presented as anomalies with respect to the global mean d13C POC value? I think that the authors explained this in lines 149-150 but it is not clear to me what they mean.

**Reply CC1 – 21:** This was due to a misunderstanding during the PANGAEA submission process, since duplicated data must not be archived in PANGAEA. This has now been clarified and corrected and we could now also include the raw data values in the data file provided (see **Reply RC2 - 3**).

**CC1 – 22:** Lines 149-150: what do the authors mean by 'anomalies contain all relevant information...during first steps of model calibration'?

**Reply CC1 – 22:** We have now included the raw data values as well in the data file so clear up this confusion (see **Reply RC2 - 3**).**

**CC1 – 23:** Please make sure that you use the terms coarse/fine and grid/interpolation consistently throughout the manuscript.

**Reply CC1 - 23:** We consequently used the namings "UVic grid" and "WOA grid". The notation are given in the introduction of section 3 as**

**Changes for CC1 – 23:** We changed II. 176-179: "The interpolated data is provided as NetCDF files on two different global grids: a  $1.8 \circ \times 3.6 \circ$  -resolution and 19 depth layers from a model that simulates  $\delta^{13}C_{POC}$  (e.g. Schmittner and Somes, 2016), in the following referred to as the UVic grid, and the  $1 \circ \times 1 \circ$ -resolution and 102 depth layer grid of the World Ocean Atlas (Garcia et al., 2018), in the following referred to as the WOA grid."

The following additional reference was included:

Garcia, H. E., K. Weathers, C. R. Paver, I. Smolyar, T. P. Boyer, R. A. Locarnini, M. M. Zweng, A. V. Mishonov, O. K. Baranova, D. Seidov, and J. R. Reagan, 2018. World Ocean Atlas 2018, Volume 4: Dissolved Inorganic Nutrients (phosphate, nitrate and nitrate+nitrite, silicate). A. Mishonov Technical Ed.; NOAA Atlas NESDIS 84, 35pp.

**CC1 – 24:** Line 165; '...seven different files, where six files contained an individual decade each' or something.

Reply CC1 - 24: We changed this part.

**Changes for CC1 – 24:** We changed II. 212-213: "[...] seven different files each of them independent of time and averaged over the available spatial information. Six of them contain and individual decade each [...]"

**Also, we will update the file description (see **Reply RC1 – 22**) according to the newly updated NetCDF files (see **Reply RC1 – 4**).**

**CC1 – 25:** Lines 165-168: here you should maybe say that: on the coarse grid, data were interpolated independent on time, averaged across depths; interpolation on the fine grid only included data with complete spatio-temporal information, averaged across times and depths.

**Reply CC1 - 25: We added this here**

**Changes for CC1 – 25:** In addition to the **Changes for CC1 – 24:** we added in II. 215-216: "[...] measurements with complete spatial-temporal information, averaged across times and space." See also the updated file descriptions in **Reply RC1 – 22**.

**CC1 – 26:** Lines 168-175: why did you interpolated data onto two different grids? Couldn't you just interpolate on the fine grid and resample if you needed values interpolated on a coarser grid?

**Reply CC1 – 27:** This was a pragmatic decision. In addition to the WOA gridded data we derived the UVic gridded data for our own plans of data-model comparison studies. Since colleagues from the UVic users community stressed the desirability of such data set, we do not want to withhold our data product. Also, a concomitant effect is that the UVic gridded data allow for better visualization on the figures. Many of the isolated grid points were very difficult to see on the global 1x1 degree WOA grid in the manuscript (e.g. Figure 5b). Nevertheless, the full spatial-temporal information is also provided on the finer WOA grid for users that prefer to use it. The large-scale patterns are virtually identical in the two grids.

**CC1 – 28:** Line 176: would be helpful if you could please add a statement to say, in poor words, what the Ferret interpolation does and how that works; essentially a sentence that explains the eqns., briefly.

**Reply CC1 - 28: We will add this.**

**Changes for CC1 – 28:** We added the II. 231-232: "These average the irregularly measured data points within the ocean grid to one single data point representing each covered grid cell."**

4 Main data set characteristics

**CC1 – 29:** Heading: you always use data set, not dataset. Make sure to be consistent.

**Reply CC1 - 29: We will consistently use "data set".**

**Changes for CC1 - 29: We changed this in II. 247, 391.**

**CC1 – 30:** Line 199: before and after the two outer modes, respectively? Density declines to 0 at d13C POC values < than the more negative mode, and at values > than the more positive mode. Is that what you mean here, right?

**Reply CC1 - 30: Yes, we stated this more clearly.**

**Changes for CC1 – 30:** We changed I. 258: "A steep decline to zero is visible outside the two outer modes."**

**CC1 – 31:** Lines 204-205: it would be helpful to mention where these locations are. Also, please change 'the smallest outlier' with the 'most negative/the lowest outlier' or something, as it is not straightforward what you mean here with smallest outlier

**Reply CC1 - 31: In agreement with another reviewer's point (RC2 - 32), we changed this part.**

**Changes for CC1 – 31:** We changed II. 263-267: "[...] measured in September or October 2003, around the location 10° N, 104° W and below 2500 m depth in the vicinity of a hydrothermal field close to the Pacific coast of middle America. The lowest outlier at  $\delta^{13}C_{POC} = -55.15\%$  is taken from Altabet and Francois (2003a) from November 1996 and at 62.52° S, 169.99° E at the ocean surface south from New Zealand."

**CC1 – 32:** Lines 206-2010: again, I think it would be helpful if you could please add where these coordinates are

**Reply CC1 - 32: We added this here.**

**Changes for CC1 – 32**: We added in II. 269-273: "[...] located 30.125° N, 42.117° W in the middle north Atlantic. [...] around 49° N, 130° W close to the American coast of the Pacific [...] Both were measured at the ocean surface in the south Pacific, in July at [...]"

**CC1 – 33:** Line 220: What does it mean that different sampling method could be attributed to 67% of the data as meta information? That 67% of the data had associated sampling method information?

**Reply CC1 - 33: Yes, we stated this more clearly.**

**Changes for CC1 – 33**: We changed II. 282-284: "Around 67 % of the data had associated sampling method information, which are contributed by eighteen different sampling methods."

CC1 - 34: Line 229: double brackets in reference

**Changes for CC1 - 34: We corrected this in I. 293.**

**CC1 – 35:** Lines 232-233: how did you account for spatial sampling bias? What do you mean here with 'by comparing with regions'? Simply that you compared d13C POC data obtained with different method within each region (Figs. 3B-d)?

**Reply CC1 – 35:** That is correct we compared observations only within the different regions as specified in Figure 3. We proposed a clearer description of the analyses in **Reply RC1 – 28**.**

**CC1 – 36:** Line 233: are they? It looks to me that in some cases different sampling methods provide different d13C POC values, e.g., between 30-60 N, bottle values are lower than values obtained with the other sampling methods

**Reply CC1 - 36: Yes. We have rephrased the whole paragraph in Reply RC1 - 28.**

**CC1 – 37:** Line 236-237: please rephrase this sentence as it is very complex and it is not clear to me what you mean

**Reply CC1 - 37: We have rephrased the whole paragraph in Reply RC1 - 28.**

CC1 - 38: Line 239: closely aligned

**Reply CC1 - 38: We have rephrased the whole paragraph in Reply RC1 - 28.**
**CC1 – 39:** In general, how do read from Fig. 3 the % of data collected with each method in each area?

**Reply CC1 – 39:** It is not possible to read this information from the plot. We took it out of the data before we created the plot. This information is given along the plot to give the reader an impression of the ratio of data. We will refine the plot and add the number of data points available for each KDE (see **Reply RC1 – 30**) to make this information easily accessible.**

**CC1 – 40:** Line 240: rephrase with 'the variance... is approx. 3 per mil lower than the variance of all d13C POC values, which is approx. 5 per mil, the highest value observed here', or something.

**Reply CC1 - 40: We rephrased this.**

**Changes for CC1 – 40**: We changed II. 317-318: "The variance of the intake and trap data is  $\approx 3\%$  and lower than the variance of all  $\delta^{13}C_{POC}$  together, which is  $\approx 5\%$ , the highest value observed here."

**CC1 – 41:** Line 242: show a pronounced, remove clearly. Also second is repeated twice. Also show a clear individual maximum, remove mostly.

**Reply CC1 - 41: We have rephrased the whole paragraph in Reply RC1 - 28.**

**CC1 – 42:** Line 245: please consider rephrasing this, e.g., 'we show the spatial distribution of d13C POC measurements across the global ocean surface and depths'. Data is plural, therefore 'most d13C POC data have been measured', please make sure to be consistent with use of plural for data throughout the manuscript.

**Reply CC1 - 42: We changed this part.**

**Changes for CC1 – 42**: We changed II. 325-326: "We show the spatial distribution of $\delta^{13}C_{POC}$ measurements across the global ocean surface and depths. Most $\delta^{13}C_{POC}$ data have been [...]"**

**CC1 – 43:** Line 250: if 80% of the data have associated depth info, depth is a fairly well-recovered metadatum, isn't it? Does that mean that most datapoints don't have associated T and sampling method info?

**Reply CC1 - 43: We removed this sentence to prevent confusion.**

**CC1 - 44:** Line 254: 'within the first 130 m'

**Reply CC1 - 44:** We will corrected this.**

**Changes for CC1 - 44: We changed I. 336: "Within the first 130 m [...]"**

CC1 - 45: Line 255: remove already

**Reply CC1 - 45: We corrected this.**

Changes for CC1 - 45: We changed I. 337: "[...] where nearly 1000 of them [...]"

CC1 - 46: Line 255-256: '200 d13C POC values are available in the depth interval [...)'

**Reply CC1 - 46: We corrected this.**

**Changes for CC1 – 46:** We changed II. 337-338: "200 d13POC values were available in the depth interval [3430 m, 3900 m)."

CC1 - 47: Line 257: add respectively at the end of this sentence

Reply CC1 - 47: We change this.

**Changes for CC1 - 47: We added "respectively" in I. 340.**

**CC1 – 48:** In addition to how many data points there are for each depth layer etc, it would be interesting to see a plot showing trends in d13C POC values with depth, similar to Figs. 6, 8 and 10 for biomes, months and years...

**Reply CC1 – 48:** To present an isight into the vertical distribution of the data we will add a vertical scatterplot of the data. See **Reply CC1 – 2**.**

**CC1 - 49:** Out of curiosity: does the dataset include any d13C POC data for the Mediterranean Sea?

**Reply CC1 – 49:** Yes, it does. In particular, it includes two bottle sampled data points by Carlier et al. (Carlier, A., Le Guilloux, E., Olu, K., Sarrazin, J., Mastrototaro, F., Taviani, M., and Clavier, J.: Stable Isotope composition of particulate organic matter and fauna in a deep Mediterranean cold-water coral bank, https://doi.org/10.1594/PANGAEA.771095, 2011.) from October 2007.

**CC1 – 50:** Line 260: I tend to prefer the use of 'grid' over 'interpolation', as the interpolation is essentially the spatial/horizontal distribution of values which can be done over a grid, isn't it?

Reply CC1 - 50: Yes, exactly. We replaced "interpolation" here with "grid" (see Reply CC1 - 23).

**CC1 – 51:** Line 261: to set context for next sentence, mention here that data are averaged across all depths'.

**Reply CC1 - 51: We clarified this.**

**Changes for CC1 – 51**: We added II. 358-359: "In both cases, we averaged data over all depths information to best visualize the horizontal coverage."

**CC1 – 52:** Line 263, don't need a full stop after Figure 5, but comma.

**Changes for CC1 - 52: We corrected this in I. 361.**

**CC1 – 53:** Line 264: not sure what 'also, data locations of ... occur' means.

**Reply CC1 – 53:** We mean that there are also smaller sample spots visible in the plot. We clarified this.

**Changes for CC1 – 53**: We changed II. 362-363: "Also, smaller sample spots occur, mainly located in the [...]"

CC1 - 54: Line 267: lowest?

**Reply CC1 - 54:** We mean that the data coverage is in comparison very sparse in this area.**

CC1 - 55: Line 269: substitute 'with' with 'of' XX per mil

**Reply CC1 - 55: We will correct this to**

**Changes for CC1 – 55**: We changed I. 368"[...] high values of $\approx -20$ ‰."**

**CC1 – 56:** Line 276: again, make sure to be consistent with use of coarse/fine grid/interpolation.

**Reply CC1 - 56:** We rephrased this.**

**Changes for CC1 – 56**: We changed II. 375-376: "[...] based on the time-independent WOA grid and restricted to [...]"

**CC1 – 57:** Line 279-280: description of colors and lines should be given in figure caption, not in the main text.

**Reply CC1 - 57: We moved this description to the caption.**

**CC1 – 58:** Line 283: what does 'but 14' mean? Biomes were numbered from 9 to 17, where 15-17 had to be cut to the given lateral range. Also, consider using longitudinal rather than lateral here.

**Reply CC1 - 58: We changed this part.**

**Changes for CC1 – 58**: We changed II. 382-386: "[...] cover the Atlantic Ocean and extend to the Arctic Sea and parts of the Southern Ocean. The biomes are numbered 9 to 17, excluding 14. The biomes 15 to 17 are representing parts of the Southern Ocean and were restricted to 70° W and 20° E. Their locations are shown in Figure 7."

**CC1 – 59:** Line 284: I think there is a mistake here, location of biomes in the Atlantic is shown in Fig. 6c, not Fig. 10.

**Changes for CC1 - 59: Yes, indeed. We corrected this in I. 386.**

**CC1 – 60:** Lines 285-287: please insert per mil symbol. Also consider changing 'the final biomes' with 'the biomes with more positive d13C POC values' or something.

Reply CC1 - 60: We added the per mil symbol and rephrased the last sentence.

**Changes for CC1 – 60**: We changed II. 389-390: "The biomes with more positive  $\delta^{13}C_{POC}$  are in the lower latitudes and show similarly higher values from -23 to -21‰."

**CC1 - 61:** Line 293: Fig. 7 here, not 5.

**Reply CC1 - 61: We corrected this.**

CC1 - 62: Line 295-296: latest data are (plural), and again in the following line

**Reply CC1 - 62: We corrected this throughout the manuscript (see Changes for CC1 - 4).**

6.1 Seasonal trends

**CC1 – 63:** You don't describe Figs. 8b,d but I think they're informative as they show the seasonal trend in d13C POC values.

**Reply CC1 - 63:** In agreement with another reviewer's point **(RC1 - 39)** we moved panels (b) and (d) to a table.**

**CC1 – 64:** Also, please make sure the distinction between winter/summer is clear for the /S hemisphere in this paragraph.

**Reply CC1 – 64:** We agree that this part was a bit confusing. We discussed the point more precise and rewrote the whole paragraph according to **Reply RC2 – 44**.**

**CC1 – 65:** Line 328-329: remove 'both'. Also second is repeated twice.

Reply CC1 - 65: We corrected this.

**Changes for CC1 – 65:** We changed II. 449-450: "All, but the 1980s show one clear maximum in their approximated densities. The 1980s show a second expressed density maximum [...]"

**CC1 - 66:** Lines 329: main maximum shift or the shift in main maxima. Remove 'with every decade lower'.

Reply CC1 - 66: We rephrased this.

**Changes for CC1 – 66**: We changed II. 450-451: "The main maximum shift from the 1960s at  $\delta^{13}C_{POC}$  ≈ -19.9‰ to the 2010s at  $\delta^{13}C_{POC}$  ≈ -23‰."

**CC1 – 67:** Lines 334-339: I would remove these lines, if you really want to keep Figs. 10c,d in. Or you could also remove the figures and just say that there are not enough data in the SO to investigate multi- decadal trends.

**Reply CC1 – 67:** We will remove the panels (b) and (d) and only show (a) and (c), where we can use the last one to emphasize the need for more Southern Ocean data. Also, we will add the number of used data points for each KDE (see **Reply RC1 – 30**). We would move (b) to an own figure to present the visible decrease and add a shaded area for the variance around the graph to give an insight in the certainty as you suggested in **CC1 – 81** for a similar plot.

**CC1 – 68:** It would be nice to have a paragraph in Conclusions with examples of research and technical questions that could be tackled/answered with datasets as such. These applications should link back to themes presented in the Introduction.

**Reply CC1 – 68:** We added a statement in the Conclusions about some of the broader applications of this dataset. This shall be placed at the end of the first paragraph as:

**Changes for CC1 – 68:** We added: II. 468-471: "This new  $\delta^{13}C_{POC}$  data set provides the best coverage to date that will be a useful tool to help constrain many marine carbon cycling processes

**and pathways from ocean-atmosphere exchange to marine ecosystems, as well as to better understand observations and validate models."**

**CC1 – 69:** Additionally – and this my reflect my own research interests – I think that the authors should stress the importance of their dataset for calibration/validation of process-based, mechanistic models. A major issue related with the application of these models in ecology, for instance, has been the lack of suitable calibration/validation datasets, resulting in large and mostly unknown uncertainties (models trusted more than data, as they're based on fairly well-understood mechanisms whereas data are scarce and often incomparable).

**Reply CC1 - 69:** We added this to the introduction.

**Changes for CC1 – 69:** We added II. 80-81: "But until today, there is a lack of suitable data sets as constraints. This results in large and mostly unknown uncertainties in model results."

We added in II. 217-219: "One major aim of this work is to support reliable validation and calibration of  $\delta^{13}C_{POC}$  -simulating models. Hence, we chose for the coarser interpolation the grid of the version 2.9 UVic model, as used e.g. in Schmittner and Somes (2016)."

**CC1 – 70:** Datasets like this one provide a validation tool for mechanistic model, and potential for the development of data-based models of the spatio-temporal distributions of stable isotopes in marine ecosystems. An approach that has been successfully used to develop data-based isoscapes is the INLA method (St John Glew et al. 2019 MEE, St John Glew et al. 2020 ESSOAr), which allows separating spatial from non-spatial components of isotope variance when predicting spatial isotope patterns. This dataset could be suitable for such approach as it contains some meta information (e.g., sampling method, depth, month, decade, etc.) which can be included as factor to estimate non-spatial variance when predicting spatial variation from environmental covariate sets.

**Reply CC1 – 70:** Thank you very much for this idea! We rephrased the last sentence to emphasize this point and pointed it out in the introduction of the interpolated data sets.

**Changes for CC1 – 70**: We added II. 500-502: "The data set shows promise to better understand, constrain and predict carbon cycling as it provides a validation tool for mechanistic models and supports separation of non-spatial components in  $\delta^{13}C_{POC}$  variations."

We rephrased II. 217-218: "One major aim of this work is to support reliable validation and calibration of  $\delta$ 13CP OC -simulating models. Hence, we chose the grid of the UVic model version 2.9, as used e.g. in [...]"

**CC1 – 71:** Table 1 & others: don't need a capital letter after semi-column.

Reply CC1 - 71: We corrected this.

**Changes for CC1 - 71: We corrected this in all captions.**

CC1 - 72: The second column lists in which ..., without comma after lists.

Reply CC1 - 72: We removed this.

**Changes for CC1 - 72: We removed the comma in the caption of Table 1.**

**CC1 – 73:** The third and fourth columns, plural;

**Reply CC1 - 73: We correct this.**

**Changes for CC1 - 73: We corrected this in the caption of Table 1.**

CC1 - 74: also unnecessary comma between show and from;

Reply CC1 - 74: We remove this.

Changes for CC1 - 74: We corrected this in the caption of Table 1.

CC1 - 75: also show from what values to which values (or something).

**Reply CC1 - 75: We corrected this.**

**Changes for CC1 – 75**: We changed in the caption of Table 1 "[...] from what values to which values [...]"

**CC1 - 76:** Table4: change inspired with based on, or something.

**Reply CC1 - 76: We change Table 4 to a histogram according to RC1 - 35.**

CC1 - 77: Not sure what the sentence starting with 'below 50 m...' means, why only below 50 m?

**Reply CC1 – 77:** The first 50 m are only one layer in the presented grid. To better resolve the distribution within this, we split it up. See also **Reply CC1 – 76.**

**CC1 – 78**: In the last sentence, depth range not depths range.

**Reply CC1 - 78: See Reply CC1 - 76.**

**CC1 – 79**: Figure 1: in caption don't need capital V for values after semi-column. Please fix this here and in other figs' captions and throughout text.

**Reply CC1 - 79: We will fixed this here and in the other captions as well.**

**Changes for CC1 - 79: We corrected this in all captions.**

**CC1 – 80:** Figure 6b: can you plot mean lat for each biome on the x-axis, rather than biome number? Or at least an arrow N to S below the x-axis? You need to make this fig as much self-explanatory as possible.

**Changes for CC1 - 80: We added the mean latitudes from the biomes as the labels.**

We changed the caption from Figure 7 (former 6): "North-south trend of sampled  $\delta$ 13CPOC values is visualized by a cross section over the Atlantic ocean. Biomes (Fay and McKinley, 2014) define the latitudinal bands of the interpolated data set. (a) presents for each biome a Gaussian KDE approximating the density of the contained  $\delta$ 13CPOC data. Different colors mark the individual biomes and a black line shows the general global  $\delta$ 13CP OC distribution. The number in brackets in each KDE label counts the number of  $\delta$ 13CP OC measurements used for the respective graph. (b) shows in a box plot the steep decline of  $\delta$ 13CP OC values from the tropical biomes towards the higher latitudes. The x-axis provides the mean latitudes of the biomes introduced in (a). The y-axis measures the  $\delta$ 13CP OC value. (c) shows the biomes locations. Each biome is drawn in the color of its corresponding density estimate in (a) above. The biome numbers increase from the north to the south."

**CC1 – 81:** Also, b can you plot some confidence intervals around means in panel b, given by variance of KDE? Alternatively, you could plot boxplots of for each KDE, without black line connecting means.

**Changes for CC1** We changed this panel to a boxplot, but also appreciate the idea of the shaded variances and used this for Figure 11.**

**CC1 - 82:** Figure 8b,d: Similar comments to Fig. 6b.

**Changes for CC1 – 82:** We discarded panels (b) and (d) and show the values in a 2-rows table following another reviewer's point (see **RC1 – 39**).**

**CC1 – 83:** Figure 9: in caption, 'grid-locations of d13C POC data, colored by sampling decades' or something. Find a clear way to say that the grids of sample locations are shown here, colored by the decades in which the samples were collected.

**Changes for CC1 – 83:** We rephrased this as: "Grid locations of the $\delta^{13}C_{POC}$ data, colored by sampling decades."**

CC1 - 84: Aren't there any grids with multiple samples collected in different decades?

**Reply CC1 - 84: Yes, this might happen. The plot sequence was 1960s to 2010s. We clarified this.**

**Changes for CC1 – 84:** We changed in the caption from Figure 10: "The different colors indicate the different sample decades and were plotted increasing in time above each other."**

**CC1 - 85:** Figure 10b,d: Similar comments to Figs. 6b and 8b,d.

Also, wouldn't show panels c,d for Southern ocean, but just mention in the text that the SO was excluded from analysis as available data are sufficient to derive KDE for only three decades. If you really want to keep panels c,d in for consistency and as justification for insufficient data in the SO, then don't describe patterns in the main text.

**Reply CC1 – 85:** We removed the panels (b) and (d) and only show (a) and (c), where we can use the last one to emphasize the need for more Southern Ocean data. We moved (b) to an own figure to present the visible decrease. This should then show a shaded area of the variances around the graph as described above (see **Reply CC1 – 67**).

**Changes for CC1 – 85:** Figure 11 only shows panels (a) and (c), now. Panel (b) was moved to Figure 12 and added a shaded area for the variance around the median.

We changed the caption from Figure 11: "The decadal shift of  $\delta$ 13CP OC values for all, but the Southern Ocean (a) and only the Southern Ocean (b) shown by estimated densities of  $\delta$ 13CPOC values. The differently colored graphs refer to the individual decades. Southern Ocean data are sparsely covered and does not provide enough data for a reasonable comparison."

We added to Figure 12 the caption: "The decadal shift of  $\delta$ 13CP OC values in the uppermost 130 m for all, but the Southern Ocean:  $\delta$ 13CP OC decadal median against the decades. The shaded area around the graph marks the variance of the respective decade in each direction."

We rephrased II. 441-445: "[...] available decades by density estimates in Figure 11 (see also Table A5 in the Appendix) and by their median in Figure 12. The first visualize the sparse coverage of the Southern Ocean outside of the 1990s, which is why it is not part of any further discussion here."

In I. 446 we deleted "here".

In I. 453 we deleted "1960s"

We rephrased I. 454: "[...] construct a comparable KDE. Due to this very [...]"

We deleted II. 455-460.

**CC1 – 86:** Panel b: why does the y-axis go down to -30 per mil, when the minimum mean d13C POC value > - 24 per mil?

**Reply CC1 - 86:** We made both rows of this plot share both axis for better comparability. We feared, if we would give the upper row (panel b) a different scale than the lower (panel d), this would lead to false interpretation of the relative magnitude of both changes. But according to **Reply CC1 - 85** we will discard this panel, since it does not contain enough meaningful information.

Changes for CC1 - 86: see Changes for CC1 - 85.

CC1 - 87: I have seen that dataset is stored in Pangaea; do you also plan to submit it to Isobank?

**Reply CC1 – 87:** We contacted Brian Hayden many months ago about this possibility, but he told us that Isobank was, at that time, not yet ready. We will follow up on this again after publication.

**Description of a global marine particulate organic carbon-13 isotope data set**

Maria-Theresia Verwega1, 2, Christopher J. Somes1, Markus Schartau1, Robyn E. Tuerena3, Anne Lorrain4, Andreas Oschlies1, and Thomas Slawig2

1GEOMAR - Helmholtz Centre for Ocean Research, Kiel, Germany
 2Kiel University, Kiel, Germany

[revised manuscript text omitted]

named meta information. Suspicious or rounded Rounded values were adjusted to their source values as well as data with 155 interchanged longitudinal information, which is in detail shown in Table 1.

Averaging was only applied for depth ranges, these were included as their arithmetic mean. Sample timeframes were only included when lying completely within one month and year. Sample depth given as "surface" was denoted as 1.

Wherever multiple types of In two cases we identified multiple  $\delta^{13}C_{POC}$  e.g. similar measurements based on different methods, were given within one source, we chose only one type. In Westerhausen and Sarnthein (2003)data sets from a

- single event (time, place, investigator), where the data had been subject to different stages of processing or different types 160 of measurements: In Westerhausen and Sarnthein (2003), we chose the "mass spectrometer" data set because this was the originally measured one. In Trull and Armand (2013a) and in Trull and Armand (2013b) Trull and Armand (2013a) and in Trull and Armand (2013b) , we used the "blanc corrections" data set of  $\delta^{13}$ C, since this set of  $\delta^{13}$ Corg values is recommended to be considered (Trull and Armand, 2001).
- The primary source of the Tuerena and Lorrain data is was mentioned in our data set in the "Project/cruise" column. In the 165 data set from Tuerena et al. (2019), this was originally labeled as "source", in the Lorrain data set as "campaign". In both data sets the Longitude was converted to  $[-180^\circ, 180^\circ]$  from a  $[0^\circ, 360^\circ]$  format <del>. We used the transformation</del>

$$Long_{new} = \begin{cases} Long_{old} - 360^{\circ} & \text{ for all } Long_{old} \in (180^{\circ}, 360^{\circ}] \\ Long_{old} & else \end{cases}$$

by Equation 2. In the data of MacKenzie et al. (2019) MacKenzie et al. (2019) we deleted a typo where the depth value was set 170 equal to the negative Longitude value. We disregarded trap duration given in Voss and von Bodungen (2003) Voss and von Bodungen (2003) , which was given as the negative value -1.

**Content and structure of the data set 3**

The data collection is made available in files of raw and interpolated values respectively -(Verwega et al., 2021). The raw data is are a csv file that includes the anomalies of the  $\delta^{13}C_{POC}$  measurements, their anomalies with respect to their mean and all 175 available meta information. The interpolated data is are provided as NetCDF files on two different global grids: a  $1.8^{\circ} \times 3.6^{\circ}$ resolution grid and 19 depth layers from a model that simulates  $\delta^{13}C_{POC}$  (e.g. Schmittner and Somes, 2016), in the following referred to as the UVic grid, and the  $1^{\circ} \times 1^{\circ}$ -resolution and 102 depth layer grid of the World Ocean Atlas –(Garcia et al., 2018) , in the following referred to as the WOA grid. Interpolation required availability of full spatial information (latitude, longitude and depth) of included  $\delta^{13}C_{POC}$  data to locate them on the grid.

180

On the WOA grid we provide thirteen NetCDF files containing only data with full spatio-temporal metadata: One is, averaging all observations from each year together, each year accounting for a time increment on the time axis. The other twelve files are averaging only observations from an individual month with again each year accounting for a time increment on the time axis. These files provide a variety of analysis opportunities, but also limited content of  $\delta^{13}C_{POC}$  data.

Table 1. Changes that were introduced to data taken from Goericke (1994)Goericke (1994): The the first column names the publication or author of the primary data set. The second column lists - in which part of the data we applied changes. The third and fourth columns show - from what values to which values they have been changed and the last columns gives the reason for this.

[revised manuscript text omitted]

220 it consists of  $100 \times 100$  cells with a resolution of  $1.8^{\circ} \times 3.6^{\circ}$ , arranged from 0 to  $360^{\circ}$  in longitude (LON) and -90 to  $90^{\circ}$  in latitude . The vertical grid (LAT). Vertically, it is split up into 19 vertical layers (DEPTH), decreasing in resolution with depth. The two uppermost layers reach down to depths of 50 and 130 m respectively, which represent m respectively, and they are supposed to comprise the upper ocean's euphotic zone.

The finer interpolation was carried out the WOA grid is based on the  $1^{\circ} \times 1^{\circ}$  grid of the World Ocean Atlas (Garcia et al.,

- 225 2018)and. It has a horizontal resolution of 360 arranged from -180 to  $180^{\circ}$  in longitude (LON) and 180 arranged from -90 to  $90^{\circ}$  in latitude (LAT) direction. Vertically, it is split up into 102 layers (DEPTH). The time axis (TIME) increments for each year from 1964 to 2015 by one and has a size of 52. This interpolation only includes  $\delta^{13}C_{POC}$  data with full spatio-temporal metadata coverage, i.e. additional to latitude, longitude and depth, we also required and included year and month information. Here, the provided NetCDF file includes also the year ranging from 1964 to 2015 on the l/t axis and the month ranging from 1
- 230 to 12 on the m/e axis.

235

FERRET scripts were used for the interpolations. These averaged the irregularly measured data points within the ocean grid to one single data point representing each covered grid cell. The interpolation function SCAT2GRIDGAUSS FERRET function by NOAA's Pacific Marine Environmental Laboratory by NOAA's Pacific Marine Environmental Laboratory performed the spatial averaging under PyFerret v7.5. Calculations in this functions function are based on a work by Kessler and McCreary (1992) Kessler and McCreary (1992) and can be summarized as follows: Let let  $(x_1, y_1), ..., (x_n, y_n) \subseteq \mathbb{R}^2$  be an equidistant grid and

 $(\tilde{x_1}, \tilde{y_1}), ..., (\tilde{x_m}, \tilde{y_m}) \subseteq \mathbb{R}^2$  be irregular measurement locations of a real tracer  $D_j, j \in \{1, ..., m\}$ . Then the value  $D_i \in \mathbb{R}$  at grid point  $(x_i, y_i), i \in \{1, ..., n\}$  becomes interpolated as

$$D_{i} := \frac{\sum_{j=1}^{m} D_{j} W_{i,j}}{\sum_{j=1}^{m} W_{i,j}}$$
(4)

where

245

265

240
$$W_{i,j} := \begin{cases} 0; & \tau_{i,j} < e^{-CX} \\ 0; & \tau_{i,j} < e^{-CY} \\ \tau_{i,j}; & else \end{cases}$$
(5)

with  $\tau_{i,j} := \exp\left(-\left(\frac{(x_j - x_i)^2}{X^2} + \frac{(y_j - y_i)^2}{Y^2}\right)\right)$  is the Gaussian weight function and  $X, Y \in \mathbb{R}$  are scaling arguments and  $C \in \mathbb{R}$  the cut-off parameter. We set to X = 1.8, Y = 0.9 and C = 1 in the our script.

Since the interpolation into the finer\_WOA grid excluded all data without full spatio-temporal metadata coverage, we focus following descriptions of interpolated data on the coarse-UVic grid interpolations. These also include data without month-information in the six decadal files and even completely without temporal information in the seventh time-independent file.

**4 Main dataset data set characteristics**

The final data set includes 4732 individual δ13CPOC measurements of seawater samples. We show the distribution of δ13CPOC values by Gaussian kernel density estimation (KDE) in Figure 1. KDEs are a non-parametric density estimation (Silverman, 1986) for approximation of probability density functions, which is theoretically similar to a histogram but with a continuous curve not dependent on rigid intervals. We applied a Python implementation from the SciPy stats-package (Virtanen et al., 2020) to create the results presented here. Likewise, we derived conditional probability densities of δ13CPOC values, given the different measurement method applied (Figure 3).

**4.1 Range and outlier values**

- The data distribution is presented by its KDE in Figure 1. The interval of  $\delta^{13}C_{POC}$  values ranges over [-55.15, -4.5] with a mostly smooth distribution. Most of our data exhibit values around  $\delta^{13}C_{POC} \approx -24\%$ , which becomes clearly identifiable as a single maximum in the KDE. Two smaller modes are visible at around  $\delta^{13}C_{POC} \approx -27.5\%$  and  $\delta^{13}C_{POC} \approx -22\%$  (see also Table A1 in the Appendix). A steep decline to zero follows after is visible outside the two outer modes. The steep decline of the KDE stops at around  $\delta^{13}C_{POC} \approx -14\%$ . Between  $\delta^{13}C_{POC} \approx -37\%$  and  $\delta^{13}C_{POC} \approx -55.15\%$
- 260 as well as between  $\delta^{13}C_{POC} \approx -14\%$  and  $\delta^{13}C_{POC} \approx -4.5\%$  the KDE closely aligns to the x-axis, what which indicates very little data points lying lie in this range.

Below  $\delta^{13}C_{POC} = -37\%$  we find 17 data points ranging down to  $\delta^{13}C_{POC} = -55.15\%$ . Down to  $\delta^{13}C_{POC} = -48\%$  these were all taken from Lein and Ivanov (2009) and Lein et al. (2006) Lein and Ivanov (2009) and Lein et al. (2006), measured in September or October 2003and, around the location 10° N, 104° W. The smallest and below 2500 m depth in the vicinity a hydrothermal field close to the Pacific coast of middle America. The lowest outlier at  $\delta^{13}C_{POC} = -55.15\%$  is